# Sphingolipid metabolism orchestrates establishment of the hair follicle stem cell compartment

Franziska Peters[1,2,3] [ID], Windie Höfs[2]* [ID], Hunki Lee[2]* [ID], Susanne Brodesser[4] [ID], Kai Kruse[5] [ID], Hannes C.A. Drexler[5] [ID], Jiali Hu[3,6] [ID], Verena K. Raker[7] [ID], Dominika Lukas[6] [ID], Esther von Stebut[6] [ID], Martin Krönke[4,8,9] [ID], Carien M. Niessen[3] [ID], and Sara A. Wickström[1,2,10] [ID]

**Sphingolipids serve as building blocks of membranes to ensure subcellular compartmentalization and facilitate intercellular communication. How cell type–specific lipid compositions are achieved and what is their functional significance in tissue morphogenesis and maintenance has remained unclear. Here, we identify a stem cell–specific role for ceramide synthase 4 (CerS4) in orchestrating fate decisions in skin epidermis. Deletion of CerS4 prevents the proper development of the adult hair follicle bulge stem cell (HFSC) compartment due to altered differentiation trajectories. Mechanistically, HFSC differentiation defects arise from an imbalance of key ceramides and their derivate sphingolipids, resulting in hyperactivation of noncanonical Wnt signaling. This impaired HFSC compartment establishment leads to disruption of hair follicle architecture and skin barrier function, ultimately triggering a T helper cell 2–dominated immune infiltration resembling human atopic dermatitis. This work uncovers a fundamental role for a cell state–specific sphingolipid profile in stem cell homeostasis and in maintaining an intact skin barrier.**

## Introduction

The skin epidermis is a life-essential bidirectional barrier that protects animals from extrinsic insults, including pathogens, toxins, and irradiation, while preventing water loss from inside the body. Disruption of the barrier leads to a strong innate immune reaction, and when unrepaired, leads to chronic inflammatory skin diseases such as atopic dermatitis (AD) (Eyerich et al., 2015; Paternoster et al., 2015; Weidinger and Novak, 2016).

Given their function as a first line of defense, skin keratinocytes sustain constant damage, and to maintain tissue integrity, these damaged cells are removed via terminal differentiation and subsequent tissue turnover rather than apoptosis (Kato et al., 2021). This homeostatic self-renewal process is driven by stem cells (Tumbar et al., 2004; Blanpain and Fuchs, 2009; Schneider et al., 2009; Blanpain and Fuchs, 2014, Hsu et al., 2014; Gonzales and Fuchs, 2017, Chacón-Martínez et al., 2018). The epidermis contains multiple stem cell populations that are required for compartment-specific tissue maintenance and repair. While the basal epidermal stem cells drive self-renewal of the interfollicular epidermis (IFE), the hair follicle and sebaceous glands (SGs) are maintained by distinct stem cells,

the hair follicle stem cells (HFSCs) residing within the bulge stem cell niche (Hsu et al., 2011; Blanpain and Fuchs, 2014, Chacón-Martínez et al., 2017; Geueke and Niemann, 2021). While the key role of the bulge HFSCs in maintaining the renewal of the hair follicle and the hair shaft itself is well established, how this adult stem cell compartment becomes established during postnatal development is less clear.

The capacity of stem cells to self-renew or differentiate can be attributed to distinct metabolic states. Emerging evidence suggests that lipid metabolism plays a fundamental role in stem cell homeostasis (van Gastel et al., 2020), but the mechanisms are unclear. Ceramides, synthesized by ceramide synthases (CerS), are the essential building blocks of all cellular membranes and can also serve as signaling molecules. Ceramides have been implicated in regulating the balance between self-renewal and differentiation of neuronal stem cells, influencing signaling pathways that control cell fate decisions (Bieberich et al., 2003; Wang et al., 2008; He et al., 2014). CerS2–6 are expressed in mammalian skin (Levy and Futerman, 2010). CerS3 is essential for the formation of the functional epidermal barrier

---

[1]Stem Cells and Metabolism Research Program, Faculty of Medicine, University of Helsinki, Helsinki, Finland; [2]Department of Cell and Tissue Dynamics, Max Planck Institute for Molecular Biomedicine, Münster, Germany; [3]Department Cell Biology of the Skin, Cologne Excellence Cluster on Cellular Stress Responses in Aging Associated Diseases, Center for Molecular Medicine Cologne, University Hospital Cologne, University of Cologne, Cologne, Germany; [4]Faculty of Medicine and University Hospital of Cologne, Cluster of Excellence on Cellular Stress Responses in Aging Associated Diseases, University of Cologne, Cologne, Germany; [5]Max Planck Institute for Molecular Biomedicine, Münster, Germany; [6]Department of Dermatology, University of Cologne, Cologne, Germany; [7]Department of Dermatology, University of Münster, Münster, Germany; [8]Institute for Medical Microbiology, Immunology and Hygiene, University of Cologne, Cologne, Germany; [9]Center for Molecular Medicine Cologne, University of Cologne, Cologne, Germany; [10]Helsinki Institute of Life Science, Biomedicum Helsinki, University of Helsinki, Helsinki, Finland.

*W. Höfs and H. Lee contributed equally to this paper. Correspondence to Sara A. Wickström: sara.wickstrom@mpi-muenster.mpg.de; Franziska Peters: anna.peters@uk.bonn.de.

through the synthesis of extracellular ω-hydroxylated ultra-long-chain ceramides, crucial components of the outermost layer of the skin, the stratum corneum (Jennemann et al., 2012; Eckl et al., 2013; Radner et al., 2013). CerS4 has been shown to regulate SG homeostasis, hair follicle cycling, and homeostatic epidermal barrier function (Ebel et al., 2014; Peters et al., 2015, 2020). CerS4 has the highest substrate specificity toward long-chain ceramides, Cer18:0–22:0, key molecules in intracellular sphingolipid metabolism and essential building blocks of all cellular membranes (Tidhar et al., 2018). However, how CerS4 regulates epidermal homeostasis and HFSC dynamics have remained unclear.

Here, we show that CerS4 is crucial for sphingolipid metabolism required to guide stem cell lineage progression toward the establishment of the adult HFSC compartment. Deleting CerS4 in epidermal stem cells leads to defective establishment of the HFSC state, and these cells are inappropriately routed into upper hair follicle (uHF) and inner bulge (IB) fates due to aberrant noncanonical Wnt signaling. As a result, hair follicle architecture and barrier function are disrupted, leading ultimately to a T helper type 2 (Th2) cell immune dominance with resemblance to AD. Collectively, these data demonstrate a central role for lipid metabolism in tuning signaling required for stem cell fate regulation and the essential role of the HFSCs in maintaining the skin barrier.

## Results

### CerS4 is required for the establishment of the HFSC compartment

Our previous work has identified a central role for CerS4 in hair follicle cycling (Ebel et al., 2014; Peters et al., 2015, 2020). To investigate the role of sphingolipid metabolism in controlling HFSC behavior and to understand if HFSC intrinsic alterations in sphingolipid metabolism are associated with skin barrier disease, we carried out detailed analysis of the epidermis in mice where CerS4 was deleted in all epidermal stem cell compartments and their differentiated progeny using keratin-14-Cre (Hafner et al., 2004; Peters et al., 2015) (from here on CerS4$^{epi-/-}$). Histological analyses in early postnatal mice revealed no obvious differences in the morphology of the IFE (Fig. S1 A), but the pilosebaceous unit of these adolescent CerS4$^{epi-/-}$ mice showed first abnormalities at the first hair follicle catagen phase at postnatal day (P)16, with a delayed entry into catagen (Fig. 1 A and Fig. S1 A). This delay was associated with aberrant expression of cell fate regulators such the transcription factor Lhx2 that is essential for maintaining HFSC quiescence and ensuring linage stability (Folgueras et al., 2013). Lhx2 showed reduced expression in the hair follicle, whereas keratin-6, a marker for the companion layer and epidermal stress, was upregulated in the basal layer of the hair follicle (Fig. S1, B and C). Despite the delay in the catagen regression phase, both control and CerS4$^{epi-/-}$ hair follicles reached the telogen-resting phase at P19, but structural abnormalities, including increased size of SGs persisted (Fig. 1 A).

To understand if these structural abnormalities of the pilosebaceous unit were associated with defects in the HFSC compartment, we quantified levels of CD34$^+$ integrin-α6$^+$ HFSCs from P21, P47, 6-mo, and 1-year-old control and CerS4$^{epi-/-}$ mice (Fig. 1 B). Interestingly, CerS4$^{epi-/-}$ mice showed reduced HFSC numbers already at P21 during postnatal hair morphogenesis when the HFSC compartment is becoming established (Fig. 1 C), and this reduction became even more prominent over time, coinciding with the previously reported progressive hair loss phenotype (Fig. 1 C and Fig. S1 D) (Peters et al., 2020). Consistent with the morphological defects and progressive stem cell loss, analyses of quiescence of bulge cells using in vivo BrdU incorporation revealed no strong alterations at P21 but loss of quiescence in the next telogen phase at P47 (Fig. S1, E and F), consistent with previous observations (Peters et al., 2015).

To understand the role of CerS4 in the establishment of the HFSC compartment and in epidermal homeostasis more broadly, we isolated dorsal epidermis from P19 control and CerS4$^{epi-/-}$ mice when both mice had reached telogen and generated single-cell transcriptome libraries (single-cell RNA sequencing [scRNAseq]; BD Rhapsody). Two mice/genotype were processed and sequenced. After individual quality control and analyses of all mice separately, replicates were merged and analyzed together (17,577 control cells/17,096 CerS4$^{epi-/-}$ cells) to capture all cell populations. Cell types and states were identified based on cluster-specific gene expression and annotated based on previously published marker gene profiles (Joost et al., 2016). Our analyses revealed that the majority of the cells derived from the IFE. The IFE was divided in basal cells (IFE basal I–IV), differentiated (IFE D I–II), and keratinized (IFE K I–II) clusters. Furthermore, all major subsets of the hair follicle were detected and clustered. We detected two outer bulge (OB) stem cell compartments (OB I–II), the IB, and the uHF (uHF I–III), as well as sebocytes (SG). Further, Langerhans cells (LH) and T cells (TC) formed the two non-epidermal clusters (Fig. 1 E). Importantly, these hair follicle populations showed alterations upon loss of CerS4, characterized by a striking expansion of the OB stem cell (OB I), the uHF (uHF I–II), and SG populations in CerS4$^{epi-/-}$ skin (Fig. 1, D and E; and Fig. S2 A).

CerS4 transcripts were detected in the hair follicle and in SG clusters, albeit with low levels as expected for an enzyme, as also validated by in situ RNA hybridization (Fig. S2, B and C). This hair follicle–dominant expression pattern was confirmed by quantitative RT-PCR (qRT-PCR) analysis of FACS-purified (1) Sca1$^-$ CD34$^+$ integrin-α6$^+$ HFSCs, (2) Sca1$^-$ CD34$^-$ integrin-α6$^+$ hair follicle progenitors, and (3) Sca1$^+$ CD34$^-$ integrin-α6$^+$ IFE progenitors, where CerS4 mRNA expression was restricted to the HFSC compartment (Fig. S2 D).

As our data indicated high CerS4 mRNA expression in SGs, and a previous study had suggested that the hair follicle phenotype of CerS4-deficient mice results from altered sebum composition (Ebel et al., 2014), we asked if the observed hair follicle phenotypes were a result of altered SG function in CerS4-deficient mice. To this end, we deleted CerS4 specifically in the mature SG using SCD3-Cre (Dahlhoff et al., 2016). Strikingly, mice lacking CerS4 in mature SGs (from here on CerS4$^{SG-/-}$) did not display any obvious epidermal or hair follicle phenotypes even in aged mice. These mice showed no macroscopic signs of alopecia, and careful histological analysis of skin sections

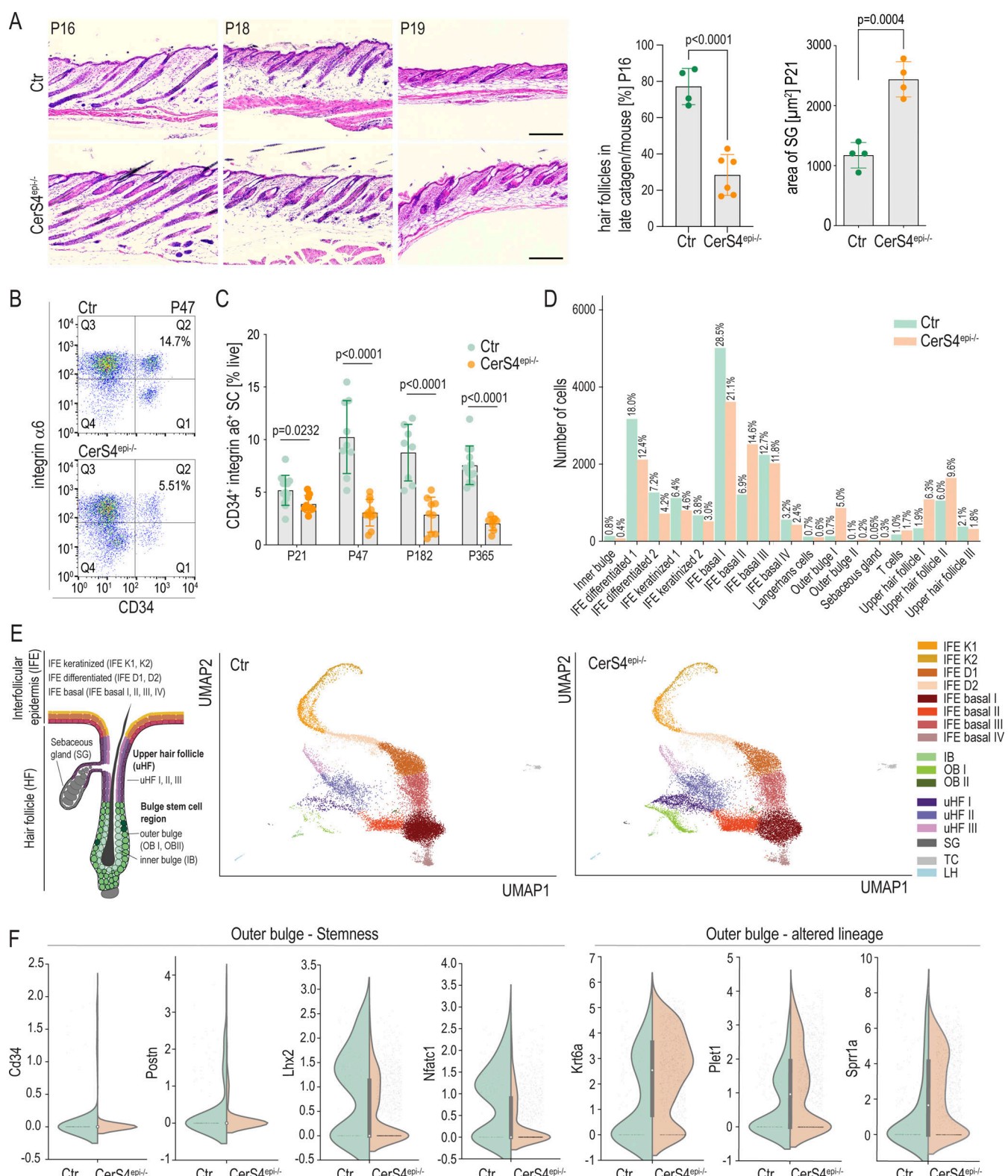

Figure 1. **CerS4 is required for the establishment of the HFSC compartment. (A)** Representative images and quantification of hair follicle stage of H/E-stained back skin sections of control (Ctr) and CerS4epi−/− back skin at P16, 18, and 19 showing a developmental delay in hair follicle morphogenesis and enlarged SGs (scale bars 100 μm; n = 4–6 mice/genotype; unpaired t test). **(B)** Representative FACS plots of P47 murine epidermal cells stained for CD34 and integrin α6. **(C)** FACS-mediated quantification of CD34+ integrin α6+ HFSCs at P21 (n = 13 Ctr, 9 CerS4epi−/− mice), P47 (n = 9 Ctr, 11 CerS4epi−/− mice), P128 (n = 8 Ctr, 9 CerS4epi−/− mice), and P365 (n = 12 Ctr, 8 CerS4epi−/− mice). Note progressive loss of stem cells in CerS4epi−/− mice (mean ± SD; unpaired t test). **(D)** Cell counts of cell clusters determined by scRNAseq. **(E)** Schematic illustration of the IFE and the pilosebacous unit containing the SG and the hair follicle and its bulge stem cell compartment. Keratin-14-Cre is expressed in basal cells throughout the tissue, allowing keratin-14-Cre–mediated deletion of CerS4

(CerS4[epi−/−]) in all stem cell compartments. Schematic and UMAP projections of cells identified using scRNAseq. Note comparable presence of all epidermal cell states but increased OB population in CerS4[epi−/−] mice (*n* = 2 mice/genotype, pooled). **(F)** Gene expression of indicated stem and progenitor cell marker genes in the OB stem cell compartment in Ctr and CerS4[epi−/−] skin. IFE = interfollicular epidermis, IB = inner bulge, OB = outer bulge, uHF = upper hair follicle, SG = sebaceous gland, TC = T cells, LH = Langerhans cells.

showed no alterations in hair follicle morphology or structural abnormalities of the IFE (Fig. S2, E–H). Finally, flow cytometry analyses of freshly isolated epidermal cells from P47, P58, and 6-mo-old control and CerS4[SG−/−] mice showed comparable levels of CD34+ integrin-α6+ HFSCs (Fig. S2 I). The data demonstrate that CerS4 function in mature SGs is not responsible for the observed phenotypes in the HFSC compartment nor is it necessary to control structural integrity of the hair follicle, the SG, or IFE.

We thus focused our attention to the HFSC compartment, in particular on the observed expansion of the OB stem cell population (OB I) in the CerS4[epi−/−] mice. Gene expression profiling of key hair follicle OB stem cell markers and transcriptional master regulators, such as *Cd34* and *Postn*, revealed that their expression was slightly reduced in CerS4-deficient OB cells. In particular, levels of *Lhx2*, already observed to be altered on the protein level, as well as another master regulator of quiescent HFSCs, *Nfatc1* (Horsley et al., 2008), were substantially reduced (Fig. 1 F). Strikingly, markers of differentiation into IB-like and uHF states, including *Krt6a*, *Plet1*, and *Sprr1a*, were increased in this compartment (Fig. 1 F and Fig. S2 J), indicative of altered lineage identity of the OB cells. These notions were further validated by Gene Set Enrichment Analyses (GSEA [Subramanian et al., 2005]), where CerS4[epi−/−] cells showed enrichment of SG and uHF transcripts (Fig. S2 K). Collectively, these data show that while deletion of CerS4 leads to expansion of the OB compartment, these cells display altered lineage identity characterized by reduced expression of key stem cell genes and increased expression of non-lineage genes. This altered lineage identity likely explains the shifts in population ratios and drives progressive loss of the HFSC compartment, resulting in alopecia.

## CerS4 controls HFSC lineage specification

As our data pointed to altered lineage identity and population proportions of the early HFSCs, we sought to predict lineage trajectories of these cells from the scRNAseq data using Slingshot (Street et al., 2018). Ordering cells isolated from control skin along a path starting from the OB stem cells population revealed a trajectory from OB stem cells to IB stem cells and a path from OB stem cells to the uHF clusters. Cell lineage inference of CerS4[epi−/−] skin indicated an additional, abnormal trajectory from the OB HFSCs toward the sebaceous cluster (Fig. 2, A and B). Pseudotime analyses further confirmed altered linage progression, when focusing on a trajectory along uHF cells, OB, and IB stem cells in CerS4[epi−/−] mice. Here, the uHF marker Plet1 remained high in the OB stem cells, whereas expression of the OB stem cell factor Lhx2 was not turned on efficiently. Instead, the OB cells prematurely upregulated the IB marker keratin-6 (Fig. 2 C).

The data so far indicated that while the OB-like population was expanded in CerS4[epi−/−] skin, these cells displayed altered stem cell properties, characterized by decreased expression of key stem cell transcription factors and ectopic expression of IB-like and uHF genes. We thus sought to further investigate how this altered differentiation occurs. Immunofluorescence analyses of back skin sections from P21 mice confirmed decreased *Lhx2* expression in the CerS4[epi−/−] OB stem cell region of CerS4[epi−/−] mice also at this stage, most prominently visible in the upper part of the bulge (Fig. 2 D). Lhx2 loss has been reported to lead to a failure to maintain HFSC quiescence and a progressive transformation of the compartment into SGs (Folgueras et al., 2013). Interestingly, a reduction in the pearl on a string-like cell composition in the upper bulge and an increased cell size resembling a sebocyte-like cell shape was observed in CerS4[epi−/−] back skin (Fig. 2, D and E). Consistent with the RNAseq data, we observed expansion of the IB identity marker keratin-6 protein expression into OB stem cells and along the infundibulum in CerS4[epi−/−] hair follicles, whereas in control mice keratin-6 was restricted to the IB at this stage (Fig. 2 E). Importantly, lineage tracing of CerS4-deficient bulge HFSCs using Lgr5eGFP-CreERT2 (Jaks et al., 2008) revealed abnormal localization of HFSC progeny in the uHF and SG compartments (Fig. 2 F), confirming altered lineage trajectories. Collectively, these data indicate that CerS4 deficiency triggers an abnormal lineage trajectory of OB stem cells into upper and inner hair follicle as well as SG-like fates, providing a likely mechanism for the inefficient establishment of the quiescent HFSC compartment and its subsequent further gradual depletion.

## CerS4 regulates HFSC differentiation in a stem cell–autonomous manner

To gain insight into the cellular and molecular mechanisms by which CerS4 cell autonomously regulates HFSC lineage trajectories, we utilized an ex vivo organoid culture system that consists of a balanced mixture of HFSCs, hair follicle outer root sheath cells, and IB cells, but no significant levels of other epidermal cell types (Chacón-Martínez et al., 2017; Kim et al., 2020). We generated organoids from freshly isolated P19 control and CerS4[epi−/−] epidermis and confirmed that CerS4 expression was maintained in both HFSC (CD34+ integrin α6+) and non-HFSC (CD34− integrin α6+) populations in control organoids (Fig. 3, A and B). Strikingly, CerS4[epi−/−] organoids showed altered morphology characterized by smaller size and loss of cohesion of peripheral cells from the organoid clusters (Fig. 3 C), as well as strongly attenuated growth and impaired long-term maintenance within a 14-day culture period (Fig. S3 A). In agreement, CerS4[epi−/−] organoids showed reduced HFSC numbers (Fig. 3 D), whereas the non-HFSC population was only slightly but not statistically significantly reduced (Fig. S3 B). No increase in apoptosis at 4 or 7 days in culture and only a minor increase at 14 days in culture was detected in CerS4[epi−/−]

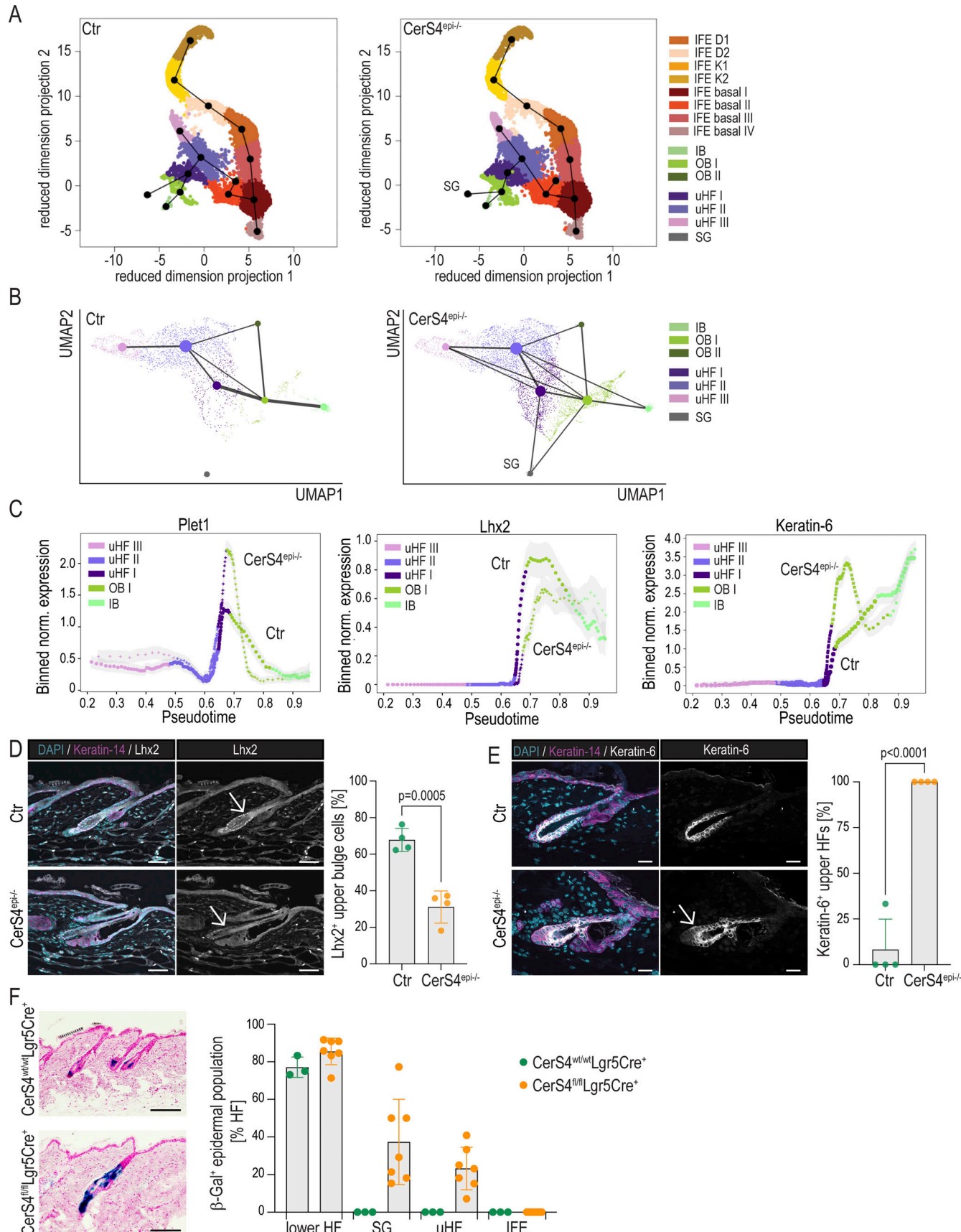

Figure 2. **CerS4 controls HFSC lineage specification. (A)** Linage trajectories as determined by the SlingShot of Ctr and CerS4^epi−/− cells from scRNAseq data (*n* = 2 mice/genotype, pooled). Note abnormal trajectory from OB cells into SG. **(B)** Lineage trajectories as determined by Paga of Ctr and CerS4^epi−/− cells from

scRNAseq data (*n* = 2 mice/genotype, pooled). Note trajectories from the uHF I cell cluster and OB cells to the SG population in CerS4^epi−/− skin. **(C)** Pseudotime analyses of Ctr and CerS4^epi−/− cells from scRNAseq data (*n* = 2 mice/genotype, pooled) confirming altered linage progression with expression of Plet1 remaining high, Lhx2 low, and keratin-6 high in CerS4^epi−/− OB stem cells. **(D)** Representative images and quantification of P21 back skin sections stained for Lhx2 (grey) and keratin-14 (magenta) show reduced expression of Lhx2 (arrows) in bulge stem cells (scale bars 50 µm; *n* = 4 mice/genotype; mean ± SD; unpaired *t* test). **(E)** Representative images and quantification of P21 control and CerS4^epi−/− back skin sections stained for keratin-6 (grey), keratin-14 (magenta) show ectopic keratin-6 expression (arrows) in the OB and uHF of CerS4^epi−/− mice (scale bars 25 µm; *n* = 4 mice/genotype; mean ± SD; unpaired *t* test). **(F)** Representative images and quantification of lineage-tracing analysis of β-galactosidase⁺ Lgr5 progeny from Ctr and CerS4^fl/fl-Lgr5eGFP-CreERT2 (Lgr5Cre⁺) mice. Quantification of hair follicles containing β-galactosidase⁺ in the SG, uHF, and IFE are shown (scale bars 80 µm; *n* = 3 Ctr and 7 = CerS4^fl/fl Lgr5Cre⁺ mice).

organoids (Fig. S3 C). In contrast, EdU incorporation experiments showed reduced proliferation of CerS4^epi−/− organoids already at day 4 (Fig. 3 E; and Fig. S3, D and E). Overall, in both control and CerS4^epi−/− organoids, the CD34⁺ integrin α6⁺ HFSCs showed higher rates of proliferation than the CD34⁻ integrin α6⁺ progenitors, indicating that HFSCs are the main drivers of organoid growth.

To understand the mechanisms of reduced HFSC numbers and to mechanistically link the organoid phenotypes to the in vivo observations of altered differentiation trajectories, we performed quantitative proteomics on d8 control and CerS4^epi−/− organoids. A total of 6,609 proteins could be quantified across all replicates, of which 399 (false discovery rate [FDR] q < 0.05, s0 = 0.01, fold change >2) were differentially expressed (Fig. 3 F and Table S1). Gene Ontology Biological Processes (GOBP) term analysis implicated metabolic processes, epidermal development, and inflammation as most strongly altered, with no major differences in proteins associated with cell death and growth (Fig. 3 G and Fig. S3 F). Importantly, proteins with increased expression in CerS4^epi−/− organoids included keratin-6 and Plet1, key markers of the differentiated IB/companion layer and the uHF, respectively, that were also increased in the OB population in vivo. Other companion layer/uHF marker proteins were also upregulated, including Klk5, Spink5, Sprr5, and filaggrin (Yang et al., 2004; Furio et al., 2015; Ziegler et al., 2019), pointing to a cell-autonomous nature of the HFSCs differentiation defect in CerS4^epi−/− HFSCs. The most downregulated proteins included Serpinb9, Isg15, Gbp2, Gbp7, and Gbp10, which relate to immune responses (Bird et al., 2014; Perng and Lenschow, 2018, Tretina et al., 2019), indicating that loss of stem cells was associated with attenuation of keratinocyte intrinsic immune regulation.

To investigate if these differential protein expression patterns would indeed reflect differences in HFSC differentiation trajectories in CerS4^epi−/− HFSCs, we systematically compared the differences in protein expression to known markers of distinct progenitor cell lineages. Indeed, GSEA revealed enrichment of uHF markers as well as the SG in the upregulated proteome of CerS4^epi−/− organoids along with negative enrichment of basal outer root sheath/stem cell compartment markers (Fig. 3 H and Fig. S3 G).

Consistent with the data obtained from proteomics, immunofluorescence analyses of control and CerS4^epi−/− organoids revealed increased levels of keratin-6⁺cells in CerS4^epi−/− organoids (Fig. 3 I). An increase in keratin-6⁺ cells was also found in the periphery of the organoid clusters surrounding the CD34⁺ HFSCs, explaining the differences seen in organoid morphology with brightfield imaging (Fig. 3 C). These loosely attached differentiating cells do not contribute to proliferation in the

organoids (Fig. S3 A), explaining the difference in cell numbers. Collectively, these data indicated reduced HFSC maintenance and increased differentiation of cells into an uHF, SG, and IB/companion layer states in the organoids, confirming that CerS4 activity in HFSCs is essential for maintenance of a stable, undifferentiated HFSC population.

## CerS4 activity is required to maintain membrane lipid homeostasis

Given the critical roles of sphingolipids in membrane composition in intracellular compartments and the plasma membrane (Mullen et al., 2011) (Fig. 4 A), we hypothesized that CerS4 is required to establish a specific membrane topology to orchestrate signaling in HFSCs. To test this, we performed quantitative lipidomics analyses of control and CerS4^epi−/− organoids to quantify sphingolipids.

Analysis of sphingolipids showed a moderate, albeit statistically not significant, decrease in ceramides with a fatty acid chain length of C18–C22 reflecting CerS4 substrate preferences in CerS4^epi−/− organoids (Fig. 4 B). A decrease in ceramide C20:0 was also seen in purified WT non-HFCSs compared with HFSCs (Fig. 4 C), linking this fatty acid to the stem cell state. More importantly, an increase in ceramide C16:0 was observed in CerS4^epi−/− organoids (Fig. 4 B). A similar increase was observed in FACS-purified control non-HFSCs compared with control HFSCs (Fig. 4 C), thus linking ceramide C16:0 to stem cell differentiation. Ceramides with a fatty acid chain length of C14:0, C24:0, C24:1, C26:0, and C26:1 were largely unaltered (Fig. 4 B and Fig. S4).

Ceramides serve as the starting point for the synthesis of sphingomyelin that plays a crucial role in maintaining the structural integrity and fluidity of cell membranes for proper signaling. Importantly, analysis of sphingomyelin levels revealed a decrease in sphingomyelins containing C18:0, C18:1, C20:0, C22:0, C22:1, C24:1, C26:0, and C26:1 fatty acids, indicative of altered membrane composition in CerS4^epi−/− organoids (Fig. 4 D). Thus, loss of CerS4 results in a decrease in specific sphingomyelin levels likely due to the reduction of its direct substrates, the C18:0, C18:1, C20:0, C22:0, and C22:1 ceramides, and compensatory dysregulation in other species. As ceramide and sphingomyelin cycling are essential for proper cell signaling the imbalance in ceramide and sphingomyelin levels likely impacts transmembrane signaling.

## Altered response to Wnt signaling drives abnormal stem cell states

To investigate potential transmembrane signaling defects by which CerS4-dependent lipid homeostasis controls HFSC

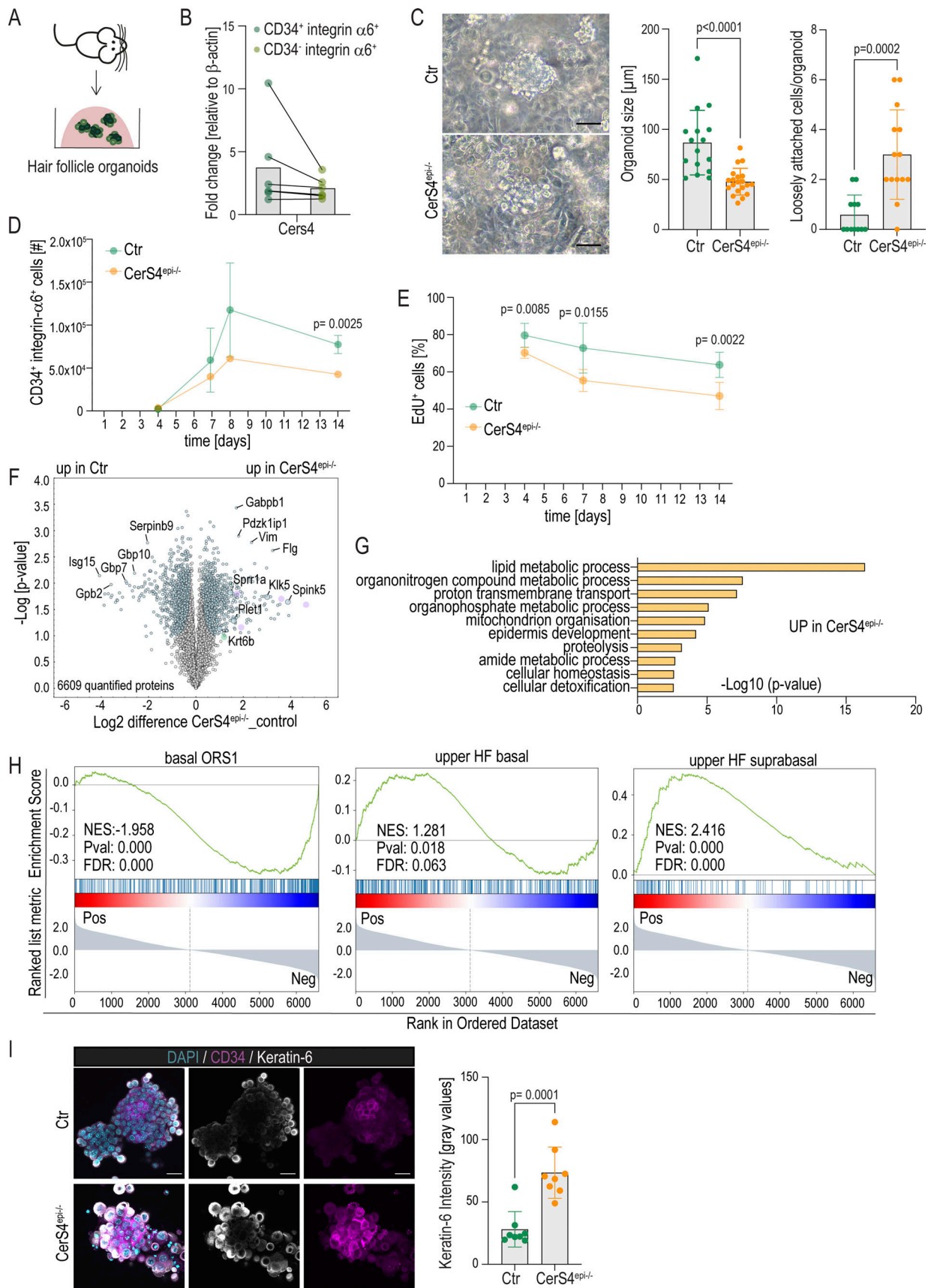

Figure 3. **CerS4 regulates HFSC differentiation in a stem cell–autonomous manner. (A)** Outline of HFCS organoid generation. **(B)** qRT-PCR analyses of Cers4 expression of FACS purified CD34⁺ α6⁺ integrin HFSCs and CD34⁻ integrin α6⁺ non-HFSCs from organoids (*n* = 5 independent experiments).

**(C)** Representative brightfield images and quantification of the size and loss of cohesion of control (Ctr) and CerS4epi−/− organoids cultured for 5 days. Scale bar 50 µm. **(D)** FACS-based quantification CD34+ integrin α6+ HFSCs from organoids cultured for time points indicated show reduced HFSCs in CerS4epi−/− organoids (n = 4 mice/genotype for 14 days; six mice/genotype all other time points; mean ± SD; unpaired t test). **(E)** Quantification of EdU-positive cells after a 24-h chase of control and CerS4epi−/− organoids cultured for indicated time points (n = 6 mice/genotype; mean ± SD; unpaired t test). **(F)** Volcano plot of differentially expressed proteins in control and CerS4epi−/− organoids (n = 4 mice/genotype). **(G)** GO-term enrichment analyses of differentially expressed proteins in CerS4epi−/− organoids. **(H)** GSEA of the differential protein expression of Ctr and CerS4epi−/− organoids indicate underrepresentation of proteins from gene expression signatures from OB and overrepresentation of proteins from uHF signatures (from Joost et al., 2016) in CerS4epi−/− organoids. **(I)** Representative images and quantification of control and CerS4epi−/− organoids stained for keratin-6 (grey) and keratin-14 (magenta) show increased keratin-6 and abundance of loosely attached, differentiating cells in CerS4epi−/− (scale bar 20 µm; n = 8 organoids/genotype; mean ± SD; unpaired t test). NES, normalized enrichment score.

maintenance and differentiation, we returned to the in vivo scRNAseq data and focused on cell–cell communication that is known to be critical for determining HFSC state and behavior (Hsu et al., 2011). Using receptor–ligand interaction analyses (CellPhoneDB [Efremova et al., 2020]), we analyzed cellular communication within the OB compartment as well as between the OB and IB. This analysis revealed an increase in Wnt5a signaling as the most prominently implicated signaling change in CerS4epi−/− OB cells (Fig. 5 A). Wnt5a activates noncanonical Wnt signaling, although in some contexts it can also induce the canonical pathway (Mikels and Nusse, 2006). Further, BMP and canonical Wnt were predicted to be increased in communication between OB and IB cells (Fig. S4 B).

Noncanonical Wnt is a multifaceted pathway characterized by two major downstream arms: the planar cell polarity/RhoGTPase/cJUN-N–terminal kinase (JNK) module and the calcium (Ca2+)/CaM-dependent kinase II (CaMKII)/PKC module (van Amerongen, 2012). Kyoto Encyclopedia of Genes and Genomes (KEGG) pathway and GO-term analyses from differentially expressed genes in the OB indeed revealed that genes involved in β-catenin–independent Wnt signaling and Wnt/planar cell polarity pathways were enriched among genes upregulated in the CerS4-deficient OB (Fig. 5 B and Table S2). Also, components of the actin cytoskeleton, key transcriptional targets of noncanonical Wnt (Qin et al., 2023) were enriched in the CerS4-deficient OB (Fig. 5 B). In contrast, transcript levels of central canonical Wnt targets, including Axin2 and CyclinD, were not substantially altered (Fig. S4 C), whereas Lef1 and Tcf4 showed only moderately increased expression in CerS4epi−/− OB cells (Fig. S4 D). Immunofluorescence analysis of back skin confirmed an increase in Lef1 protein levels in CerS4epi−/− OB stem cells in vivo (Fig. S4 E). As Lef/Tcf and Wnt–β-catenin pathways may act antagonistically or cooperatively, and Tcf4 acts as a repressor of β-catenin in the hair follicle (Tang et al., 2008; Nguyen et al., 2009), we proceeded to directly test the role of noncanonical and canonical Wnt in CerS4-deficient cells. We first manipulated noncanonical Wnt by inhibiting the CaMKII and JNK arms of the pathways using specific inhibitors KN93 (5 µM) and SP600125 (5 µM), respectively (Brown and Bayer, 2024, Castro-Torres et al., 2024). Flow cytometry–mediated quantification of HFSC (CD34+/integrin α6+) and progenitor (CD34−/integrin α6+) numbers in control organoids revealed no change in response to inhibiting JNK (Fig. 5, C and D). In contrast, inhibiting specifically the noncanonical Wnt/Ca2+–CaMKII pathway was sufficient to prevent stem cell loss in CerS4epi−/− -derived organoids (Fig. 5, C and D).

Consistently, immunofluorescence analyses revealed that inhibition of the noncanonical Wnt/Ca2+ pathway was sufficient to prevent expression of keratin-6 in the CerS4epi−/− HFSC organoids (Fig. 5 E).

To test the role of canonical Wnt, we increased Wnt activity in HFSC organoids by treating them with Chir99021 (1 µM), a small molecule inhibitor of the glycogen synthase kinase 3, an essential canonical Wnt inhibitor (Fig. S4 F). FACS quantification of HFSC (CD34+/integrin α6+) and progenitor (CD34−/integrin α6+) numbers in control organoids revealed no change in HFSCs but a clear decrease in progenitor cell numbers triggered by high Wnt (Fig. S4 G). These data indicate that these control progenitors, but not HFSCs respond to increased Wnt by increased differentiation and associated cell cycle exit. Interestingly, the number of CerS4-deficient stem cells was strongly reduced upon activation of Wnt, while the progenitor response was similar to controls (Fig. S4 G). Immunofluorescence analyses revealed a Wnt-induced increase in Sox9 expression in CD34hi control HFSCs, as expected (Fig. S4 H). Consistent with their higher sensitivity to Wnt, CerS4-deficient HFSCs responded to the increased Wnt with a stronger increase in Sox9 expression (Fig. S4 H). Further, CerS4epi−/− organoids showed a decrease in compactness and increased Krt6 expression throughout the organoid (Fig. S4 I), explaining the differences seen in HFSC (CD34+/integrin α6+) and progenitor (CD34−/integrin α6+) numbers in flow cytometry-mediated quantification (Fig. S4 I).

To understand how CerS4 could be controlling both canonical Wnt and noncanonical Wnt/Ca2+ signaling, we analyzed CerS4 localization by expressing HALO-tagged CerS4 in isolated primary CerS4epi−/− keratinocytes. As reported (Riebeling et al., 2003), CerS4 was mainly found localized in the endoplasmic reticulum that controls intracellular Ca2+ release (Fig. 5 F). We thus hypothesized that CerS4 regulates intracellular Ca2+ release. Live imaging analyses of Ca2+ oscillations revealed increased spontaneous Ca2+ oscillations compared to control cells (Fig. 5 G). These enhanced Ca2+ oscillations were restored to control levels by rescuing CerS4 expression in the CerS4epi−/− keratinocytes (Fig. 5 F and Video 1). As expected, these oscillations were upstream of CaMKII signaling, as adding KN93 to the cells did not attenuate Ca2+ flashes (Video 1).

Collectively, these data show that in CerS4epi−/− HFSCs there is an increased noncanonical Wnt/Ca2+ activity that triggers stem cell differentiation through CaMKII activity. The data further suggest that due to this differentiated state, the stem cells are sensitized to canonical Wnt, further enhancing

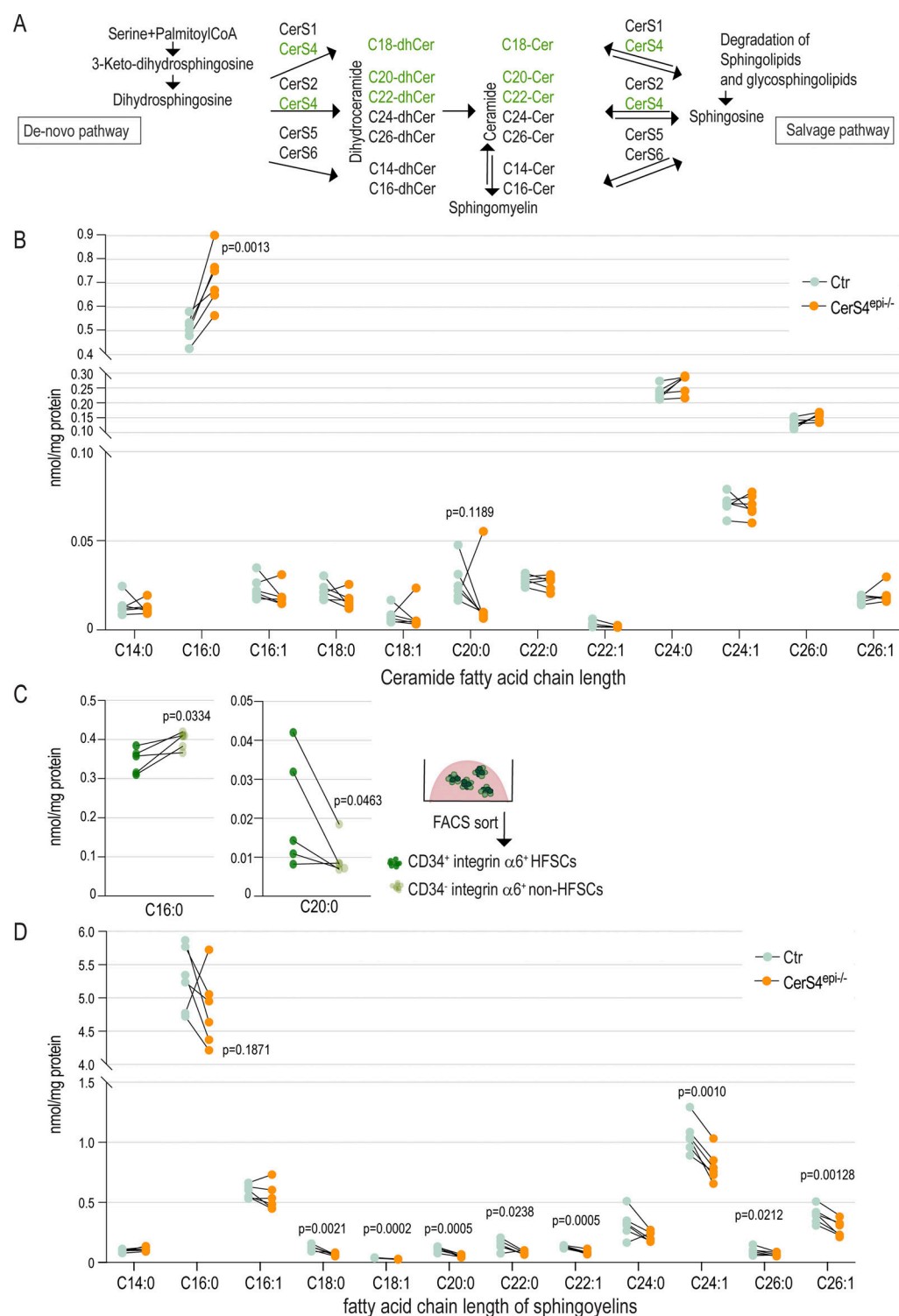

Figure 4. **CerS4 activity is required to maintain membrane lipid homeostasis. (A)** Schematic of the substrate specificity of CerS. **(B)** Ceramide levels in control (Ctr) and CerS4$^{epi-/-}$ organoids determined by quantitative LC-ESI-MS/MS analysis. Note increased C16:0 and slightly decreased C18:0 and C20:0 ceramides ($n$ = 6 mice/genotype; ratio-paired $t$ test). **(C)** Levels of C16:0 and C20:0 ceramides in FACS-purified CD34$^+$ integrin α6$^+$ HFSCs and CD34$^-$ integrin α6$^+$ non-HFSCs determined by quantitative LC-ESI-MS/MS analysis ($n$ = 5 mice/genotype; ratio-paired $t$ test). **(D)** Sphingomyelin levels in Ctr and CerS4$^{epi-/-}$ organoids determined by quantitative LC-ESI-MS/MS analysis. Note decreased C18:0, C18:1, C20:0, C22:0, C22:1, C24:1, C26:0, and C26:1 sphingomyelins ($n$ = 6 mice/genotype; ratio-paired $t$ test). LC-ESI-MS/MS, LC coupled to electrospray ionization tandem MS/MS.

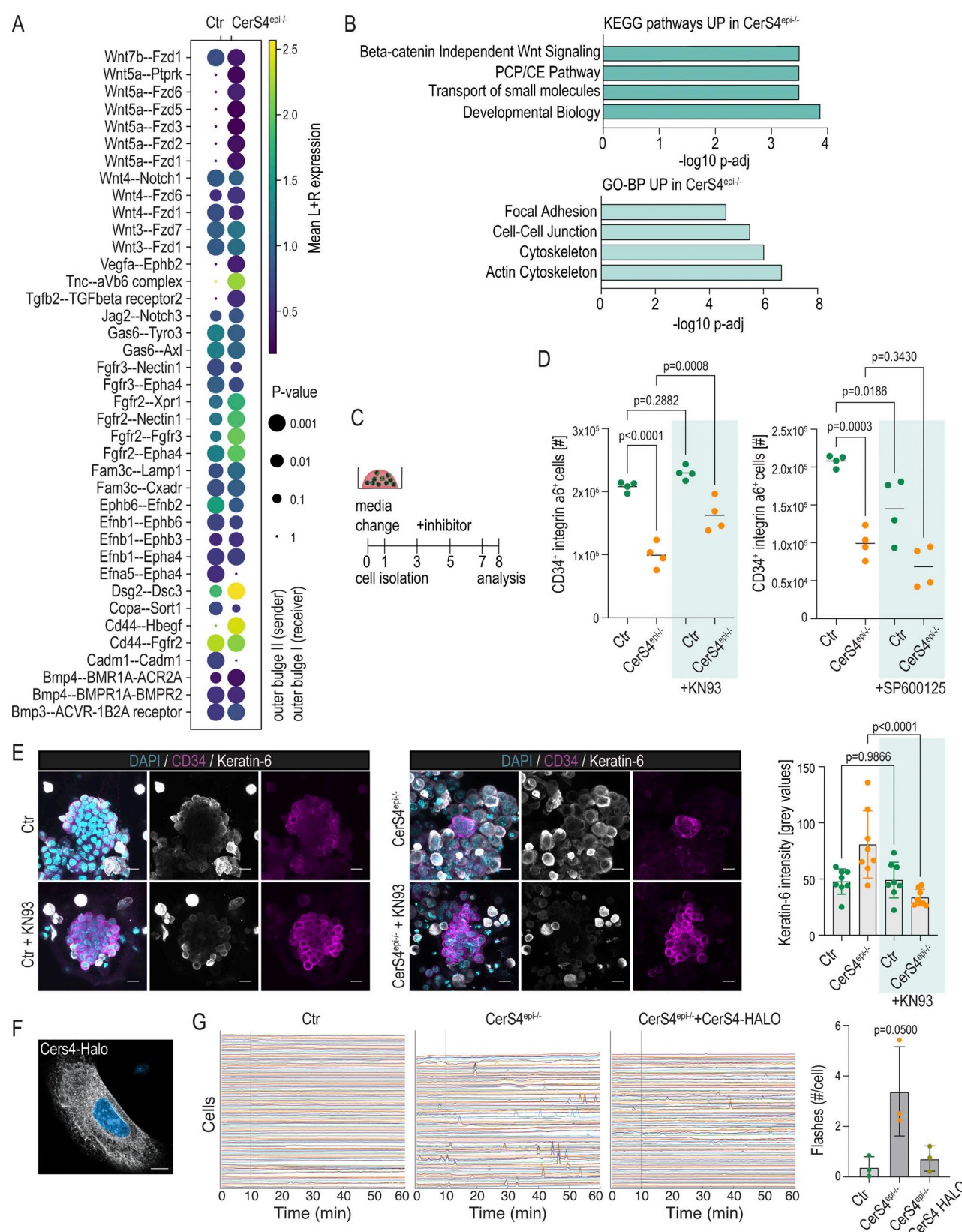

Figure 5. **Altered response to Wnt signaling drives abnormal stem cell states. (A)** Receptor–ligand interaction analyses from scRNAseq data of control (Ctr) and CerS4$^{epi-/-}$ skin. Communication within OB cells is illustrated. Note altered Wnt signaling in CerS4$^{epi-/-}$ OB. **(B)** KEGG- and GOBP-term enrichment analyses of differentially expressed genes in CerS4$^{epi-/-}$ OB stem cells. **(C)** Experimental outline for noncanonical Wnt inhibition using 5 µM KN93 or 5 µM SP600125. **(D)** FACS-based quantification of HFSCs from Ctr and CerS4$^{epi-/-}$ organoids treated with KN93 or SP600125. Note an increase in HFSCs in KN93-treated CerS4$^{epi-/-}$ organoids and a decrease in HFSCs in SP600125-treated CerS4$^{epi-/-}$ organoids ($n$ = 4 mice/genotype, mean ± SD; unpaired $t$ test).

**(E)** Representative images and quantification of Ctr and CerS4$^{epi-/-}$ organoids treated with KN93 and stained for keratin-6 (grey) and keratin-14 (magenta). Note decreased keratin-6 expression and increase of cell cohesion in KN93-treated CerS4$^{epi-/-}$ organoids (scale bar 20 µm; $n$ = 4 organoids/genotype; mean ± SD; unpaired $t$ test). **(F)** Representative images of CerS4$^{epi-/-}$ cells transfected with CerS4-HALO. Note localization in the endoplasmic reticulum. Scale bar 10 µm. **(G)** Normalized intensity tracks and quantification of single Ctr, CerS4$^{epi-/-}$, and CerS4$^{epi-/-}$ + CerS4-HALO keratinocytes imaged with Fluo-4 AM Ca$^{2+}$ probe to identify oscillations. Grey straight line marks addition of 5 µM KN93. Note that enhanced oscillations were restored to control levels by rescuing CerS4 expression ($n$ = 3 independent experiments with >30 cells/condition/experiment; mean ± SD; Kruskal–Wallis). PCP/CE = planar cell polarity/convergent extension.

differentiation, collectively explaining the observed inefficient establishment and gradual depletion of bulge HFSCs ex vivo and in CerS4-deficient mice.

### Deletion of Cers4 leads to inflammatory alterations resembling human AD

CerS4 deletion in mice showed macroscopic and histological characteristics of an AD-like eczema (Peters et al., 2020). We next tested whether this phenotype may be secondary to CerS4 deficiency-related impaired HFSC dynamics. AD is associated with immune dysregulation, particularly a Th2 dominance (Weidinger et al., 2018). Innate lymphoid cells type 2 (ILC2) represent a subset of innate lymphoid cells (Neill et al., 2010) that can promote allergic inflammation at multiple barrier organs (Vivier et al., 2018). ILC2 can be activated by signals from the epidermis to promote the differentiation of Th2 cells, eventually amplifying Th2 responses (Salimi et al., 2013). Hence, we investigated immune cell populations in CerS4$^{epi-/-}$ skin at the time point when hair follicle abnormalities were present but changes in IFE architecture and an increased transepidermal water loss (Peters et al., 2020) were only beginning to emerge.

Immunofluorescence analyses of back skin sections from P19, P21, and P58 mice showed no difference in CD45$^+$ leukocyte numbers and localization when comparing the CerS4$^{epi-/-}$ to control skin, indicating no difference in overall immune cell numbers in CerS4$^{epi-/-}$ skin (Fig. S5 A). Consistently, FACS analyses revealed no differences in CD45$^+$ leucocytes in CerS4$^{epi-/-}$ mice at P44. To be able to determine immune cell subsets based on their surface protein expression, the B cell and TC lineage was stained using antibodies against CD3$^+$, CD19$^+$, TCRβ$^+$, TCRγδ$^+$, CD5$^+$, and FCeR1$^+$ and gated as linage positive or negative. Analysis of immune cell subsets showed a slight increase in Th2 cells (viable CD45$^+$ cells, CD4$^+$, and GATA3$^+$) and an increase in ILC2 (viable CD45$^+$ cells, lineage$^-$, Nkp46$^-$, CD90.2$^+$, and GATA3$^+$) in CerS4$^{epi-/-}$ skin (Fig. 6 A). No alterations in CD4$^+$ (viable CD45$^+$ cells and CD4$^+$ cells), ILC1 (viable CD45$^+$ cells, linage$^-$, Eomes$^-$, CD11b$^-$, and Rorγt$^-$), ILC3 (viable CD45$^+$ cells, linage$^-$, Rorγt$^+$, and Nkp46$^-$), Th17 (viable CD45$^+$ cells, CD4$^+$ cells, and Rorγt$^-$), and natural killer cells (viable CD45$^+$ cells, linage$^-$, and Nkp46$^+$) were observed (Fig. 6 A). The data indicate a shift toward a Th2 dominance in CerS4$^{epi-/-}$ mice, associated with increased ILC2 cell numbers.

This AD-like inflammatory profile prompted us to investigate if CerS4 deletion would trigger gene expression alterations that share features found in human AD. To this end, we compared data from CerS4$^{epi-/-}$ skin with ortholog transcriptional profiles of 20 patients diagnosed with AD (Federico et al., 2020). GSEA of the healthy versus affected human skin areas compared with CerS4$^{epi-/-}$ skin revealed an overlapping transcriptional profile

of involved skin of patients with AD and CerS4$^{epi-/-}$ skin (Fig. 6 B and Fig. S5 B). The data indicate that CerS4$^{epi-/-}$ mice in vivo mimic transcriptional characteristics of human AD. Finally, we asked if these changes are related to altered HFSC function. Mining the Human Phenotype Ontology database with the differential protein expression dataset from control and CerS4$^{epi-/-}$ organoids revealed erythroderma, epidermal acanthosis, abnormal epidermal morphology, and inflammatory abnormality of the skin as the most prominent hits (Fig. 6 C and Table S3), where erythroderma describes exacerbation of an underlying skin disease, such as AD. Collectively, these data indicate that cell-autonomous defects in HFSC homeostasis induced by loss of CerS4 trigger a transcriptional signature, a HFSC intrinsic protein expression profile, and a Th2-dominated immune phenotype that shares features with human AD.

## Discussion

Emerging evidence suggests that lipid metabolism plays a fundamental role in stem cell homeostasis (van Gastel et al., 2020), but the mechanisms remain unclear. Our study shows that a specific ceramide profile is required for the establishment of an adult stem cell compartment, the bulge HFSCs. Deletion of the ceramide synthase CerS4 that is specifically expressed by the hair follicle cells led to dysregulation of cell fate trajectories, where cells were routed toward uHF and IB identities instead of HFSCs. Mechanistically, the hair follicle cells show altered ceramide and sphingomyelin profiles, associated with aberrant noncanonical Wnt signaling. The inability to establish the HFSC compartment leads to progressive alopecia and strikingly, a transcriptional signature, a keratinocyte intrinsic protein expression profile, and a Th2-dominated immune phenotype that shares features of human AD.

The finding that CerS4 expression was restricted to the hair follicle where it regulated a specific cell fate transition of HFSCs indicates that stem cells require establishment and dynamic maintenance of a specific CerS4-dependent lipid profile to facilitate cell type–specific signaling. Our in vivo and organoid data point to a cell-autonomous defect in Wnt/Ca$^{2+}$ signaling as the mechanism for altered stem cell differentiation. Wnt signaling is essential for establishing the hair follicle compartment and for proper regulation of HFSCs. The proper regulation of both canonical and noncanonical Wnt signaling is crucial for maintaining the balance between stem cell proliferation and differentiation. The canonical Wnt pathway is particularly important in the context of HFSC development, where Wnt signaling needs to be attenuated to allow establishment of HFSCs. Elevation of Wnt/β-catenin signaling abolishes HFSC specification and attenuates HFSC marker expression (Xu et al., 2015).

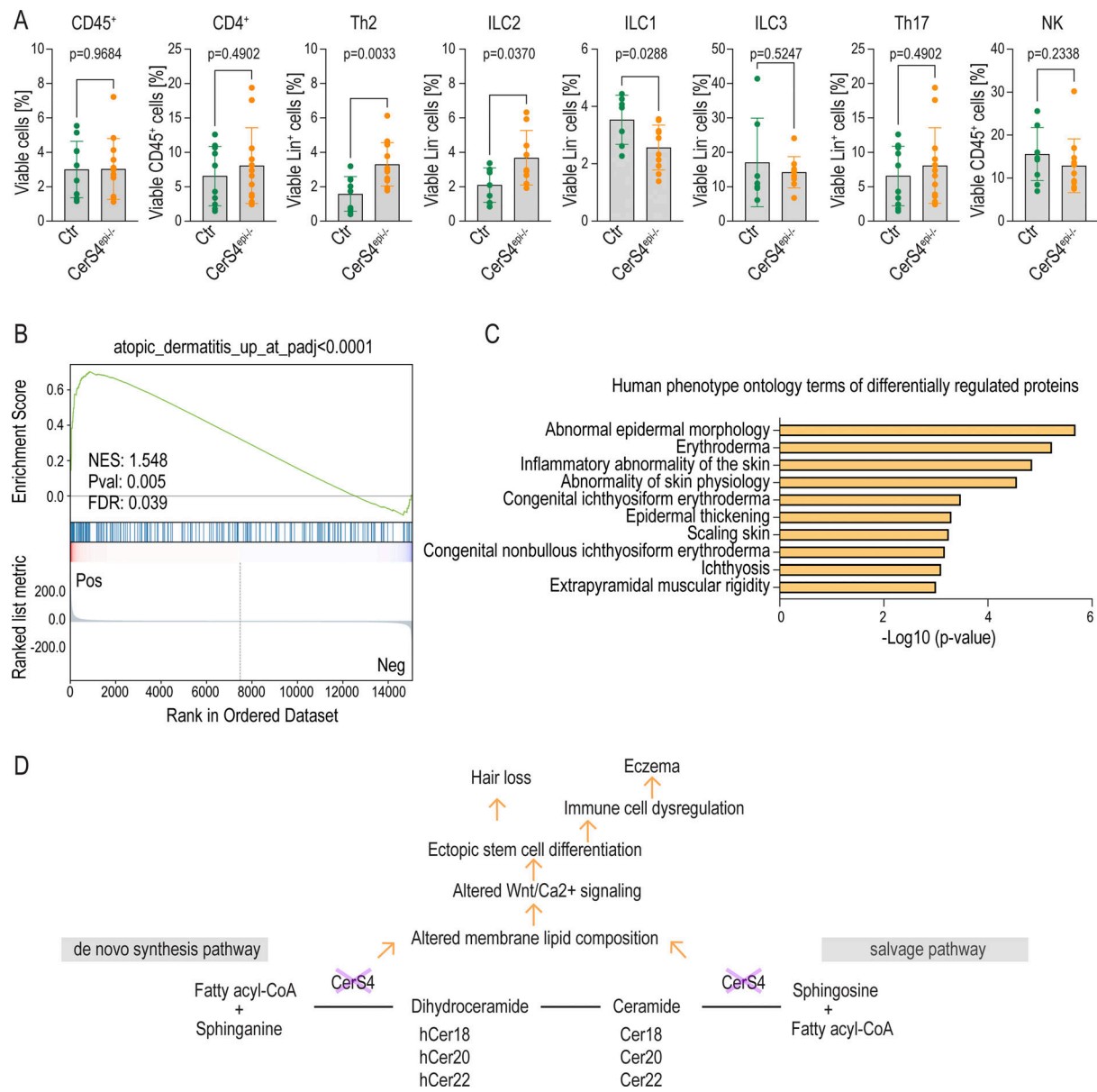

Figure 6. **Deletion of Cers4 leads to inflammatory alterations resembling human AD. (A)** FACS analysis of CD45, CD4⁺, Th2, ILC2, ILC1, ILC3, Th17, and natural killer (NK) cells in control (Ctr) and CerS4$^{epi-/-}$ ears (Ctr = 10, CerS4$^{epi-/-}$ = 12 mice/genotype, mean ± SD; unpaired $t$ test). **(B)** GSEA of transcriptional profiles from patients diagnosed with AD (Federico et al., 2020) in differentially expressed genes in Ctr and CerS4$^{epi-/-}$ skin. NES, normalized enrichment score. **(C)** Annotation enrichment analysis of in human phenotype ontology terms in differentially expressed proteins in Ctr and CerS4$^{epi-/-}$ organoids. **(D)** Model of how CerS4 deletion in HFSCs leads to hair loss and eczema.

Further, the Wnt pathway is important during homeostatic tissue maintenance in the context of hair follicle cycling, where Wnt signaling is known to play a role in the activation of HFSCs during the anagen (growth) phase, whereas it is not required to maintain HFSC identity. In contrast, noncanonical Wnt has been shown to be required for maintenance of the HFSC identity (Veltri et al., 2018). Further, noncanonical Wnt has been shown to attenuate canonical Wnt activity (van Amerongen et al., 2012). Our findings are consistent with this previous work as we observe hyperactivity of noncanonical Wnt that leads to a dual effect of aberrant HFSC differentiation but also attenuated proliferation when stimulated to expand in the organoids. In

addition, the finding that CerS4-deficient cells are sensitized to canonical Wnt inhibition compared with control cells in the organoids suggests that the hyperactive noncanonical Wnt axis could be further suppressing canonical Wnt activity in activated HFSCs. The fact that no major differences in canonical Wnt is observed in vivo could be expected based on previous observations showing that this pathway is not active in the telogen phase of the hair cycle (Niemann et al., 2002; Plikus et al., 2008; Lien et al., 2014).

Ca²⁺ is a central second messenger in the noncanonical Wnt pathway, as well as in the antagonism between the canonical and noncanonical pathways, where activation of noncanonical

Wnt elevates intracellular $Ca^{2+}$, whereas elevating intracellular $Ca^{2+}$ inhibits β-catenin–dependent signaling (Kohn and Moon, 2005). Also, ceramides have been proposed to function as second messengers through mechanisms that are unclear but are thought to involve changes in local membrane structure: changes in molecular order and domain formation, phase destabilization and membrane fusion, membrane permeability, and a transbilayer lipid movement (Venkataraman and Futerman, 2000). While the mechanistic details of how altered ceramide and sphingomyelin profiles lead to dysregulated $Ca^{2+}$ signaling are unclear, it is well known that both membrane morphology composition are critical for Wnt and $Ca^{2+}$ activity (van Meer et al., 2008; Pinot et al., 2014; Harayama and Riezman, 2018). High resolution in vitro studies of membrane dynamics are now required to understand the role of CerS4 in regulating the dynamics of intracellular $Ca^{2+}$ release and signaling.

We had previously reported that deletion of CerS4 led to an eczema-like phenotype, but the molecular and cellular mechanisms were unclear (Peters et al., 2020). We now find that CerS4 expression is restricted to the hair follicle, and the inability of establishment of the HFSC compartment is the first phenotype to be detected in the CerS4-deficient mice. Intriguingly, the mice gradually develop eczema, implicating the HFSC compartment as a key component of the epidermal barrier. Ultrastructural analyses have demonstrated that a large fraction of the skin normal flora resides in hair follicle openings, indicating that hair follicles, in addition to the IFE, are mediators of immune tolerance toward the microbiome (Chen et al., 2018). It is known that a specialized microenvironment within hair follicles shields the resident stem cells from immune system surveillance and attack, leading to the HFSC barrier being immune privileged. The immune privilege is attributed to the presence of immunosuppressive "no danger" signals, local immune-regulatory cells, and the physical seclusion provided by the follicular structure (Christoph et al., 2000; Wang et al., 2014; Agudo et al., 2018). In that respect, it is interesting to note that the altered stem cell differentiation in CerS4-deficient HFSCs was associated with reduced expression of immuno-modulatory proteins. Additional mechanistic studies are required to understand the impact of impaired HFSC compartment establishment on both the initiation and maintenance of immune tolerance, as this is crucial for unraveling drivers of Th2-driven barrier diseases such as AD, allergic reactions, and asthma.

# Materials and methods
## Mice
C57BL/6N mice with floxed CerS4 alleles were generated in cooperation with Taconic Artemis. CerS4 complete knockout mice were generated as described before (Peters et al., 2015). To achieve epidermis-specific deletion, CerS4$^{fl/fl}$ mice were crossed to K14-Cre mice (Hafner et al., 2004; Peters et al., 2015). To achieve SG-specific deletion, CerS4$^{fl/fl}$ mice were crossed to SCD3-Cre mice (Dahlhoff et al., 2016). To achieve HFSC-specific deletion CerS4$^{fl/wt}$ mice were crossed to Lgr5eGFP-CreERT2 mice (Jaks et al., 2008). To lineage trace Lgr5 progeny CerS4$^{fl/wt}$ Lgr5eGFP-CreERT2 mice were crossed to ROSA26 lacZ mice

(Friedrich and Soriano, 1991). Experiments were performed with CerS4$^{fl/fl}$K14Cre$^{+}$ (termed: CerS4$^{epi-/-}$), CerS4$^{fl/fl}$SCD3-Cre$^{+}$(termed: CerS4$^{SG-/-}$), CerS4$^{fl/fl}$K14Cre$^{-}$, and CerS4$^{fl/fl}$SCD3-Cre$^{-}$ (termed: control), as well as with CerS4$^{fl/fl}$Lgr5Cre$^{+}$ROSA26 lacZ$^{fl/fl}$ and CerS4$^{wt/wt}$Lgr5Cre$^{+}$ROSA26 lacZ$^{fl/fl}$ male and female littermates at the indicated P.

## Immunofluorescence and immunohistochemistry
Tissue biopsies were fixed with 4% PFA in PBS, embedded in paraffin, and sectioned. For hematoxylin-eosin (H/E) staining, sections were deparaffinized using a graded alcohol series, and staining was performed using standard protocols with Mayer's hematoxylin and eosin solutions (both from Sigma-Aldrich). The sections were mounted with xylene-based mounting medium (Entellan; Sigma-Aldrich) and imaged using an Olympus CKX53 brightfield microscope equipped with an Olympus EP50 camera and a 20× air objective. Image acquisition was controlled by Olympus cellSens softaware (v4.2); images were obtained at room temperature (RT).

For immunofluorescence, sections were deparaffinized using a graded alcohol series, blocked in 10% normal goat serum, and incubated with primary antibodies diluted in Dako Antibody Diluent over night at 4°C. Bound primary antibody was detected by incubation with Alexa Fluor 488– or Alexa Fluor 568–conjugated antibodies (Invitrogen). Nuclei were counterstained with DAPI (Invitrogen). After washing, slides were mounted in Elvanol. The following antibodies were used: keratin-14 (#GP-CK14; Progen), keratin-6 (#prb-169P; Covance), Lhx2 (#ab184337; Abcam), Lef1 (#2230S; Cell Signaling), keratin-15 (#NBP3–23918; Novus Biologicals), anti-BrdU (#B2531; Merck), and CD45 (#12–0451; eBioscienc2).

Organoid cultures were rinsed once in PBS, followed by fixation with 2% PFA in PBS for 30 min at RT. Fixed cells were rinsed three times with 100 mM glycine and PBS, then permeabilized and blocked for 2 h at 37°C in 0.3% Triton X-100 and 5% BSA in PBS. Cells were stained with primary antibodies in 0.3% Triton X-100 and 1% BSA in PBS overnight at RT. Secondary, fluorescent antibodies were used to detect primary antibody binding, and nuclei were visualized with DAPI. Slides were mounted with Elvanol mounting medium. The following antibodies were used: SOX9 (sc-166505; Santa Cruz Biotechnology), keratin-6 (prb-169P; Convance), and CD34 (14-0341; eBioscience). Secondary antibodies anti-mouse Alexa Fluor 488 (Invitrogen) and anti-rabbit Alexa Fluor 568 (Invitrogen).

All fluorescence images were collected by laser scanning confocal microscopy (LSM980; Zeiss) controlled by ZEN software (version 3.7), using 40× dry or 63× immersion objectives. Images were acquired at RT using sequential scanning of frames, after which planes were projected as a maximum intensity confocal stack. Images were collected with the same settings for all samples within an experiment. Image processing (linear brightness and contrast enhancement) was performed with Fiji Software version 2.9.0 and Adobe Photoshop CS5.

## scRNAseq
Back skin of P19 female mice (two mice/genotype) were dissected, and single-cell suspensions were generated by digesting

cells in a mixture of DNAse I (40 µg/ml), Liberase (0.125 mg/ml; Roche Cat. NO. 05401119001), and papain (30 U/ml; Worthington Cat. NO. LK003150) in minimum essential medium eagle, spinner modification (S-MEM) (Gibco) for 60 min at 37°C. After washing with ice-cold S-MEM, single cells were counted using Luna-II automated cell counter (Logos Biosystems), labeled with Mouse Immune Single-Cell Multiplexing Kit (BD), and loaded on a microwell cartridge of the BD Rhapsody Express system (BD) following the manufacturer's instructions. Single-cell whole transcriptome analysis libraries were prepared according to the manufacturer's instructions using BD Rhapsody WTA Reagent kit (633802; BD) and sequenced on the Illumina NextSeq 500 using High Output Kit v2.5 (150 cycles; Illumina) for 2 × 75-bp paired-end reads with 8-bp single index aiming sequencing depth of >20,000 reads per cell for each sample.

Raw FASTQ reads were quality trimmed using fastp (version 0.23.2 length cutoff 20, quality cutoff 15). Adapter trimming was disabled due to the presence of BD Rhapsody multiplexing sequences. The unique molecular identifier (UMI), complex barcode, and sample tags were extracted and demultiplexed using custom scripts. Reads were mapped to the mouse reference genome version GRCm39, with Gencode annotations vM29, filtered for protein-coding genes and long non-coding RNAs (lncRNAs), using STAR version 2.7.10a (Dobin et al., 2013) (--soloType Droplet–soloCellFilter None–soloFeatures GeneFull_Ex50pAS–soloCBstart 1–soloCBlen 27–soloUMIstart 28–soloUMIlen 8–soloCBwhitelist rhapsody_whitelist.txt–soloMultiMappers EM).

Raw counts were imported as AnnData (Virshup et al., 2021, Preprint) objects. We removed low complexity barcodes with the knee plot method (<1,200 counts per cell) and further filtered out cells with a high mitochondrial mRNA percentage (>20%). Doublets were predicted with scrublet (Wolock et al., 2019). Finally, each sample's gene expression matrix was normalized using scran (1.22.1 [Lun et al., 2016]) with Leiden clustering (Traag et al., 2019) input at resolution 0.5. At this stage samples were merged with scanpy (1.9.1 [Wolf et al., 2018]). For 2D embedding, the expression matrix was subset to the 3,000 most highly variable genes (sc.pp.highly_variable_genes, flavor "seurat"). The top 100 principal components were calculated and batch-corrected using Harmony (0.0.5 [Korsunsky et al., 2019]). The principal components served as basis for k-nearest neighbor calculation (sc.pp.neighbors, $n$_neighbors = 30), which were used as input for Uniform Manifold Approximation and Projection (UMAP) (McInnes et al., 2018, Preprint) layout (sc.tl.umap, min_dist = 0.3). Leiden clustering at resolution 0.8 was used to annotate cell clusters based on previously published marker genes (Joost et al., 2016).

## Pseudobulk differential expression analysis (DEA)
To compare gene expression between control and CerS4$^{epi-/-}$ conditions, we performed "pseudobulk" DEA (Squair et al., 2021) on the scRNAseq dataset. First, cells were randomly divided into two pseudoreplicates, and raw expression counts of cells belonging to the same pseudoreplicate were aggregated for each gene. Pseudobulk data were imported into R, and DEA was run using the Wald test for generalized linear model (GLM) coefficients implemented in the DESeq2 package (1.34.7 [Love

et al., 2014]). P-values were adjusted for multiple testing using independent hypothesis weighting (1.22.0 [Ignatiadis et al., 2016]).

## Trajectory and pseudotime analysis
Lineage trees were calculated independently for control and CerS4$^{epi-/-}$ using SlingShot (2.2.1 [Street et al., 2018]), with the starting cluster chosen as IB. Clusters Langerhans cells and TC were excluded from the analysis. Diffusion pseudotime (Haghverdi et al., 2016; Wolf et al., 2019) was calculated using the scanpy (1.9.1 [Wolf et al., 2018]) functions "sc.tl.diffmap" and "sc.tl.dpt" on an AnnData subset of the clusters: IB, OB1, OB2, uHF1, uHF2, and uHF3. The starting cell was chosen from the IB cluster. Pseudotime expression profiles were generated by binning cells according to diffusion pseudotime into 200 bins, and plotting the average normalized expression × alongside a 95% confidence interval (±1.96 * sd[x]/sqrt[x]). Cluster labels and colors are assigned based on a "majority vote" of cells in each bin.

## Ligand–receptor interactions
CellphoneDB (2.1.4 [Efremova et al., 2020]) was used to calculate ligand–receptor interactions independently on the control and CerS4$^{epi-/-}$ conditions ("cellphonedb method statistical_analysis --counts-data hgnc_symbol"). Gene symbols were converted to their human orthologs using Ensembl BioMart (Martin et al., 2023).

## Gene expression comparisons and gene set enrichment
We obtained raw RNA counts from (Suárez-Fariñas et al., 2015) and ran DEA using the Wald test for generalized linear models (GLM) coefficients implemented in the DESeq2 package (1.34.7 [Love et al., 2014]) to compare lesional versus non-lesional samples. P-values were adjusted for multiple testing using independent hypothesis weighting (1.22.0 [Ignatiadis et al., 2016]).

For gene set enrichment, we ranked differential expression results by –log10 (P-value) prefixed with the sign of the log-fold change and used GSEApy (0.10.8 [Fang et al., 2023]) "prerank" to calculate the enrichment statistics on various gene sets. Proteomics differential expression results were tested for enrichment against the marker gene sets from Table S2 in Joost et al. (2016). To compare AD genes to the gene expression changes in CerS4$^{epi-/-}$ versus control, we selected differentially expressed genes from the lesional versus non-lesional comparison in the AD data (see above) using an adjusted P-value cutoff of 0.0001. Mouse gene symbols were then converted to human gene symbols using Ensembl BioMart (Martin et al., 2023) orthologs.

## HFSC organoid culture
HFSC organoids were cultured essentially as described previously (Chacón-Martínez et al., 2017). Briefly, epidermal progenitors were isolated from back skin of P19 mice by incubating skin pieces in 0.5% trypsin (Gibco) for 50 min at 37°C. After separating the epidermis from the underlying dermis, cells were passed through 70-µm cell strainer (BD Biosciences) and pelleted at 900 rpm for 3 min. For 3D culture, 8 × 10$^4$ cells were suspended in 20 µl ice-cold 1:1 mixture of keratinocyte growth

medium (KGM; S-MEM [Sigma-Aldrich], 5 µg/ml insulin [Sigma-Aldrich], 10 ng/ml EGF [Sigma-Aldrich], 10 µg/ml transferrin [Sigma-Aldrich], 10 µM phosphoethanolamine [Sigma-Aldrich], 10 µM ethanolamine [Sigma-Aldrich], 0.36 µg/ml hydrocortisone [Calbiochem], 2 mM glutamine [Gibco], 100 U/ml penicillin and 100 µg/ml streptomycin [Gibco], 10% chelated fetal calf serum [Gibco], 5 µM Y27632, 20 ng/ml mouse recombinant VEGF, and 20 ng/ml human recombinant FGF-2 [all from Miltenyi Biotec]) and growth factor-reduced Matrigel (Corning) that was dispensed as a droplet in 24-well cell culture dishes. The suspension was allowed to solidify for 30 min after which it was overlaid with 500 µl medium. All cultures were incubated at 37°C, 5% $CO_2$. Medium was exchanged the next day after initial seeding and thereafter every second day. Chir99021 (# SML1046; Sigma-Aldrich), KN93 (# K1385; Sigma-Aldrich), and SP600125 (#S5567; Sigma-Aldrich) were added where indicated.

## Flow cytometry
Single-cell suspensions were prepared from murine back skin as described above or from organoid cultures by mechanical homogenization and incubation in 0.5% trypsin (Gibco) and 0.5 mM EDTA for 10 min at 37°C. Cells were rinsed once with KGM and stained with fluorescently labeled antibodies for 30 min on ice. After two washes with FACS Buffer (2% fetal calf serum, 2 mM EDTA in PBS), cells were analyzed in a BD FACSCanto II or sorted in either a BD FACSAria II or a BD FACSAria Fusion. Data were analyzed using FlowJo software version 10.9. Expression of cell surface markers was analyzed on live cells after exclusion of cell doublets and dead cells using 7AAD (eBioscience), DAPI (Sigma-Aldrich), or fixable viability dye 405 (eBiosciences). Apoptotic cells were labeled with Alexa Fluor 555–Annexin V (Invitrogen). The following antibodies were used: eFluor660-or FITC-CD34 (clone RAM34; eBioscience), and PE-Cy7-or FITC-α6 integrin (clone GoH3; eBioscience).

Immune cell FACS was performed from ear skin from which cells were isolated using DNAse I 40 µg/ml (11284932001; Roche) and 20 U/ml Collagenase Type I Worthington (LS0004194) for 90 min at 37°C in RPMI1640, followed by gentleMACS tissue dissociation using gentleMACS C tubes and subsequent separation using 70-µm cell strainers. After centrifugation at 300 g, 4°C, for 10 min, cells were resuspended in 500 µl PBS containing 2% FBS.

The following antibodies were used: Fc Block (14-0161-86; Invitrogen), live/dead eF780 (65-0865-14; Invitrogen), CD45 FITC (103108; BioLegend), Gata3 PE (12-9966-42; eBioscience), NKp46 eF710 (46-3351-82; eBioscience), CD4 PE-Cy7 (100421; BioLegend), Eomes eF660 (50-4875-82; eBioscience), CD90.2 AF700 (105320; BioLegend), Rorγt BV421 (562894; BD Biosciences), CD25 BV 605 (563061; BD Biosciences), CD11b super bright 702 (67-0112-82; eBioscience), CD3 biotin (344820; BioLegend), CD19 biotin (115504; BioLegend), TCRb biotin (109204; BioLegend), TCRgd biotin (13-5711-82; eBioscience), CD5 biotin (BioLegend [100604], FCeR1 biotin [134304; BioLegend], and SA V510 anti-biotin [405234; BioLegend]).

## EdU incorporation
Organoids were grown in the presence or absence of 9.4 µm EdU (Thermo Fisher Scientific) for 24 h before analysis. After preparing single-cell suspensions, cells were stained with a fixable viability dye eFluor405 followed by antibody staining before fixation in 2% PFA for 10 min at RT. Cells were subsequently permeabilized in 0.025% Triton X-100, PBS for 10 min, and incubated 30 min in EdU reaction cocktail (100 mM Tris, pH 8.5, 1 mM $CuSO_4$, 0.5 µM Alexa Fluor 594-Azide [Thermo Fisher Scientific], and 100 mM ascorbic acid). After two washes with PBS, cells were analyzed by flow cytometry as described above.

## BrdU incorporation
For short-term BrdU-pulse experiments, mice were injected i.p. with 25 µg/mg bodyweight BrdU (#B5002; Merck) 30 min before scarification. Tissue biopsies were fixed (4% PFA), embedded in paraffin, sectioned, and stained as described above. Images were obtained as described above.

## RNAscope
Paraffin-embedded skin sections were prepared as described above. Sections were labeled using the RNAscope Multiplex Fluorescent Detection Kit v2 (# 323100; ACDBio). A20zzz probe of murine CerS4 mRNA was designed by the ACD Probe Design Team. The Opal 520 Reagent Pack (FP1487001KT; Akoya Biosciences) was used in a dilution of 1:1,000 for the fluorophore step to develop the channel associated with the CerS4 probe. Nuclei were counterstained with DAPI (Sigma-Aldrich). Slides were mounted with Elvanol mounting medium and imaged as described above.

## qRT-PCR
RNA was isolated using RNeasy Plus Mini Kit (QIAGEN). 500 ng of RNA was subjected to RT using SuperScript IV VILO Master Mix (Thermo Fisher Scientific) following the manufacturer's protocol. PCR was performed with Scientific QuantStudio 5 and 7 Real-Time PCR System (Thermo Fisher Scientific) using the DyNAmo Color Flash SYBR Green Mix (Thermo Fisher Scientific). Gene expression was quantified using the ΔΔCt method using normalization to GAPDH. Primer sequences were as previously reported (Peters et al., 2015).

## Lineage tracing using Lgr5eGFP-CreERT2
To achieve inducible deletion of CerS4 and simultaneous expression of β-galactosidase in Lgr5+ stem cells, CerS4[fl/fl] mice were crossed with Lgr5-EGFP-Ires-CreERT2 (Jaks et al., 2008) mice and Rosa26R-Lacz Cre reporter mice (Friedrich and Soriano 1991). Cre recombinase was activated by injecting 10 mg tamoxifen (T5648; Sigma-Aldrich) dissolved in 50 µl sunflower oil (S5007; Sigma-Aldrich) i.p. for seven consecutive days starting at P21. Mice were analyzed at P85.

Back skins were fixed in 0.2% glutaraldehyde and 2% PFA in PBS at 4°C for 30 min, and then washed with 0.1% NP-40, 2 mM $MgCl_2$ in PBS once. The fixed skins were stained with 1 mg/ml X-Gal, 5 mM K3Fe(CN), 5 mM K4Fe(CN) in 0.1% NP-40, 2 mM $MgCl_2$ in PBS for 24 h, then washed three times for 10 min in 0.1%NP-40, 2 mM $MgCl_2$ in PBS, and then embedded in paraffin. Paraffin sections were deparaffinized and rehydrated in xylene, isopropanol, 95% ethanol, 75% ethanol, and 50% ethanol each for 5 min. Sections were stained with nuclear fast red for 10 min and

rinse briefly in double distilled water (ddH$_2$O). Sections were dehydrated in 50% ethanol, 75% ethanol, 95% ethanol, and isopropanol each for 10 s, and cleared in two changes of xylene for 10 s each. The sections were mounted with xylene-based mounting medium (Entellan; Sigma-Aldrich) and imaged using an Olympus CKX53 brightfield microscope equipped with an Olympus EP50 camera and a 20× air objective. Image acquisition was controlled by Olympus cellSens softaware (v4.2), images were obtained at RT.

### Proteomics
Organoid pellets from four mice/genotype were lysed and digested using the Preomics iST kit (P.O.00001) according to the manufacturer's instructions, except that the volume of the LYSE buffer was increased to 70 µl to completely solubilize the organoids. Following peptide elution and lyophilization, peptides were then subjected to tandem mass tag (TMT) labeling in 3-(4-(2-hydroxyethyl)piperazin-1-yl)propane-1-sulfonic acid (EPPS) buffer (20 µl, 200 mM) using TMT labels 127N–130C (42 µg peptide/replicate and channel; 200 µg labeling reagent). After mixing of the eight channels in an equal ratio, the mixture was desalted on Empore 3M cartridges, followed by phosphopeptide enrichment on Fe-NTA spin columns (# A32992; Thermo Fisher Scientific; data not shown). Unbound peptides were lyophilized and offline fractionated by high pH-reversed phase HPLC (YMC Triart C18 column, 250 × 4.6 mm ID) into 60 fractions for the subsequent measurements of the total proteomes. All fractions were further desalted on C18 stage tips prior to mass spectrometry (MS) analysis.

Nanoflow LC-MS/MS analysis was carried out as previously published (Bekker-Jensen et al., 2017). Briefly, peptide samples were reversed-phase separated on a fused silica capillary column (length 25 cm; ID 75 µm; Dr. Maisch ReproSil-Pur C18-AQ, 1.9 µm) using an Easy nLC 1200 nanoflow system that was online coupled via a Nanospray Flex ion source to an Orbitrap HF mass spectrometer (Thermo Fisher Scientific). Bound peptides of each fraction were eluted using short gradients from 7–45% B (80% ACN, 0.1% formic acid) in 30 min, followed by a washout at 90% B, before returning again to starting conditions (total runtime 38 min). The mass spectrometer was operated in the positive ion mode, switching in a data-dependent fashion between survey scans in the orbitrap (mass range m/z = 350–1,400; resolution R = 60,000; target value = 3E6; and max injection time [IT] 100 ms) and MS/MS acquisition (High-energy collisional dissociation) of the 20 most intense ion peaks detected (resolution R = 15,000; target value = 1E5; max IT = 15ms; and normal collision energy [NCE] = 27). Dynamic exclusion was enabled and set to 30 s.

Raw MS data were processed using MaxQuant (v. 2.1.4.0) with the built-in Andromeda search engine. Tandem mass spectra were searched against the mouse UniProtKB database (UP000000589_10090.fasta; version from 01/2022) concatenated with reversed sequence versions of all entries and also containing common contaminants. Carbamidomethylation on cysteine residues was set as fixed modification for the search, while oxidation at methionine, acetylation of protein N termini and deamidation on asparagine and glutamine were set as variable modifications. Trypsin was defined as the digesting enzyme, allowing a maximum of two missed cleavages and requiring a minimum length of six amino acids. The maximum allowed mass deviation was 20 ppm for MS and 0.5 Da for MS/MS scans. The match between run function was enabled. Protein groups were regarded as being unequivocally identified with an FDR of 1% for both the peptide and protein identifications.

Data transformation and evaluation were performed using Perseus software (version 1.6.15.0). Proteins identified by a single-modified site only, common lab contaminants, as well as proteins containing reverse sequences derived from the decoy database search, were removed from the dataset prior to any further analysis. For quantification, TMT reporter intensity values were first log$_2$ transformed, demanding valid values in all of the replicates in both experimental groups. This step reduced the number of quantifiable protein groups from 6,803 to 6,609; however, at the same, time imputation of missing values was avoided. Reporter intensities were quantile normalized and significant differences between controls and knockout samples were determined using a Student's $t$ test with a permutation-based FDR of 0.05 set as cutoff. Only proteins, which showed an additional greater than twofold difference were considered for further evaluation.

### Lipidomics
Levels of ceramides and sphingomyelins were determined by LC coupled to electrospray ionization tandem MS/MS. Organoids and FACS-purified cells were homogenized in Milli-Q water using the Precellys 24 homogenizer (Peqlab) at 6,500 rpm for 30 s. The protein content of the homogenate was routinely determined using bicinchoninic acid. Lipid extraction, alkaline hydrolysis of glycerolipids and LC coupled to electrospray ionization tandem MS/MS analysis of ceramides and sphingomyelins were performed as previously described (Hammerschmidt et al., 2023).

### CerS4-HALO-tag and Ca²⁺ imaging
The open reading frame of CerS4 (NM_026058.4) was cloned into the pRP-Halo vector using Vector Builder. Epidermal progenitors from CerS4$^{epi-/-}$ and control mice were grown as organoids to enrich for HFSCs, after which they were plated on thick Matrigel hydrogels for transfection and live imaging. Cells were transfected using Lipofectamine 3000 according to the manufacturers' instructions. Next day, 6 µM Fluo-4 AM (acetoxymethyl esther) Ca$^{2+}$ dye (R&D Systems) and 60 nM JF646 Janelia Fluor HaloTag ligand (Promega) were added, after which cells were imaged for 60 min using an Andor Dragonfly 505 high-speed spinning disk confocal microscope (Oxford Instruments) equipped with 488- and 637-nm lasers, an Andor Zyla 4.2 sCMOS camera, and an environmental chamber set at 37°C and 5% CO$_2$. Acquisition was carried out with 60× water immersion objectives using the Fusion 2.0 software. For analysis, cells were segmented using Cellpose (Stringer et al., 2021), after which cytoplasmic Fluo-4 AM intensity was measured.

### Statistics and reproducibility
Statistical analyses were performed using GraphPad Prism software (version 10; GraphPad). Statistical significance was determined by the specific tests indicated in the corresponding figure legends. Only two-tailed tests were used. In all cases

where a test for normally distributed data was used, normal distribution was confirmed with the Kolmogorov–Smirnov test ($\alpha = 0.05$). All experiments presented in the manuscript were repeated at least in three independent biological replicates.

### Online supplemental material

Fig. S1 shows the additional analyses of skin phenotype in epidermis-specific CerS4 knockout mice. Fig. S2 shows the scRNAseq analyses of cellular phenotype in epidermis-specific CerS4 knockout mice and skin phenotype in SG-specific CerS4 knockout mice. Fig. S3 shows the analyses of HFSC organoid cultures from CerS4-deficient mice. Fig. S4 shows the altered differentiation trajectories and Wnt signaling in CerS4-deficient HFSCs. Fig. S5 shows immunostainings of CD45-positive cells in CerS4-deficient skin and differential gene expression analyses in eczema patients. Video 1 shows the live imaging of $Ca^{2+}$ oscillations in control, CerS4-deficient, and CerS4-deficient keratinocytes rescued with CerS4-HALO cDNA. Table S1 contains the differentially abundant proteins in control and CerS4$^{epi-/-}$ HFSC organoids. Table S2 contains the differentially expressed genes in the OB populations of control and CerS4$^{epi-/-}$ mice from scRNAseq data. Table S3 contains the selected significantly upregulated proteins in CerS4$^{epi-/-}$ HFSC organoids that are found in the Human Phenotype Ontology database.

### Data availability

All data supporting the findings of this study are available from the corresponding author on request. Single-cell sequencing data are available at GEO (GSE252821). Proteomics data have been deposited to the ProteomeXchange Consortium through the PRIDE partner repository (Perez-Riverol et al., 2022), dataset ID: PXD047573.

## Acknowledgments

We thank Dr. Marlon R. Schneider (University of Lepizig, Leipzig, Germany) for the SCD3-Cre mice. We thank Anu Luoto, Sandra Heising, and Manuela Haustein for their technical assistance; the Max Planck Institute Sequencing Core Facility for support with sequencing; the BioOptics and Biomedicum Helsinki Imaging Unit for imaging support; and the HiLIFE Laboratory Animal Centre Core Facility, the University of Helsinki, and the Max Planck Institute Animal facility for their support with animal experiments.

This work was supported by the Max Planck Society, the Sigrid Juselius Foundation, the Helsinki Institute of Life Science, the European Research Council under the European Union's Horizon 2020 research and innovation programme (grant agreement 770877 - STEMpop), and the Academy of Finland Center of Excellence BarrierForce (all to S.A. Wickström). F. Peters is supported by the Walter Benjamin Fellowship from the German Research Foundation (Project number 455963994).

Author contributions: F. Peters: conceptualization, data curation, formal analysis, funding acquisition, investigation, methodology, software, supervision, validation, visualization, and writing—original draft, review, and editing. W. Höfs: investigation and writing—review and editing. H. Lee: data curation, formal analysis, investigation, software, validation, visualization, and writing—review and editing. S. Brodesser: formal analysis and investigation. K. Kruse: data curation, formal analysis, software, validation, and visualization. H.C.A. Drexler: formal analysis and investigation. J. Hu: investigation. V.K. Raker: formal analysis, investigation, methodology, project administration, visualization, and writing—review and editing. D. Lukas: data curation, formal analysis, investigation, methodology, validation, visualization, and writing—review and editing. E. von Stebut: data curation, investigation, resources, supervision, and validation. M. Krönke: conceptualization and resources. C.M. Niessen: conceptualization, funding acquisition, methodology, resources, supervision, and writing—original draft, review, and editing. S.A. Wickström: conceptualization, formal analysis, funding acquisition, investigation, methodology, project administration, resources, supervision, visualization, and writing—original draft, review, and editing.

Disclosures: S.A. Wickström reported a patent to WO2017060240A1 issued. No other disclosures were reported.

Submitted: 14 March 2024

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

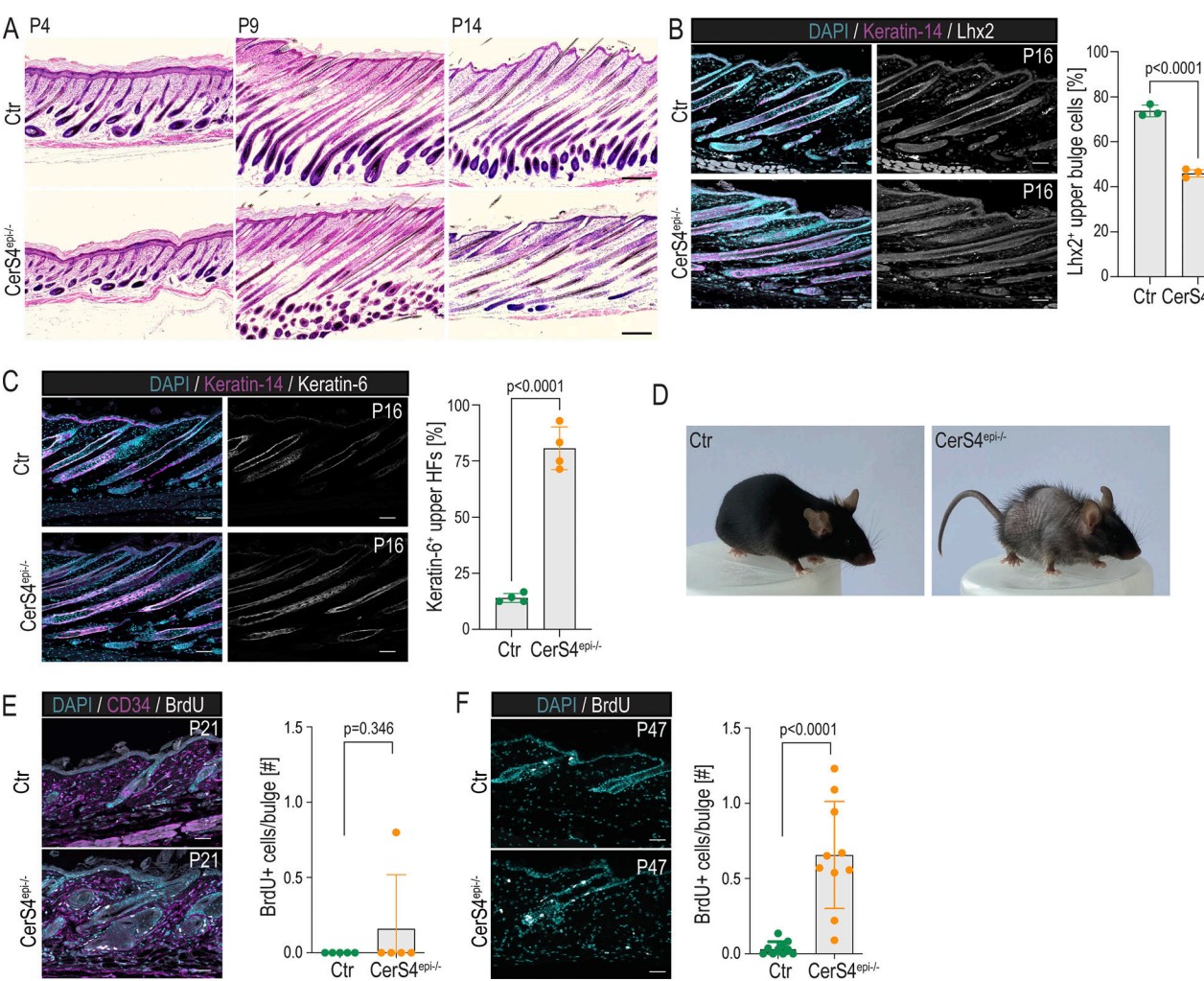

Figure S1. **Analyses of skin phenotype in epidermis-specific CerS4 knockout mice. (A)** Representative images and quantification of hair follicle stage of H/E-stained back skin sections of control (Ctr) and CerS4[epi−/−] back skin at P4, 9, and 14 (scale bars 100 μm; n = 4–6 mice/genotype). **(B)** Representative images and quantification of P16 back skin sections stained for Lhx2 (grey) and keratin-14 (magenta) show reduced expression of Lhx2 in bulge stem cells (scale bars 50 μm; n = 3–4 mice/genotype; mean ± SD; unpaired t test). **(C)** Representative images and quantification of P16 Ctr and CerS4[epi−/−] back skin sections stained for keratin-6 (grey), keratin-14 (magenta) show ectopic keratin-6 expression in the uHF of CerS4[epi−/−] mice (scale bars 25 μm; n = 4 mice/genotype; mean ± SD; unpaired t test). **(D)** Representative photographs of adult Ctr and CerS4[epi−/−] littermates show hair loss in CerS4[epi−/−] mice. **(E)** Quantification of P21 Ctr and CerS4[epi−/−] back skin sections with BrdU labeling and CD34 staining to mark the bulge stem cell compartment (scale bars 25 μm; n = 5 mice/genotype; mean ± SD; unpaired t test). **(F)** Quantification of P47 Ctr and CerS4[epi−/−] back skin sections labeled with BrdU show increased BrdU incorporation in CerS4[epi−/−] stem cells (scale bars 25 μm; n = 10 mice/genotype; mean ± SD; unpaired t test).

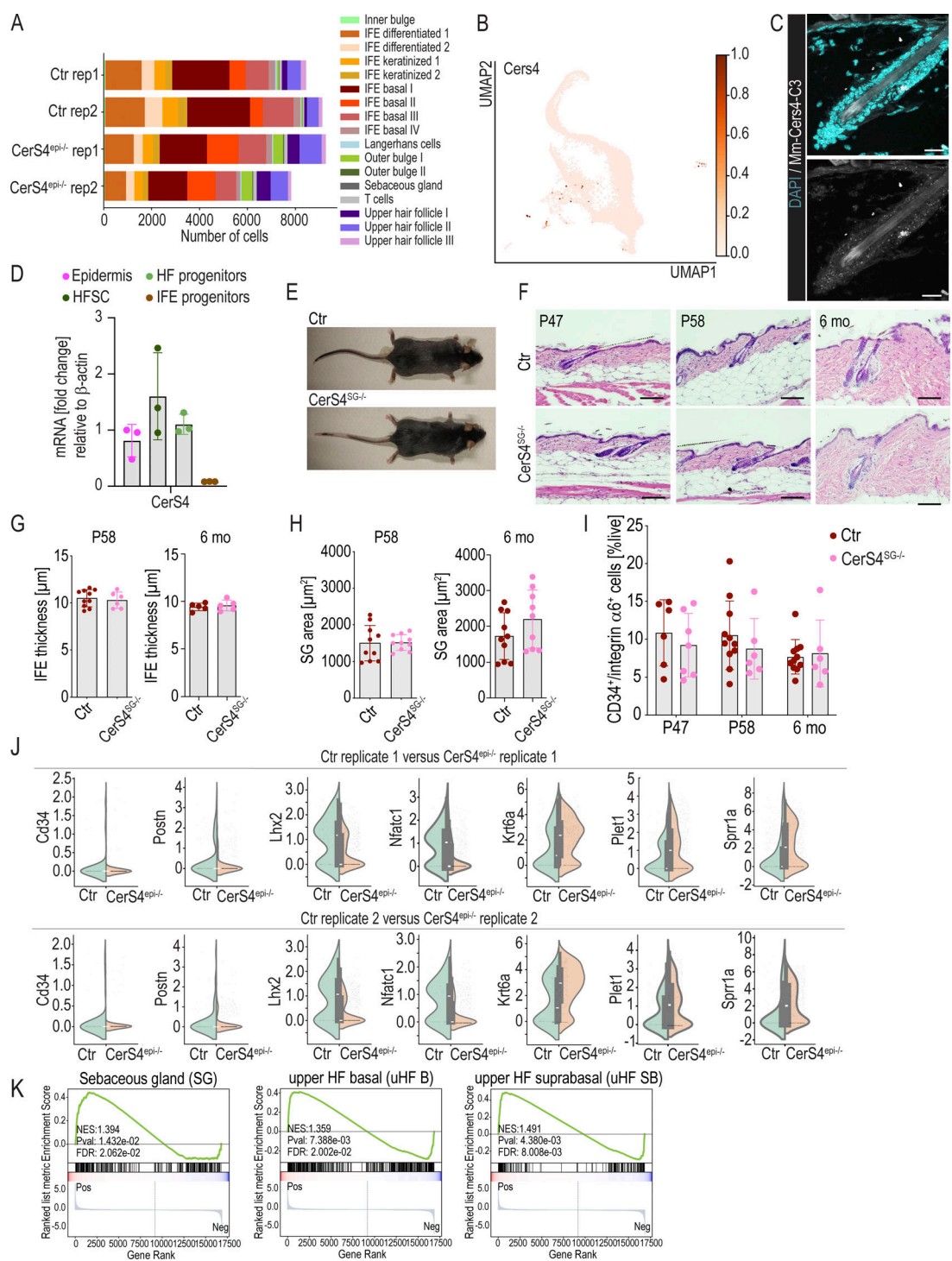

Figure S2. **Single cell RNA sequencing analyses in epdermis-specofof CerS4 knockout and characterization of sebaceous gland-specifc CerS4 knockout mice. (A)** Cell counts of cell clusters determined by scRNAseq in two control (Ctr) and two CerS4[epi−/−] mice in a replicate resolved resolution. **(B)** UMAP projection of CerS4-expressing cells in scRNAseq data of control mice. **(C)** In situ hybridization of CerS4 mRNA expression in back skin sections of control mice. Scale bars 20 μm. **(D)** RT-qPCR analysis of control FACS-sorted HFSCs, HF progenitors, and IFE progenitor cells and total epidermis (n = 3 mice). **(E)** Macroscopic images of Ctr and CerS4[SCD3+/−] mice. **(F)** H/E-stained back skin sections of Ctr and CerS4[SG−/−] back skin at P47, P58, and 6 mo. Scale bars 100 μm. **(G)** Quantification of the thickness of the IFE at P58 (n = 10 control, 6 CerS4[SG−/−] mice) and 6 mo (n = 5 mice/genotype) from back skin sections. **(H)** Quantification of the size of SG at P58 (n = 10 mice/genotype) and 6 mo (n = 10 control, 9 CerS4[SG−/−] mice) from back skin sections. **(I)** FACS analysis of CD34+ integrin α6+ HFSCs from P47 (n = 6 control, 7 CerS4[epi−/−] mice), P58 (n = 11 control, 6 CerS4[epi−/−] mice), and 6-mo-old mice (n = 11 control, 6 CerS4[epi−/−] mice). **(J)** Gene expression levels of selected stem and progenitor cell marker genes from scRNAseq data in the OB stem cell compartment in Ctr and CerS4[epi−/−] skin in a replicate resolved presentation. Note that Ctr and CerS4[epi−/−] replicates show comparable changes. **(K)** GSEA of differentially expressed genes in Ctr and CerS4[epi−/−] OB stem cells indicate overrepresentation of SG signature genes and divergence in uHF signature genes (from Joost et al. [2016]) in Ctr versus CerS4[epi−/−] OB stem cells. NES, normalized enrichment score.

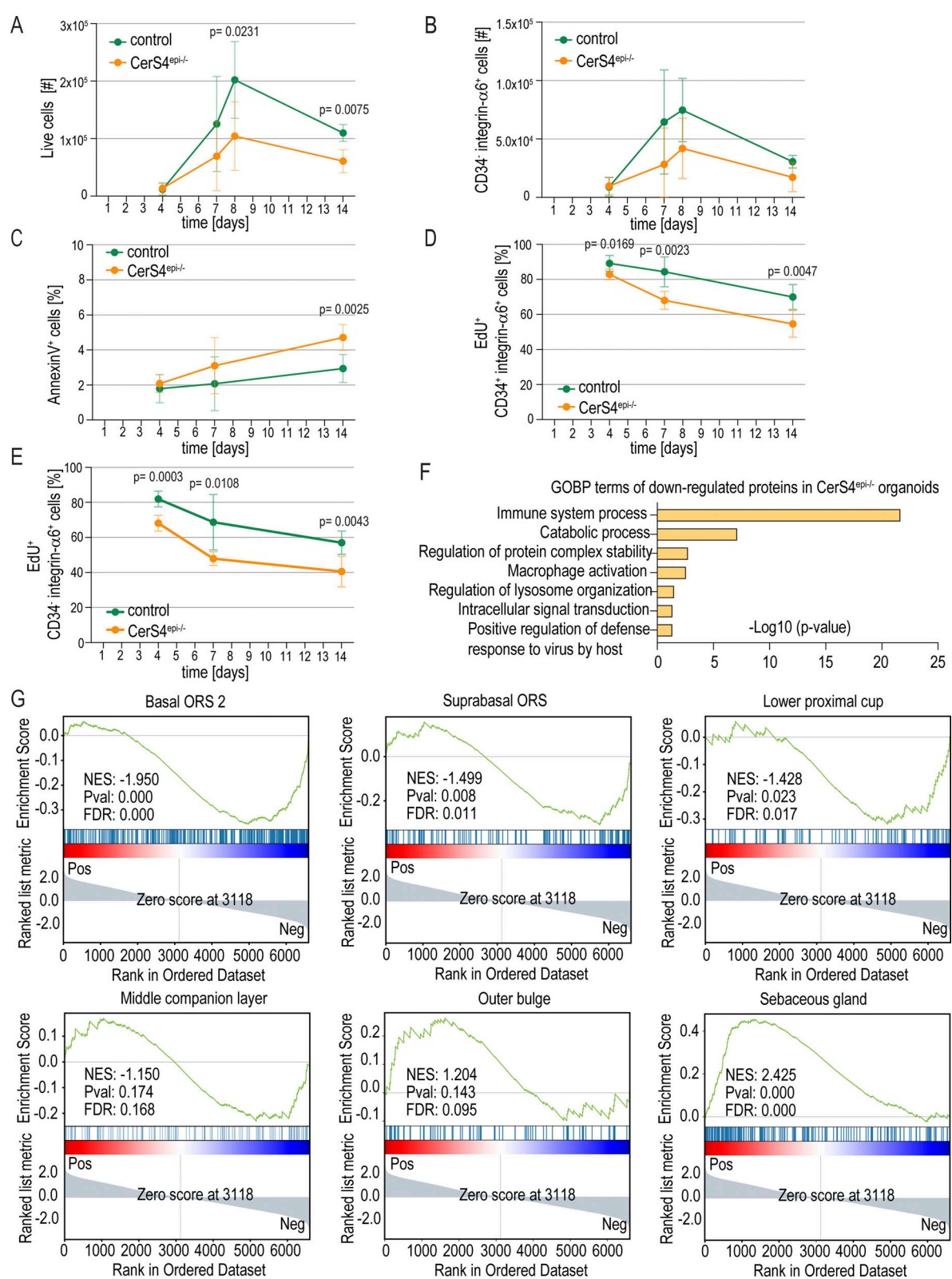

Figure S3. **Analyses of hair follicle stem cell organoids from CerS4-deficient mice. (A and B)** FACS-based quantification of live cells (A) and CD34⁻ integrin α6⁺ non-HFSCs (B) from Ctr and CerS4^epi−/− organoids cultured for indicated times ($n$ = 4 mice/genotype for 14 days, six mice/genotype for all other time points; mean ± SD; unpaired $t$ test). **(C)** FACS-based quantification of annexin-positive cells of Ctr and CerS4^epi−/− organoids cultured for indicated time points ($n$ = 6 mice/genotype; mean ± SD; unpaired $t$ test). **(D and E)** FACS-based quantification of EdU-positive live cells in CD34⁺ integrin α6⁺ HFSCs (D) and CD34⁻ integrin α6⁺ non-HFSCs (E) after a 24-h chase in organoids cultured for indicated time points ($n$ = 6 mice/genotype; mean ± SD; unpaired $t$ test). **(F)** GO-term enrichment analysis from proteins downregulated in CerS4^epi−/− organoids. **(G)** GSEA of known markers of distinct skin hair follicle progenitor cell linages (Joost et al., 2016) from differentially expressed proteins in Ctr and CerS4^epi−/− organoids. NES, normalized enrichment score.

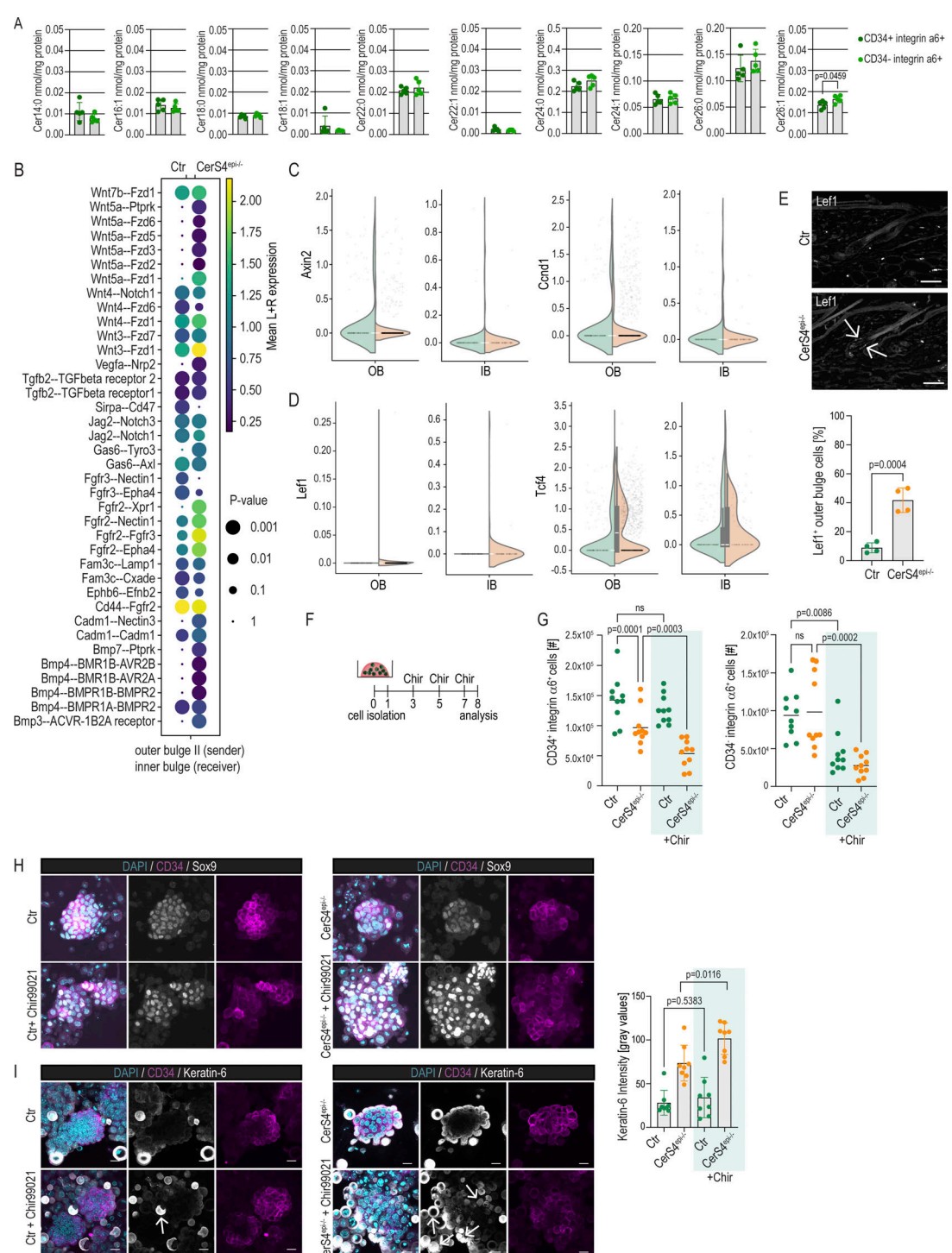

Figure S4. **Altered differentiation trajectories and Wnt signaling in CerS4-deficient HFSCs. (A)** Ceramide levels in FACS-purified CD34+ integrin α6+ HFSCs and CD34− integrin α6+ non-HFSCs determined by quantitative LC-ESI-MS/MS analysis ($n = 5$ mice/genotype; ratio paired $t$ test). **(B)** Receptor–ligand interaction analyses (CellPhoneDB) from scRNAseq data of control (Ctr) and CerS4epi−/− skin. Communication from the OB cells to the IB is illustrated. **(C and D)** RNA expression levels of selected canonical Wnt target genes in Ctr and CerS4epi−/− skin from scRNAseq data. **(E)** Representative images and quantification of P21 Ctr and CerS4epi−/− back skin sections stained for Lef1 (grey) and keratin-14 (magenta). Note increased Lef1 expression (arrows) in CerS4epi−/− OB (scale bar 50 μm; $n = 4$; mean ± SD; unpaired $t$ test). **(F)** Experimental outline for Wnt activation using Chir99021. **(G)** FACS-based quantification of HFSCs and non-HFSCs from Ctr and CerS4epi−/− organoids treated with Chir99021 (1 μM). Note a decrease in HFSCs in Chir99021-treated CerS4epi−/− organoids but not in Ctr organoids ($n = 10$ mice/genotype, mean ± SD; unpaired $t$ test). **(H)** Representative images of Ctr and CerS4epi−/− organoids treated with Chir99021 and stained for Sox9 (grey) and keratin-14 (magenta). Scale bars 20 μm. **(I)** Representative images and quantification of Ctr and CerS4epi−/− organoids treated with Chir99021 and stained for keratin-6 (grey) and keratin-14 (magenta). Note increased keratin-6 expression and loss of cell cohesion in CerS4epi−/− organoids, indicated by arrows (scale bar 20 μm; $n = 8$ organoids/genotype; mean ± SD; unpaired $t$ test). LC-ESI-MS/MS, LC coupled to electrospray ionization tandem MS/MS.

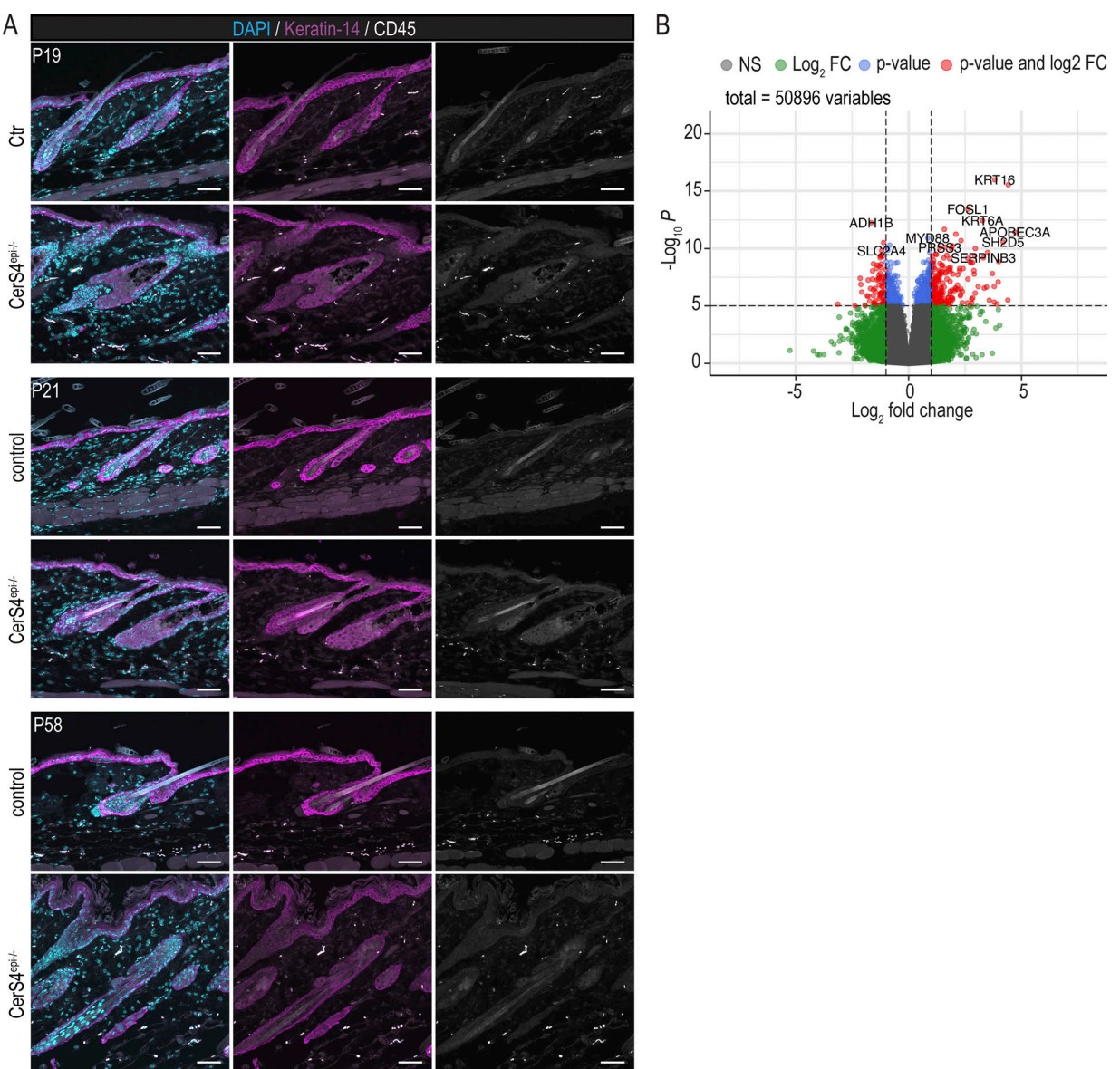

Figure S5. **Analyses of CD45-positive cells in CerS4-deficient mouse skin and differential gene experssion analyses in atopic dermatitis patient skin.** **(A)** Representative images of P19, P21, and P58 Ctr and CerS4$^{epi-/-}$ back skin sections stained for CD45 (grey) and keratin-14 (magenta). Scale bars 50 μm. **(B)** Differential gene expression of 20 patients diagnosed with AD. Positive values indicate genes that are upregulated in lesional skin and negative values indicate genes that are upregulated in non-lesional skin from Federico et al. (2020). NS = not significant; FC = fold change.

Video 1. **Representative movie of control (Ctr), CerS4$^{epi-/-}$, and CerS4$^{epi-/-}$ cells transfected with CerS4-HALO imaged with Fluo-4 AM Ca$^{2+}$ probe to identify oscillations.** 5 μM KN93 was added at 10 min. Note that enhanced oscillations were upstream of CaMKII signaling restored to control levels by rescuing CerS4 expression. Images were aquired at 30 frames/second. Scale bars 10 μm.

**Provided online are Table S1, Table S2, and Table S3. Table S1 shows the differentially abundant proteins in control and CerS4$^{epi-/-}$ HFSC organoids. Table S2 shows the differentially expressed genes in the OB populations of control and CerS4$^{epi-/-}$ mice from scRNAseq data. Table S3 shows the selected significantly upregulated proteins in CerS4$^{epi-/-}$ HFSC organoids that are found in the Human Phenotype Ontology database.**

