## [Peer Review File · The Journal of Cell Biology]

Sphingolipid metabolism orchestrates establishment of the hair follicle stem cell compartment

Franziska Peters, Windie Höfs, Hunki Lee, Susanne Brodesser, Kai Kruse, Hannes Drexler, Jiali Hu, Verena Raker, Dominika Lukas, Esther von Stebut, Martin Krönke, Carien Niessen, and Sara Wickström

Corresponding Author(s): Sara Wickström, Max Planck Society and Franziska Peters, Max Planck Institute for Molecular Biomedicine

Review Timeline:

Submission Date:	2024-03-14
Editorial Decision:	2024-03-21
Revision Received:	2024-11-04
Editorial Decision:	2024-12-23
Revision Received:	2025-01-04

Monitoring Editor: Elaine Fuchs

Scientific Editor: Tim Fessenden

Transaction Report:

DOI: <https://doi.org/10.1083/jcb.202403083>

Revision 0

Review #1

1. Evidence, reproducibility and clarity:

Evidence, reproducibility and clarity (Required)

Deletion of CerS4 in the entire mouse epidermis throughout development via the K14-Cre results in enlarged sebaceous glands and perturbed HFSC molecular phenotype. There is low or no expression of CD34, a known marker of the HFSCs along with apparent reduction of several other HFSC markers and acquisition of a more differentiated cell phenotype in these cells. Interestingly, skin and hair follicles seem to remain normal otherwise up to advanced age, though this contradicts the notion that HFSC were indeed affected at the functional level. The data does not demonstrate 'gradual decline' in the HFSC compartment, as claimed by the authors, but rather seem to indicate that the adult HFSC compartment is not properly established in its molecular signatures. Organoid cultures document defects in HFSC, which included reduced proliferation in the CerS4 KO cells. Lipid composition in plasma membranes was also affected by the CerS4 KO. Associated with this, Wnt signal transduction is also affected according to experiments that enhance the strength of wnt signals via a specific small molecular agonist of the pathway. Finally, the authors discover a resemblance of the mouse KO immune-phenotype, with human atopic dermatitis. The study is likely of interest to a specialized readership in skin biology and dermatology and adds to previous studies on CerS4 in skin that erroneously placed its role in the sebaceous gland. [The authors here demonstrate that deletion of CerS4 in the sebaceous glands via SCD3-Cre led to no phenotype, contradicting the previous assessment that CerS4 is important in sebaceous glands.] The study would need to be corrected in a few of its interpretations regarding stem cells to better match the data, as indicated below.

****Major revisions:****

Fig1B - the data seems to simply shows that bulge cells express less or no CD34 and not that 'CerS4^{epi}/- mice showed reduced HFSC numbers'; the primary FACS data should be shown somewhere too.

The conclusion that stemness is affected, and HFSCs lose their normal gene expression signature is at more convincing after looking at other HFSC markers down the road in the paper. However, in the absence of functional assays that would demonstrate stem cell function is lacking and seeing that hair follicles are maintained and grow in long-term, the notion that stem cells are lacking in these conditions is not supported by the data.

The conclusion after figure 2: "Collectively, these data indicate that CerS4-deficiency triggers gradual depletion of the quiescent HFSC compartment." There is no data showing gradual depletion of the quiescent HFSC compartment. We would need to see a gradual activation of HFSCs with over proliferation to conclude this. There is some data albeit not always convincing (see NFAC1 staining in Fig. 5C) indicating loss of markers associated with quiescence but there is no data indicating 'gradual' loss of markers.

****Minor revisions:****

Figure 1C legend - please spell out what are the abbreviations for the different subpopulations; please show these populations as % as opposed to absolute numbers.

Figure 1D - please make it clear in the cartoon what the different sub-populations listed are;

Is OB1 a CD34- HFSC population?

Fig. 3B - the colors of the legend do not match the colors in the data so it is confusing as to which one is which!

Fig 5C - the differences in NFATc1 are not visible in the images shown

2. Significance:

Significance (Required)

Deletion of CerS4 in the entire mouse epidermis throughout development via the K14-Cre results in enlarged sebaceous glands and perturbed HFSC molecular phenotype. There is low or no expression of CD34, a known marker of the HFSCs along with apparent reduction of several other HFSC markers and acquisition of a more differentiated cell phenotype in these cells. Interestingly, skin and hair follicles seem to remain normal otherwise up to advanced age, though this contradicts the notion that HFSC were indeed affected at the functional level. The data does not demonstrate 'gradual decline' in the HFSC compartment, as claimed by the authors, but rather seem to indicate that the adult HFSC compartment is not properly established in its molecular signatures. Organoid cultures document defects in HFSC, which included reduced proliferation in the CerS4 KO cells. Lipid composition in plasma membranes was also affected by the CerS4 KO. Associated with this, Wnt signal transduction is also affected according to experiments that enhance the strength of wnt signals via a specific small molecular agonist of the pathway. Finally, the authors discover a resemblance of the mouse KO immune-phenotype, with human atopic dermatitis. The study is likely of interest to a specialized readership in skin biology and dermatology and adds to previous studies on CerS4 in skin that erroneously placed its role in the sebaceous gland. [The authors here demonstrate that deletion of CerS4 in the sebaceous glands via SCD3-Cre led to no phenotype, contradicting the previous assessment that CerS4 is important in sebaceous glands.]

3. How much time do you estimate the authors will need to complete the suggested revisions:

Estimated time to Complete Revisions (Required)

(Decision Recommendation)

Less than 1 month

4. Review Commons values the work of reviewers and encourages them to get credit for their work. Select 'Yes' below to register your reviewing activity at Web of Science Reviewer

Recognition Service (formerly Publons); note that the content of your review will not be visible on Web of Science.

Yes

Review #2

1. Evidence, reproducibility and clarity:

Evidence, reproducibility and clarity (Required)

The manuscript authored by Peters et al. titled "Sphingolipid metabolism orchestrates the establishment of the adult hair follicle stem cell niche to control skin homeostasis" elucidates the critical role of ceramide synthase 4 (CerS4) in the epidermal stem cell niche, particularly in regulating hair follicle bulge stem cells (HFSCs). Using epidermal specific CerS4 knockout mice as an in vivo model and hair follicle organoid culture as an ex vivo model, the authors conducted a comprehensive analysis, which includes cutting edge approaches such as scRNA-seq, proteomics, and lipidomics. The results highlight CerS4's function in the establishment/maintenance of the HFSC niche, as absence of CerS4 changes HFSCs' number and differentiation state. Potential underlying mechanisms identified include altered membrane lipid profiles and Wnt signaling responsiveness. Possible link to a chronic inflammatory skin disease, atopic dermatitis, is also implicated. The data presented are generally of high quality, and the work is significant as it uncovers a new regulator of HFSC fate with mechanistic connection to lipid metabolism.

However, some issues were identified, most of them having to do with in vivo characterization and data interpretation:

****Major:****

1. The in vivo HFSC phenotype can be better characterized. "Collective, these data show that CerS4 in HFSCs is essential to establish the adult stem cell compartment and to assure lineage fidelity." - this statement premature based on order of the data shown. Also the trajectory difference shown in Figure 2A is not striking. Subclustering out the relative cell subsets and redo the analysis might help to tease out the difference. Additional experiments such as lineage tracing would be useful to support the notion that there is lineage fidelity issue in the mutant - though it is understood that this is quite involved and may lie outside the scope of the current study. Are bulge cells in the mutant proliferative? - the authors should consider in vivo Edu labelling experiment or the like to assess the quiescence/proliferation of the bulge cells. Finally, analyzing hair follicles at earlier stages might help to clarify when and where the bulge and sebaceous gland changes start - is possible that aberrant divergence of bulge/sebaceous fates occur prior to the establishment of a stable bulge fate?

2. The exclusion of IFE contribution is not backed up by data. Figure 6D - model emphasizes HFSC involvement in atopic dermatitis, but this could be due to epidermal barrier defect. Barrier defect

could already be present even though IFE morphology appears normal. Maybe TEWL is measured at the time of analysis and shows no change - if so, this data should be included. HFSC changes might contribute but the involvement of IFE cannot be excluded. The conclusion that "CerS4 expression was restricted to the hair follicle" is not supported by data. IFE expression is apparent in Figure S1C. Along this line, there is also an apparent expansion of IFE basal II in the mutant (Figure 1C).

3. Figure 1 - single cell analysis was done using only 2 pairs of mice, and data in E lack statistical assessment. At the very least, data for individual pairs should be shown in supplemental data to ensure that changes are consistent in both mutant mice rather than being dominated by dramatic alteration in only 1 mutant mouse.

****Minor:****

1. CerS4SCD3-/+ nomenclature is mis-leading.
2. Figure 2- "Furthermore, we observed expansion of the inner bulge identity marker Krt6 protein expression into outer bulge stem cells and along the infundibulum in CerS4epi-/- hair follicles, whereas in control mice Krt6 was restricted to the inner bulge (Fig. 2C)." - Krt6 staining is presented in Fig 2D, not 2C.
3. Figure 3C - size of the organoids should be quantified with statistics. The images shown do not support the statement that "Strikingly, CerS4epi-/- organoids showed altered morphology characterized by smaller size and loss of cohesion of peripheral cells from the organoid clusters (Fig. 3C), ...".
4. Section titled "CerS4 regulates HFSC differentiation in a stem cell autonomous manner": "CD34-integrin- a6+ cells, which based on extensive transcriptome and marker expression analyses represent a mixture of HFSCs, hair follicle outer root sheath (ORS) cells and inner bulge cells (collectively termed non-HFSCs)." - shouldn't the CD34- integrin- a6+ population also contain IFE stem/progenitor cells? Are hair follicles micro-dissected out for FACS?
5. Figure 5D - please provide the working concentration of Chir99021.
- Figure 5F - explain what arrows mean in legends.
6. Figure 6A - no significant changes in Th2 and ILC2 were observed at a 95% confidence interval. Increasing mouse number will help to increase statistical power.
7. Additional Wnt target genes such as Axin 2 should be looked at.
8. The increased BMP signaling and decreased Nfatc1 expression are seemingly contradictory.
9. Paragraph starting with "It is interesting to note that ceramide availability was shown to regulate Wnt signaling in Drosophila through strong effects on recycling endocytosis of the receptors (Pepperl et al., 2013)." Is redundant in the manuscript.

2. Significance:

Significance (Required)

The work uncovers a new regulator of HFSC fate with mechanistic connection to lipid metabolism and development signaling. The same group previously reported epidermal and hair cycling phenotypes of the same mutant mice, but this work now identifies a specific defect in HFSCs and present evidence for cellular, molecular and biochemical changes. Linking stem cell regulation to lipid metabolism is conceptually novel, and should have a broad audience. However, the study does have some limitations, such as lack of definitive evidence that CerS4 function in HFSCs is

responsible for all the defects reported here, and that lipid alterations have a causal relationship with altered Wnt signaling.

My expertise is in skin biology, stem cell control, and developmental signaling.

3. How much time do you estimate the authors will need to complete the suggested revisions:

Estimated time to Complete Revisions (Required)

(Decision Recommendation)

Between 3 and 6 months

4. Review Commons values the work of reviewers and encourages them to get credit for their work. Select 'Yes' below to register your reviewing activity at Web of Science Reviewer Recognition Service (formerly Publons); note that the content of your review will not be visible on Web of Science.

Yes

Review #3

1. Evidence, reproducibility and clarity:

Evidence, reproducibility and clarity (Required)

The authors created CerS4 mutant mice to test the role of sphingolipids in hair follicle stem cells (HFSCs) and the hair cycle. This work extends previous studies that show that loss of this enzyme leads to defects in the hair cycle and eventually hair loss. In this study the authors look early on in the course of the deletion in an attempt to understand why loss of this enzyme leads to the phenotype described previously. They use single cell profiling, proteomics, and in situ imaging to pinpoint issues in the stem cell niche that drive phenotypes and propose novel interactions between sphingolipid metabolism, Wnt signaling, and inflammation in regulation of HFSC homeostasis. The data are nicely presented, and the text is well written. The conclusions are clearly defined.

While the data are clearly presented, there are numerous issues that are confusing to this reviewer. In addition, some of the phenotypes described are subtle, and thus do not make a convincing case.

1. In figure 1C, the cell proportion analysis suggests there are no OBII or SG in WT. I am not sure how this could be possible. In addition, there appears to be almost no sebaceous cells in either, but the mutant supposedly has much larger sebaceous glands (in Fig 2). In Fig S1I, there is no change in bulge cells? In Fig 1B, there is less HFSCs in the mutant than in the WT, but in 1C, there is more OBI in the mutant. The results in Fig 1B and C are confusing. Also, the schematic in Fig 1 is hard to read, the authors should color code the text with the image.

2. In Figure 5, the signaling chart shows a strong upregulation of non-canonical Wnt signaling in the

mutant bulge. Canonical Wnt signaling appears to be unchanged between wt and ko. Thus, it is not clear why the authors came to the conclusion that Wnt signaling is induced in the mutant. They further show expression of Lef1 and Nfatc1, but these are not typical markers used to denote canonical wnt activation, as implied. In fact, the data in Fig S4B suggest the induction of Lef1 and Tcf4 is actually very subtle. Instead, the authors should use nuclear b-catenin or transcriptional targets such as Axin or CyclinD. The authors should in fact explore the observation of Wnt5, as that appears to be the most dramatic change. In addition, the authors should use an ontological analysis with the single cell data from the tissue in the same manner that they did for organoids to take another look at molecular consequences of loss of CerS4.

3. The authors suggest that much of the phenotype is due to inflammation. In Fig 6A, they showed analysis of CD45 cells in the skin. However, the only change was a very subtle change in Th2 cells, while no other CD45+ cells were altered.

4. The authors showed upregulation of Immune response in Fig 6C, but then in Fig S2, the genes downregulated are also related to immune response...how do the authors reconcile this?

5. The author propose that the phenotype in CerS4 null mice is due to disruption of the stem cell Niche. However, the authors have not shown evidence for such an effect through any in situ analysis. The single cell approaches are valuable, but in that case the niche is dissociated. The organoid work is also nice, but not exactly a stem cell niche either. The authors should instead test their hypothesis through an in situ analysis.

2. Significance:

Significance (Required)

The authors created CerS4 mutant mice to test the role of sphingolipids in hair follicle stem cells (HFSCs) and the hair cycle. This work extends previous studies that show that loss of this enzyme leads to defects in the hair cycle and eventually hair loss. In this study the authors look early on in the course of the deletion in an attempt to understand why loss of this enzyme leads to the phenotype described previously. They use single cell profiling, proteomics, and in situ imaging to pinpoint issues in the stem cell niche that drive phenotypes and propose novel interactions between sphingolipid metabolism, Wnt signaling, and inflammation in regulation of HFSC homeostasis. The data are nicely presented, and the text is well written. The conclusions are clearly defined.

3. How much time do you estimate the authors will need to complete the suggested revisions:

Estimated time to Complete Revisions (Required)

(Decision Recommendation)

Between 3 and 6 months

4. Review Commons values the work of reviewers and encourages them to get credit for their work. Select 'Yes' below to register your reviewing activity at Web of Science Reviewer Recognition Service (formerly Publons); note that the content of your review will not be visible on Web of Science.

No

Revision Plan

Manuscript number: RC-2024-02343

Corresponding author(s): Sara A. Wickström

General Statements [optional]

We thank the reviewers for their thoughtful comments on our manuscript “Sphingolipid metabolism orchestrates the establishment of the adult hair follicle stem cell niche to control skin homeostasis”.

We were delighted to see that the reviewers found our work “conceptually novel”, of “broad interest” and “well-written” and “well presented”. Importantly, the main finding of this manuscript is that hair follicle stem cells contain a unique lipid profile and perturbing this profile by deleting CerS4 leads to profound defects in stem cell fate regulation through Wnt signaling. This is a completely new finding that has implications far beyond skin biology and is of general interest as also stated by Reviewer 2. Thus, we respectfully disagree with Reviewer 1 assessment that the work is of interest to specialists and that the work simply rebuts previous findings on the role of Cers4 in the sebaceous gland. Our understanding of the role of ceramides in stem cells is only beginning to emerge and this work provides a comprehensive, mechanistic study of specific function of CerS4 in hair follicle stem cells.

Description of the planned revisions

Reviewer #1 (Evidence, reproducibility and clarity (Required)):

Deletion of CerS4 in the entire mouse epidermis throughout development via the KI4-Cre results in enlarged sebaceous glands and perturbed HFSC molecular phenotype. There is low or no expression of CD34, a known marker of the HFSCs along with apparent reduction of several other HFSC markers and acquisition of a more differentiated cell phenotype in these cells. Interestingly, skin and hair follicles seem to remain normal otherwise up to advanced age, though this contradicts the notion that HFSC were indeed affected at the functional level. The data does not demonstrate 'gradual decline' in the HFSC compartment, as claimed by the authors, but rather seem to indicate that the adult HFSC compartment is not properly established in its molecular signatures. Organoid cultures document defects in HFSC, which included reduced proliferation in the CerS4 KO cells. Lipid composition in plasma membranes was also affected by the CerS4 KO. Associated with this, Wnt signal transduction is also affected according to experiments that enhance the strength of wnt signals via a specific small molecular agonist of the pathway. Finally, the authors discover a resemblance of the mouse KO immune-phenotype, with human atopic dermatitis. The study is likely of interest to a specialized readership in skin biology and dermatology and adds to previous studies on CerS4 in skin that erroneously placed its role in the sebaceous gland. [The authors here demonstrate that deletion of CerS4 in the sebaceous glands via SCD3-Cre led to no phenotype, contradicting the previous

Revision Plan

assessment that CerS4 is important in sebaceous glands.] The study would need to be corrected in a few of its interpretations regarding stem cells to better match the data, as indicated below.

We thank the reviewer for the constructive comments that will help us to improve the manuscript. In particular, it is clear that we have not been sufficiently clear in the data presentation.

Firstly, contrary to what the reviewer states, the CerS4^{epi-/-} mice have a very strong hair follicle phenotype that results in complete hair loss. Also the epidermis is not normal as an inflammatory phenotype develops later, after the hair follicle architecture and function has been disrupted. Thus, there are clear functional consequences to the hair follicle and epidermis that arise from the dysfunction of the HFSC compartment. We will edit the manuscript and add photodocumentation of the macroscopic phenotype to ensure clarity.

We fully agree with the reviewer that the initial phenotype is inability to establish the adult hair follicle stem cell niche, as shown by the single cell sequencing data and as also stated in the manuscript title. We will further edit the manuscript to clarify this conclusion. Importantly, however, some hair follicle stem cells are generated but these become gradually depleted. So there is a dual phenotype: an inability to efficiently establish and maintain the hair follicle stem cell population. We will clarify this in the text.

Finally, we want to emphasize that the main finding of this manuscript is that hair follicle stem cells contain a unique lipid profile and perturbing this profile by deleting CerS4 leads to profound defects in stem cell fate regulation through Wnt. This is a completely new finding that has implications far beyond dermatology.

Major revisions:

Fig1B - the data seems to simply shows that bulge cells express less or no CD34 and not that ' CerS4epi-/- mice showed reduced HFSC numbers'; the primary FACS data should be shown somewhere too.

Outer bulge hair follicle stem cells are defined as a population of cells that expresses CD34 and integrin- α 6. The quantifications in Fig 1B show the quantitative FACS analyses of the size of this population and indicate less CD34⁺/integrin- α 6⁺ cells in CerS4^{epi-/-} epidermis. The mean fluorescence intensity of CD34 and integrin- α 6 was not reduced in these CerS4^{epi-/-} stem cells. This FACS analysis therefore allows the conclusion that there are less CD34⁺/integrin- α 6⁺ cells in CerS4^{epi-/-} epidermis. We will include the original FACS plot data to support this notion and the quantifications.

The conclusion that stemness is affected, and HFSCs lose their normal gene expression signature is at more convincing after looking at other HFSC markers down the road in the paper. However, in the absence of functional assays that would demonstrate stem cell function is lacking and seeing that hair follicles are maintained and grow in long-term, the notion that stem cells are lacking in these conditions is not supported by the data.

Revision Plan

We appreciate that the reviewer finds the marker gene analysis convincing. To assay stem cell functionality, we have used the organoid assays (spheroid formation is classical, widely used assay for stemness). Using these functional assays we observe impaired self-renewal of stem cells (Fig. 3D), enhanced differentiation (Fig. 3I), and altered Wnt responsiveness (Fig. 5 E, F), all indicative of stem cell dysfunction and explaining the *in vivo* phenotypes of altered stem cell differentiation and inability to establish and maintain the stem cell population.

In the revised manuscript we will also include measurements of stem cell self-renewal *in vivo* using BrdU incorporation and provide more detailed description on the hair loss phenotype of the mice to further strengthen this conclusion.

The conclusion after figure 2: "Collectively, these data indicate that CerS4-deficiency triggers ... gradual depletion of the quiescent HFSC compartment." There is no data showing gradual depletion of the quiescent HFSC compartment. We would need to see a gradual activation of HFSCs with over proliferation to conclude this. There is some data albeit not always convincing (see NFAC1 staining in Fig. 5C) indicating loss of markers associated with quiescence but there is no data indicating 'gradual' loss of markers.

We agree with the reviewer that showing gradual activation of the HFSCs *in vivo* is important to conclude loss of quiescence. We will include *in situ* stainings of *in vivo* BrdU labeling and quantify proliferation in the hair follicle bulge stem cell region. Preliminary data of P47 mice already shows a clear increase in BrdU⁺ cell in the stem cell compartment in CerS4^{epi-/-} skin (see below). Further analysis at P21 will be carried out during the revision.

Minor revisions:

Figure 1C legend - please spell out what are the abbreviations for the different subpopulations; please show these populations as % as opposed to absolute numbers.

We will edit the Figure 1C legend for clarity and express the populations as %.

Figure 1D - please make it clear in the cartoon what the different sub-populations listed are;

Revision Plan

We will edit the cartoon for clarity.

Is OB1 a CD34- HFSC population?

The outer bulge 1 (OB1) is a population of cells that expresses hair follicle stem cell markers, including CD34. We will clarify this in the legend.

Fig. 3B - the colors of the legend do not match the colors in the data so it is confusing as to which one is which!

We will alter the colors to match the data and thank the reviewer for pointing this out.

Fig 5C - the differences in NFATc1 are not visible in the images shown

We apologize for the suboptimal quality of these images and will replace them with higher resolution images to more clearly demonstrate the difference.

Reviewer #2 (Evidence, reproducibility and clarity (Required)):

The manuscript authored by Peters et al. titled "Sphingolipid metabolism orchestrates the establishment of the adult hair follicle stem cell niche to control skin homeostasis" elucidates the critical role of ceramide synthase 4 (CerS4) in the epidermal stem cell niche, particularly in regulating hair follicle bulge stem cells (HFSCs). Using epidermal specific CerS4 knockout mice as an in vivo model and hair follicle organoid culture as an ex vivo model, the authors conducted a comprehensive analysis, which includes cutting edge approaches such as scRNA-seq, proteomics, and lipidomics. The results highlight CerS4's function in the establishment/maintenance of the HFSC niche, as absence of CerS4 changes HFSCs' number and differentiation state. Potential underlying mechanisms identified include altered membrane lipid profiles and Wnt signaling responsiveness. Possible link to a chronic inflammatory skin disease, atopic dermatitis, is also implicated. The data presented are generally of high quality, and the work is significant as it uncovers a new regulator of HFSC fate with mechanistic connection to lipid metabolism.

We thank the reviewer for the positive assessment of our work and finding it to be of high quality and significance. We further appreciate the constructive comments that will further help us to improve the manuscript.

However, some issues were identified, most of them having to do with in vivo characterization and

Revision Plan

data interpretation:

Major:

1. The in vivo HFSC phenotype can be better characterized. "Collective, these data show that CerS4 in HFSCs is essential to establish the adult stem cell compartment and to assure lineage fidelity." - this statement premature based on order of the data shown. Also the trajectory difference shown in Figure 2A is not striking. Subclustering out the relative cell subsets and redo the analysis might help to tease out the difference. Additional experiments such as lineage tracing would be useful to support the notion that there is lineage fidelity issue in the mutant - though it is understood that this is quite involved and may lie outside the scope of the current study. Are bulge cells in the mutant proliferative? - the authors should consider in vivo Edu labelling experiment or the like to assess the quiescence/proliferation of the bulge cells. Finally, analyzing hair follicles at earlier stages might help to clarify when and where the bulge and sebaceous gland changes start - is possible that aberrant divergence of bulge/sebaceous fates occur prior to the establishment of a stable bulge fate?

We thank the reviewer for suggesting additional analyses of the single cell sequencing data. We have performed subclustering of the relevant populations for the trajectory analyses to more clearly demonstrate the altered lineage trajectories. We will include these new analyses in the manuscript. Importantly, the *in vitro* organoids show abnormal differentiation (Fig. 3H, 3I, Supplementary Fig. 2G), closely resembling the *in vivo* phenotypes, thereby strengthening the conclusion of cell-autonomously altered lineage trajectories of the hair follicle stem cells.

We will further perform *in vivo* BrdU labeling as suggested. These experiments have already been initiated and preliminary data show increased proliferation in the bulge stem cell region of CerS4^{epi/-} mice in older mice (P47). The data will be included to emphasize long term loss of quiescence in the stem cell compartment in CerS4^{epi/-} mice.

Understanding the early development of bulge and sebaceous fates is indeed an interesting question. This will be addressed by detailed analyses of stem cell fate at early stages (P17-P21) using key markers of stem cell state and sebaceous lineages (CD34, Krt15, Lhx2, Nfatc1, SCD1 and FASN).

Finally, we have initiated lineage tracing experiments using the stem cell-specific Lgr5-Cre to conclusively demonstrate that Cers4-deletion leads to altered routing of hair follicle stem cells into upper hair follicle and sebaceous gland fates. This notion is supported by the preliminary analyses of these experiments. We will finalize these analyses and include them in the manuscript.

2. The exclusion of IFE contribution is not backed up by data. Figure 6D - model emphasizes HFSC involvement in atopic dermatitis, but this could be due to epidermal barrier defect. Barrier defect could already be present even though IFE morphology appears normal. Maybe TEWL is measured at the time of analysis and shows no change - if so, this data should be included. HFSC changes might contribute but the involvement of IFE cannot be excluded. The conclusion that "CerS4 expression was

Revision Plan

restricted to the hair follicle" is not supported by data. IFE expression is apparent in Figure S1C. Along this line, there is also an apparent expansion of IFE basal II in the mutant (Figure 1C).

We acknowledge that we have not been clear enough with the evidence that allowed us to exclude the involvement of an IFE-mediated barrier defect in the early skin inflammation phenotype. To address a potential barrier defect early on, we have performed careful analysis of TEWL. In Peters et al., 2020 we demonstrate no changes in TEWL at P0, a reduced TEWL at P21 and an increased TEWL in adult CerS4^{epi-/-} mice starting only at P33. The reduction of the TEWL in adolescent CerS4^{epi-/-} mice (P21) is likely linked to an increased production of sebaceous lipids lubricating the skin surface at this time point (Peters et al., 2020). Thus, defects in the hair follicle stem cell compartment, present at adolescence (P21) arise prior to defects in the adult (P33) IFE barrier function. We will clarify this in the manuscript.

Cers4 expression is overall low in skin, as is typical for enzymes. *In situ* stainings of Cers4mRNA (Fig.S1C) indeed show a sparse signal also in the IFE. This signal is also detected in CerS4^{-/-} sections, although the KO skin cannot be conclusively used to control background as these mice were generated by deletion of exon 3 only, and Cers4 RNAscope probes might detect remnant Cers4 RNA in these mice. Importantly, our data on FACS sorted basal cells of the IFE shows no substantial Cers4 mRNA expression in IFE progenitors (Fig. S1D) and no mRNA is detected in the IFE in the single cell sequencing. Thus, while we cannot fully exclude low levels of CerS4 expression in the IFE, the levels are substantially lower than in the HFSC and SG compartments, and the phenotype, including the slight expansion of the IFE basal II population, is very minor compared to the hair follicle phenotype. However, to avoid overinterpreting our data, we will carefully edit the conclusions to be less strong on the involvement of the IFE. Furthermore, we will perform hair follicle stem cell lineage tracing experiments to strengthen the conclusion on the hair follicle stem cell-autonomous phenotype.

3. Figure 1 - single cell analysis was done using only 2 pairs of mice, and data in E lack statistical assessment. At the very least, data for individual pairs should be shown in supplemental data to ensure that changes are consistent in both mutant mice rather than being dominated by dramatic alteration in only 1 mutant mouse.

We obviously have rigorously analyzed the replicates to ensure that the phenotype is consistently present in both. We will include the separate analysis of the mice to document this and include statistical analysis.

Minor:

1. CerS4SCD3^{-/+} nomenclature is mis-leading.

We will edit this for clarity

2. *Figure 2- "Furthermore, we observed expansion of the inner bulge identity marker Krt6 protein expression into outer bulge stem cells and along the infundibulum in CerS4epi-/- hair follicles, whereas in control mice Krt6 was restricted to the inner bulge (Fig. 2C)." - Krt6 staining is presented in Fig 2D, not 2C.*

We thank the reviewer for pointing out this mistake that we will correct.

3. *Figure 3C - size of the organoids should be quantified with statistics. The images shown do not support the statement that "Strikingly, CerS4epi-/- organoids showed altered morphology characterized by smaller size and loss of cohesion of peripheral cells from the organoid clusters (Fig. 3C), ...".*

We will include quantifications.

4. *Section titled "CerS4 regulates HFSC differentiation in a stem cell autonomous manner": "CD34-integrin- $\alpha 6$ + cells, which based on extensive transcriptome and marker expression analyses represent a mixture of HFSCs, hair follicle outer root sheath (ORS) cells and inner bulge cells (collectively termed non-HFSCs)." - shouldn't the CD34- integrin- $\alpha 6$ + population also contain IFE stem/progenitor cells? Are hair follicles micro-dissected out for FACS?*

The hair follicles are not micro-dissected out for FACS, and the entire basal cell population is initially isolated. However the organoid culture conditions specifically promote the expansion of the hair follicle lineage, whereas cells of the IFE are not expanded and long term maintained as extensively documented in previous publications using this organoid system (see for example Kim et al., Cell Metabolism 2020; Chacon-Martinez EMBOJ 2016).

5. *Figurer 5D - please provide the working concentration of Chir99021.*

We will provide the working concentration.

Figure 5F - explain what arrows mean in legends.

We will define the arrows.

6. *Figure 6A - no significant changes in Th2 and ILC2 were observed at a 95% confidence interval. Increasing mouse number will help to increase statistical power.*

We agree with the reviewer and acknowledge that this experiment was unfortunately underpowered. We will repeat it with a larger cohort.

7. *Additional Wnt target genes such as Axin 2 should be looked at.*

Revision Plan

We thank the reviewer for this suggestion, we will include analyses of additional Wnt target genes.

8. *The increased BMP signaling and decreased Nfatc1 expression are seemingly contradictory.*

Single cell sequencing showed an increased BMP-signaling of outer bulge cell cells to inner bulge cells on mRNA level (Figure S4A). No alteration in BMP signaling was detected within the outer bulge stem cell compartment (Figure 5A). Nfatc1 protein expression was analyzed in the upper bulge (Figure 5C). The data indicate no differential gene expression of ligand receptor pairs mediating BMP-signaling within the outer bulge. A decrease in Nfatc1 protein expression (Fig. 5C) together with an increased proliferation (see above) and loss of label retention (Peters et al., 2015) indicates loss of quiescence in this compartment. This data does not contradict an increased BMP signaling in the inner bulge (Figure S4A). An increase in BMP signaling in the inner bulge is in line with reduced inner bulge cell cluster detected in CerS4^{epi-/-} skin via single cell sequencing, likely contributing to the hair loss observed. We will edit this paragraph to make this more clear.

9. *Paragraph starting with "It is interesting to note that ceramide availability was shown to regulate Wnt signaling in Drosophila through strong effects on recycling endocytosis of the receptors (Pepperl et al., 2013)." Is redundant in the manuscript.*

We apologize for the accidental duplication of this paragraph and thank the reviewer for noticing this mistake.

Reviewer #3 (Evidence, reproducibility and clarity (Required)):

The authors created CerS4 mutant mice to test the role of sphingolipids in hair follicle stem cells (HFSCs) and the hair cycle. This work extends previous studies that show that loss of this enzyme leads to defects in the hair cycle and eventually hair loss. In this study the authors look early on in the course of the deletion in an attempt to understand why loss of this enzyme leads to the phenotype described previously. They use single cell profiling, proteomics, and in situ imaging to pinpoint issues in the stem cell niche that drive phenotypes and propose novel interactions between sphingolipid metabolism, Wnt signaling, and inflammation in regulation of HFSC homeostasis. The data are nicely presented, and the text is well written. The conclusions are clearly defined.

We thank the reviewer for the positive assessment of our work and finding it well written and presented. We further appreciate the constructive comments that will further help us to improve the manuscript.

While the data are clearly presented, there are numerous issues that are confusing to this reviewer. In addition, some of the phenotypes described are subtle, and thus do not make a convincing case.

1, In figure 1C, the cell proportion analysis suggests there are no OBII or SG in WT. I am not sure how this could be possible. In addition, there appears to be almost no sebaceous cells in either, but the mutant supposedly has much larger sebaceous glands (in Fig 2). In Fig S11, there is no change in bulge cells? In Fig 1B, there is less HFSCs in the mutant than in the WT, but in 1C, there is more OBI in the mutant. The results in Fig 1B and C are confusing. Also, the schematic in Fig 1 is hard to read, the authors should color code the text with the image.

It is important to emphasize that the single cell RNA sequencing was carried out at P19 when the bulge stem cell compartment only starts to be established. This explains why only few bulge stem cells are detected at this point. Nevertheless, the OBII and SG cluster is visible also in the wt in Figure 1D. We will include subclustering of the relevant subpopulations to make these populations more clearly visible also in the wt. We will also edit the labels for clarity.

Mature sebocytes are very large cells, and inherent to the single cell sequencing workflows these large cells are excluded from the sequencing libraries. Importantly, we do not detect a change in bulge stem cells in a mouse line in which CerS4 was specifically deleted only in sebocytes (Figure S11). This analysis was carried out to exclude a sebocyte intrinsic effect on the hair follicle stem cell state and fate. The data does not contradict Fig. 1, as data in Fig. 1 was generated using a different mouse line in which CerS4 is deleted in the entire epidermal stem cell population using K14Cre. We will edit the manuscript to make this more clear.

Data presented in Fig 1B and C focus on two different aspects. Fig 1B shows the inefficient establishment and maintenance of CD34⁺/integrin- α 6⁺ bulge hair follicle stem cells. The quantification is based on FACS analyses of cells expressing these cell surface molecules/stem cell markers. Fig 1C shows the quantification of the various cell states based on single cell RNA expression and subsequent clustering of the control and CerS4^{epi-/-} epidermal cells together. The “outer bulge” cluster was annotated based on these cells expressing hair follicle stem cell markers. While the CerS4^{epi-/-} epidermis shows increased number of cells in this cluster, the expression of all key stem cell genes (CD34, Sox9, Krt15, Lhx2) is reduced in CerS4^{epi-/-} outer bulge 1 compartment compared to control. Thus, while this “outer bulge” population is expanded in the KO, the stem cell properties of this population are clearly attenuated, as defined by decreased expression of key stem cell transcription factors and increased expression of differentiation genes. We will clarify this in the revised version of the manuscript and also rename this cluster “outer bulge-like” to highlight that these cells are not necessarily bona fide stem cells and might not express high levels of CD34⁺/integrin- α 6⁺ protein.

2, In Figure 5, the signaling chart shows a strong upregulation of non-canonical Wnt signaling in the mutant bulge. Canonical Wnt signaling appears to be unchanged between wt and ko. Thus, it is not clear why the authors came to the conclusion that Wnt signaling is induced in the mutant. They further show expression of Lef1 and Nfatc1, but these are not typical markers used to denote canonical wnt activation, as implied. In fact, the data in Fig S4B suggest the induction of Lef1 and Tcf4 is actually very subtle. Instead, the authors should use nuclear b-catenin or transcriptional

Revision Plan

targets such as Axin or CyclinD. The authors should in fact explore the observation of Wnt5, as that appears to be the most dramatic change. In addition, the authors should use an ontological analysis with the single cell data from the tissue in the same manner that they did for organoids to take another look at molecular consequences of loss of CerS4.

We agree with the reviewer that further analysis of canonical and non-canonical Wnt signaling will strengthen this conclusion. In our experience, nuclear β -catenin is very difficult to detect in the skin even when Wnt is highly active, but we will investigate Axin2 and CyclinD1 expression. We will also investigate Wnt5a signaling by analyzing its expression as well as its downstream target genes. We will further perform additional ontological analyses from the single cell sequencing data to strengthen the conclusions on the signaling alterations.

3, The authors suggest that much of the phenotype is due to inflammation. In Fig 6A, they showed analysis of CD45 cells in the skin. However, the only change was a very subtle change in Th2 cells, while no other CD45+ cells were altered.

We agree with the reviewer and acknowledge that this experiment was unfortunately underpowered. We will repeat these analyses with a larger cohort.

4, The authors showed upregulation of Immune response in Fig 6C, but then in Fig S2, the genes downregulated are also related to immune response...how do the authors reconcile this?

We apologize for this confusion. Importantly, keratinocyte-intrinsic downregulation of homeostatic immune modulating activity is a key driver of allergic disorders, like atopic dermatitis. This barrier intrinsic immune modulation is distinct from immune cell-mediated inflammation. There is a strong overlap of genes constituting the term “Inflammatory abnormality of the skin” (Human phenotype ontology terms) Fig 6C and “Immune system process” (GOBP terms) Fig S2F. To name some, i.e. Adam-, ALOX-, ASXL- family members are annotated by both terms. Mutations in these genes are known to cause skin diseases associated with immune dysregulation but are likewise known to regulate immune responses.

Data in Figure 6C shows enrichment of this term from both up and downregulated proteins in the CerS4^{epi/-} condition compared to control, indicating that proteins involved in “Inflammatory abnormality of the skin” are dysregulated in CerS4^{epi/-} organoids. Data in Figure S2F shows the downregulation of these proteins in CerS4^{epi/-} organoids compared to control. We will clarify this in the text and figure legends.

5, The author propose that the phenotype in *CerS4* null mice is due to disruption of the stem cell Niche. However, the authors have not shown evidence for such an effect through any *in situ* analysis. The single cell approaches are valuable, but in that case the niche is dissociated. The organoid work is also nice, but not exactly a stem cell niche either. The authors should instead test their hypothesis through an *in situ* analysis.

We have used the term niche to describe the cellular interactions between stem cells and the other niche resident cells such as the Krt6^+ inner bulge cells that have been analyzed here. We will edit the conclusions for clarity. We will further include additional immunofluorescence analyses of the bulge compartment *in situ*, as suggested (including markers for quiescence and activation).

Description of the revisions that have already been incorporated in the transferred manuscript

We have not incorporated revisions.

Description of analyses that authors prefer not to carry out

We will carry out all suggested analyses.

March 21, 2024

Re: JCB manuscript #202403083T

Prof. Sara A Wickström
Max Planck Society
Max Planck Institute for Molecular Biomedicine
Roentgenstrasse 20
Münster 48149
Germany

Dear Prof. Wickström,

Thank you for submitting your manuscript entitled "Sphingolipid metabolism orchestrates the establishment of the adult hair follicle stem cell niche to control skin homeostasis".

This work combines multiple profiling techniques to study the role of ceramide synthesis in the adult hair follicle stem cell niche. Both in vivo and in vitro observations collectively suggest that ceramide contributions to sphingolipids are required for maintaining quiescence and tuning stem cell responsiveness to Wnt signals. The concept that CerS4 is required for HFSC quiescence through Wnt signaling, and that defects arising from its absence relate to inherited inflammatory skin disorders, comprise an intriguing advance that would interest JCB readers. Although the reviewers express interest in this work, there remain some important concerns that must be resolved in order for this work to be considered for publication in JCB.

Reviewers remarked on the absence of functional outcomes, including lineage fidelity, that depend on CerS4. They also noted overlap with prior work on CerS4, missing details into Wnt signaling, and confusing observations on inflammatory signaling. We concur with reviewers that additional observations are needed to adequately support claims that the HFSC compartment is compromised, including details into how renewal and differentiation are impaired. In addition, the present manuscript and rebuttal plan lack important details into how Wnt signaling (canonical or noncanonical) is altered by loss of CerS4, as well as a causal connection linking ceramides/sphingolipids with Wnt signaling. While these issues were touched on by Reviewer 3, a suitable revision should provide mechanistic details beyond the gene expression changes of Wnt effectors noted in the revision plan. Finally, a suitably revised manuscript should document the important changes noted in organoid morphology/cohesion with quantitative data.

Please let us know if you are able to address the major issues outlined above and wish to submit a revised manuscript to JCB. Note that a substantial amount of additional experimental data likely would be needed to satisfactorily address the concerns of the reviewers. The typical timeframe for revisions is three to four months. While most universities and institutes have reopened labs and allowed researchers to begin working at nearly pre-pandemic levels, we at JCB realize that the lingering effects of the COVID-19 pandemic may still be impacting some aspects of your work, including the acquisition of equipment and reagents. Therefore, if you anticipate any difficulties in meeting this aforementioned revision time limit, please contact us and we can work with you to find an appropriate time frame for resubmission. Please note that papers are generally considered through only one revision cycle, so any revised manuscript will likely be either accepted or rejected.

If you choose to revise and resubmit your manuscript, please also attend to the following editorial points. Please direct any editorial questions to the journal office.

GENERAL GUIDELINES:

Text limits: Character count is < 40,000, not including spaces. Count includes title page, abstract, introduction, results, discussion, and acknowledgments. Count does not include materials and methods, figure legends, references, tables, or supplemental legends.

Figures: Your manuscript may have up to 10 main text figures. To avoid delays in production, figures must be prepared according to the policies outlined in our Instructions to Authors, under Data Presentation, <https://jcb.rupress.org/site/misc/ifora.xhtml>. All figures in accepted manuscripts will be screened prior to publication.

Supplemental information: There are strict limits on the allowable amount of supplemental data. Your manuscript may have up to 5 supplemental figures. Up to 10 supplemental videos or flash animations are allowed. A summary of all supplemental material should appear at the end of the Materials and methods section.

Please note that JCB now requires authors to submit Source Data used to generate figures containing gels and Western blots with all revised manuscripts. This Source Data consists of fully uncropped and unprocessed images for each gel/blot displayed in the main and supplemental figures. Since your paper includes cropped gel and/or blot images, please be sure to provide one Source Data file for each figure that contains gels and/or blots along with your revised manuscript files. File names for Source Data figures should be alphanumeric without any spaces or special characters (i.e., SourceDataF#, where F# refers to the associated main figure number or SourceDataFS# for those associated with Supplementary figures). The lanes of the gels/blots should be labeled as they are in the associated figure, the place where cropping was applied should be marked (with a box), and molecular weight/size standards should be labeled wherever possible.

If you choose to resubmit, please include a cover letter addressing the reviewers' comments point by point. Please also highlight all changes in the text of the manuscript.

Regardless of how you choose to proceed, we hope that the comments below will prove constructive as your work progresses. We would be happy to discuss them further once you've had a chance to consider the points raised. You can contact the journal office with any questions at cellbio@rockefeller.edu.

Thank you for thinking of JCB as an appropriate place to publish your work.

Sincerely,

Elaine Fuchs, Ph.D.
Editor
The Journal of Cell Biology

Tim Fessenden
Scientific Editor
Journal of Cell Biology

Revision Plan

Manuscript number: RC-2024-02343

Corresponding author(s): Sara A. Wickström

General Statements [optional]

We thank the reviewers for their thoughtful comments on our manuscript “Sphingolipid metabolism orchestrates the establishment of the adult hair follicle stem cell niche to control skin homeostasis”. We were delighted to see that the reviewers found our work “conceptually novel”, of “broad interest”, “well-written” and “well presented”.

It is important to note that the main finding of this manuscript is that hair follicle stem cells contain a unique ceramide synthase and lipid profile. Perturbing this profile by deleting CerS4 leads to profound defects in stem cell fate regulation through non-canonical and canonical Wnt signaling. This is a new finding that has implications far beyond skin biology and is of general interest as also stated by Reviewer 2. Thus, we emphasize that the goal of this manuscript was not to rebut previous findings on the role of CerS4 in the sebaceous gland, as interpreted by Reviewer 1, but to report new functions of CerS4 in signaling and cell fate regulations. Our understanding of the role of ceramides in stem cells is only beginning to emerge and this work provides a comprehensive, mechanistic study of a specific function of CerS4 in hair follicle stem cells.

In the revised version, we have thoroughly addressed all reviewer comments with a large panel of new experiments and by careful edits of the manuscript. Specifically, we now:

1. Use *in vivo* lineage tracing of hair follicle stem cells to provide conclusive evidence that CerS4 controls hair follicle stem cell fate.
2. Perform additional functional assays to demonstrate that this stem cell fate defect is due to an imbalance between non-canonical and canonical Wnt signaling and that blocking non-canonical Wnt/Ca²⁺ signaling rescues the differentiation defect in CerS4-deficient hair follicle stem cells, providing the molecular mechanism of CerS4 action.
3. As our inhibitor and sequencing data implicated hyperactivity of the calcium arm of non-canonical Wnt, we further analyze calcium signaling in isolated HFSCs and show hyperactive calcium waves in CerS4 knockout cells that can be restored by CerS4 expression. Consistently, we find that CerS4 is localized to the endoplasmic reticulum which is a key compartment for intracellular Ca²⁺ control. This proves additional insight to how CerS4 controls non-canonical Wnt/Ca²⁺ signaling.

Revision Plan

Description of the planned revisions

Reviewer #1 (Evidence, reproducibility and clarity (Required)):

Deletion of CerS4 in the entire mouse epidermis throughout development via the K14-Cre results in enlarged sebaceous glands and perturbed HFSC molecular phenotype. There is low or no expression of CD34, a known marker of the HFSCs along with apparent reduction of several other HFSC markers and acquisition of a more differentiated cell phenotype in these cells. Interestingly, skin and hair follicles seem to remain normal otherwise up to advanced age, though this contradicts the notion that HFSC were indeed affected at the functional level. The data does not demonstrate 'gradual decline' in the HFSC compartment, as claimed by the authors, but rather seem to indicate that the adult HFSC compartment is not properly established in its molecular signatures. Organoid cultures document defects in HFSC, which included reduced proliferation in the CerS4 KO cells. Lipid composition in plasma membranes was also affected by the CerS4 KO. Associated with this, Wnt signal transduction is also affected according to experiments that enhance the strength of wnt signals via a specific small molecular agonist of the pathway. Finally, the authors discover a resemblance of the mouse KO immune-phenotype, with human atopic dermatitis. The study is likely of interest to a specialized readership in skin biology and dermatology and adds to previous studies on CerS4 in skin that erroneously placed its role in the sebaceous gland. [The authors here demonstrate that deletion of CerS4 in the sebaceous glands via SCD3-Cre led to no phenotype, contradicting the previous assessment that CerS4 is important in sebaceous glands.] The study would need to be corrected in a few of its interpretations regarding stem cells to better match the data, as indicated below.

We thank the reviewer for the constructive comments that have helped us to improve the manuscript. In particular, it is clear that we have not been sufficiently clear in the data presentation and description of the results.

Importantly, the CerS4^{epi-/-} mice have a very strong hair follicle phenotype that results in progressive hair loss. Further, the epidermis is not normal as an eczema-like phenotype develops later, after the hair follicle architecture and function has been disrupted. Thus, there are clear functional consequences to the hair follicle and epidermis that arise from the dysfunction of the HFSC compartment. We have edited the manuscript as well as added photodocumentation and additional quantitative analyses of the macroscopic and hair follicle phenotypes to ensure clarity of the phenotypes (new Fig. 1A; Supplementary Fig. 1A-D).

We fully agree with the reviewer that the initial stem cell phenotype is inability to establish the adult hair follicle stem cell niche, as shown by the single cell sequencing data and as also stated in the manuscript title and outlined in the abstract and text. Importantly, some hair follicle stem cells are nevertheless generated but these become gradually depleted as shown by the FACS analyses. We have further added analyses of BrdU incorporation to show loss of quiescence in the stem cell population

Revision Plan

later on in adult mice (new Supplementary Fig. 1E, F). In conclusion, there is a dual phenotype - an inability to first efficiently establish and later to maintain the hair follicle stem cell compartment. We have carefully edited the manuscript to make this more clear.

Finally, we want to emphasize that one of the central new findings of this manuscript is that hair follicle stem cells contain a unique ceramide synthase and lipid profile. Perturbing this profile by deleting CerS4 leads to profound defects in stem cell state establishment and maintenance through an imbalance in non-canonical and canonical Wnt signaling. To strengthen this conclusion, we have added new data showing that inhibiting non-canonical, calcium-dependent Wnt signaling rescues the stem cell differentiation defect in hair follicle stem cell organoids (new Fig. 5C, D, E). Consistent with this, our new data further shows that calcium signaling is hyperactive in CerS4-deficient cells and it can be rescued by restoring CerS4 expression (new Fig. 5G; Supplementary Movie 1). Finally, this HALO-tagged CerS4 localizes within the endoplasmic reticulum of hair follicle stem cells, an important compartment for intracellular calcium homeostasis (new Fig. 5F; Supplementary Movie 1). These analyses reveal a new mechanism by which ceramide synthases regulate stem cell homeostasis that have potential implications far beyond dermatology.

Major revisions:

Fig1B - the data seems to simply show that bulge cells express less or no CD34 and not that 'CerS4^{epi} -/-' mice showed reduced HFSC numbers'; the primary FACS data should be shown somewhere too.

Outer bulge hair follicle stem cells are defined as a population of cells that expresses CD34 and has high levels of integrin- α 6. The quantifications in Fig. 1C (Fig 1B in the previous version) show the results of the FACS analyses where we quantify the relative size of this population among all epidermal cells, not the intensity of the stainings. These results show reduced population size of CD34⁺/integrin- α 6⁺ cells in CerS4^{epi} -/-' epidermis. Importantly, the mean fluorescence intensity of CD34 and integrin- α 6 was not reduced in these CerS4^{epi} -/-' stem cells. We have now also included the original FACS plot data to support this notion and the quantifications (new Fig. 1B).

The conclusion that stemness is affected, and HFSCs lose their normal gene expression signature is at more convincing after looking at other HFSC markers down the road in the paper. However, in the absence of functional assays that would demonstrate stem cell function is lacking and seeing that hair follicles are maintained and grow in long-term, the notion that stem cells are lacking in these conditions is not supported by the data.

We appreciate that the reviewer finds the marker gene analysis convincing. To assay stem cell functionality, we have used the organoid formation assay (widely used assay for stemness). Using this functional assay we observe impaired self-renewal of stem cells (Fig. 3D) and enhanced differentiation (Fig. 3I).

Revision Plan

We have now further added data showing that impaired self-renewal of CerS4^{epi-/-} stem cells results from an increase in non-canonical Wnt/calcium signaling, and inhibiting non-canonical Wnt in organoids rescues the stem cell differentiation defect (new Fig. 5C, D, E), and explaining the *in vivo* phenotypes of the inability to establish and maintain the stem cell population and driving increased stem cell differentiation.

In the revised manuscript we have also included measurements of stem cell self-renewal *in vivo* using BrdU incorporation. While stem cells initially enter quiescence upon initial formation of the stem cell niche (new Supplementary Fig. 1E), the quiescence is not maintained long term (new Supplementary Fig. 1F), indicating that the stem cell phenotype is two-fold involving initial establishment and then long-term maintenance.

We further investigate the long-term maintenance phenotype by tracing cell fate transitions *in vivo* using Lgr5-Cre mediated *in vivo* lineage tracing (new Fig. 2F) and observe abnormal localization of the stem cell progeny along the hair follicle. Collectively this data allows us to conclude that stem cell function is impaired upon epidermal and stem cell intrinsic CerS4-deletion.

The conclusion after figure 2: "Collectively, these data indicate that CerS4-deficiency triggers gradual depletion of the quiescent HFSC compartment." There is no data showing gradual depletion of the quiescent HFSC compartment. We would need to see a gradual activation of HFSCs with over proliferation to conclude this. There is some data albeit not always convincing (see NFAC1 staining in Fig. 5C) indicating loss of markers associated with quiescence but there is no data indicating 'gradual' loss of markers.

We agree with the reviewer that showing gradual activation of the HFSCs *in vivo* is important to conclude loss of quiescence. As already indicated in the response to the previous point, we have now included *in situ* stainings of *in vivo* BrdU incorporation and quantify proliferation in the hair follicle bulge stem cell region. Data from P47 mice shows a clear increase in BrdU incorporation in the stem cell compartment of CerS4^{epi-/-} skin (new Supplementary Fig. 1F). In addition, we have performed lineaged tracing of hair follicle stem cell progeny and observe that they are abnormally localized along the upper hair follicle (new Fig. 2F), collectively explaining the stem cell loss. Finally, the Nfatc1 data has been removed.

Minor revisions:

Figure 1C legend - please spell out what are the abbreviations for the different subpopulations; please show these populations as % as opposed to absolute numbers.

We have edited the Fig. 1D (Fig. 1C in the previous version) labeling and legend for clarity and also indicate the populations as %.

Revision Plan

Figure 1D - please make it clear in the cartoon what the different sub-populations listed are;

We have edited the cartoon for clarity.

Is OB1 a CD34- HFSC population?

The outer bulge 1 (OB1) is a population of cells that expresses hair follicle stem cell markers, including CD34. We have clarified this in the schematic figure 1E.

Fig. 3B - the colors of the legend do not match the colors in the data so it is confusing as to which one is which!

We have altered the colors to match the data and thank the reviewer for pointing this out.

Fig 5C - the differences in NFATc1 are not visible in the images shown

We apologize for the suboptimal quality of these images. We have replaced this data with data on non-canonical Wnt inhibition.

Reviewer #2 (Evidence, reproducibility and clarity (Required)):

The manuscript authored by Peters et al. titled "Sphingolipid metabolism orchestrates the establishment of the adult hair follicle stem cell niche to control skin homeostasis" elucidates the critical role of ceramide synthase 4 (CerS4) in the epidermal stem cell niche, particularly in regulating hair follicle bulge stem cells (HFSCs). Using epidermal specific CerS4 knockout mice as an in vivo model and hair follicle organoid culture as an ex vivo model, the authors conducted a comprehensive analysis, which includes cutting edge approaches such as scRNA-seq, proteomics, and lipidomics. The results highlight CerS4's function in the establishment/maintenance of the HFSC niche, as absence of CerS4 changes HFSCs' number and differentiation state. Potential underlying mechanisms identified include altered membrane lipid profiles and Wnt signaling responsiveness. Possible link to a chronic inflammatory skin disease, atopic dermatitis, is also implicated. The data presented are generally of high quality, and the work is significant as it uncovers a new regulator of HFSC fate with mechanistic connection to lipid metabolism.

We thank the reviewer for the positive assessment of our work and finding it to be of high quality and significance. We further appreciate the constructive comments that have helped us to improve the manuscript.

However, some issues were identified, most of them having to do with in vivo characterization and data interpretation:

Revision Plan

Major:

1. The *in vivo* HFSC phenotype can be better characterized. "Collective, these data show that *CerS4* in HFSCs is essential to establish the adult stem cell compartment and to assure lineage fidelity." - this statement premature based on order of the data shown. Also the trajectory difference shown in Figure 2A is not striking. Subclustering out the relative cell subsets and redo the analysis might help to tease out the difference. Additional experiments such as lineage tracing would be useful to support the notion that there is lineage fidelity issue in the mutant - though it is understood that this is quite involved and may lie outside the scope of the current study. Are bulge cells in the mutant proliferative? - the authors should consider *in vivo* Edu labelling experiment or the like to assess the quiescence/proliferation of the bulge cells. Finally, analyzing hair follicles at earlier stages might help to clarify when and where the bulge and sebaceous gland changes start - is possible that aberrant divergence of bulge/sebaceous fates occur prior to the establishment of a stable bulge fate?

We thank the reviewer for these expert suggestions, we have taken several steps to address these points:

1. We have included additional analyses of the single cell sequencing data. As suggested, we have now further examined the predicted differences in the trajectories of the relevant hair follicle cell subpopulations using PAGA to more clearly demonstrate the altered lineage trajectories (new Fig 2B). In addition, we have compared the control and *CerS4^{epi/-}* outer bulge stem cell signatures to published gene expression profiles from Joost et al., 2016 and find an over-representation of genes expressed in sebocytes in *CerS4^{epi/-}* outer bulge stem cells. In addition, altered gene signatures of the upper hair follicle basal and upper hair follicle suprabasal cells characterize control and *CerS4^{epi/-}* outer bulge stem cells (new Supplementary Fig. 2K). Collectively these additional analyses support the conclusions of altered lineage identities.

2. To conclusively demonstrate altered lineage trajectories *in vivo*, we have included lineage tracing experiments using the stem cell-specific *Lgr5-Cre* to demonstrate that *CerS4*-deletion leads to altered routing of hair follicle stem cells into upper hair follicle and sebaceous gland fates (new Fig. 2F), confirming the predictions for the sequencing data analyses. Importantly, the hair follicle stem cells cultured in the *in vitro* organoids show abnormal differentiation (Fig. 3H, I, Supplementary Fig. 3G), closely resembling the *in vivo* phenotypes, thereby strengthening the conclusion of cell-autonomously altered lineage trajectories of the *CerS4^{epi/-}* hair follicle stem cells.

3. As suggested, we have performed *in vivo* BrdU incorporation assays to address the impact of *CerS4* loss on long term loss of quiescence in the stem cell compartment in *CerS4^{epi/-}* mice. This new data shows increased proliferation in the bulge stem cell region of *CerS4^{epi/-}* mice in older mice (P47) (new Supplementary Fig. 1E, F).

4. Finally, we have traced the onset of the hair follicle and sebaceous gland phenotypes by analyzing earlier timepoints. A delay in entry into the first postnatal telogen becomes clearly visible at P16, and

Revision Plan

this is associated with abnormal marker expression (reduced Lhx2, increased Krt6) and sebaceous gland enlargement (new Fig. 1A; Supplementary Fig. 1A-C).

2. *The exclusion of IFE contribution is not backed up by data. Figure 6D - model emphasizes HFSC involvement in atopic dermatitis, but this could be due to epidermal barrier defect. Barrier defect could already be present even though IFE morphology appears normal. Maybe TEWL is measured at the time of analysis and shows no change - if so, this data should be included. HFSC changes might contribute but the involvement of IFE cannot be excluded. The conclusion that "CerS4 expression was restricted to the hair follicle" is not supported by data. IFE expression is apparent in Figure SIC. Along this line, there is also an apparent expansion of IFE basal II in the mutant (Figure 1C).*

We acknowledge that we have not been clear enough with the evidence that allowed us to exclude the involvement of an IFE-mediated barrier defect in the early eczema-like phenotype. To address a potential barrier defect early on, we had previously performed careful analysis of TEWL. In Peters et al., 2020 we demonstrate no changes in TEWL at P0, a reduced TEWL at P21 and an increased TEWL in adult CerS4^{epi-/-} mice starting only at P33. The reduction of the TEWL in adolescent CerS4^{epi-/-} mice (P21) is likely linked to an increased production of sebaceous lipids lubricating the skin surface at this time point (Peters et al., 2020). Thus, defects in the hair follicle stem cell compartment, present at adolescence (P21) arise prior to defects in the adult (P33) IFE barrier function.

Cers4 expression is overall low in skin, as is typical for enzymes. *In situ* stainings of Cers4mRNA in Supplementary Fig. 2C (Supplementary Fig. 1C in the previous version) indeed show a sparse signal also in the IFE. However this signal is also present in CerS4^{-/-} sections indicative of unspecific labeling, although the KO skin cannot be conclusively used to control background as these mice were generated by deletion of exon 3 only, and Cers4 RNAscope probes might detect truncated Cers4 mRNA in these mice (hence we did not include this control in the manuscript). Importantly, our data on FACS sorted basal cells of the IFE shows no substantial Cers4 mRNA expression in IFE progenitors (Supplementary Fig. 2D), consistent with the scRNAseq data (Supplementary Fig. 2B). Thus, while we cannot fully exclude low levels of CerS4 expression in the IFE, the levels are substantially lower than in the HFSC and SG compartments, and the phenotype, including the slight expansion of the IFE basal II population, is less profound compared to the hair follicle phenotype. However, to avoid overinterpreting our data, we have edited the conclusions to be less strong on the expression of the IFE.

3. *Figure 1 - single cell analysis was done using only 2 pairs of mice, and data in E lack statistical assessment. At the very least, data for individual pairs should be shown in supplemental data to ensure that changes are consistent in both mutant mice rather than being dominated by dramatic alteration in only 1 mutant mouse.*

We had rigorously analyzed the replicates to ensure that the phenotype is consistently present in both replicates prior to merging the data. We have now included this separate analysis of the mice to

Revision Plan

document robustness of the differences (new Supplementary Fig. 2J). Importantly, we also validate key changes on the protein level (Lhx2, Krt6) in independent mouse cohorts (Fig. 2D, E; Supplementary Fig. 1 B,C), strengthening this conclusion.

Minor:

1. *CerS4SCD3-/+ nomenclature is mis-leading.*

We have changes this to CerS4^{SG-/-} for clarity.

2. *Figure 2- "Furthermore, we observed expansion of the inner bulge identity marker Krt6 protein expression into outer bulge stem cells and along the infundibulum in CerS4epi-/- hair follicles, whereas in control mice Krt6 was restricted to the inner bulge (Fig. 2C)." - Krt6 staining is presented in Fig 2D, not 2C.*

We thank the reviewer for pointing out this mistake that we have corrected (Fig. 2E in the revised manuscript).

3. *Figure 3C - size of the organoids should be quantified with statistics. The images shown do not support the statement that "Strikingly, CerS4epi-/- organoids showed altered morphology characterized by smaller size and loss of cohesion of peripheral cells from the organoid clusters (Fig. 3C), ...".*

We have included better images and quantifications (new Fig. 3C).

4. *Section titled "CerS4 regulates HFSC differentiation in a stem cell autonomous manner": "CD34-integrin- $\alpha 6$ + cells, which based on extensive transcriptome and marker expression analyses represent a mixture of HFSCs, hair follicle outer root sheath (ORS) cells and inner bulge cells (collectively termed non-HFSCs)." - shouldn't the CD34- integrin- $\alpha 6$ + population also contain IFE stem/progenitor cells? Are hair follicles micro-dissected out for FACS?*

The hair follicles are not micro-dissected out for FACS, and the entire basal cell population is initially isolated. However, the organoid culture conditions specifically promote the expansion of the hair follicle lineage, whereas cells of the IFE are not expanded and long term maintained as extensively documented in previous publications using this organoid system (see for example Kim et al., Cell Metabolism 2020; Chacon-Martinez EMBOJ 2016).

5. *Figure 5D - please provide the working concentration of Chir99021.*

We have provided the working concentration of Chir99021 (1 μ M)

Figure 5F - explain what arrows mean in legends.

Revision Plan

We have defined the arrows, (Supplementary Fig. 5D in the revised manuscript).

6. *Figure 6A - no significant changes in Th2 and ILC2 were observed at a 95% confidence interval. Increasing mouse number will help to increase statistical power.*

We agree with the reviewer and acknowledge that this experiment was unfortunately underpowered. We have repeated the experiment with a larger cohort and have included CD4⁺, TH17⁺ and ILC3 cells. The subset of CD25⁺ ILC2 cells now needed to be excluded as the population was too small for further analysis. Note the dominance of a Th2 and ILC2 immune signature, a hallmark of an AD-like eczema in CerS4^{epi-/-} skin, where no general inflammation or a shift to a Th17 response, characteristic for a psoriatic-immune response was detected (new Fig. 6A).

7. *Additional Wnt target genes such as Axin 2 should be looked at.*

We thank the reviewer for this suggestion. We have now performed additional gene expression analyses of the outer bulge cells, including analyses of non-canonical and canonical Wnt target genes. While a strong signature of non-canonical Wnt targets is observed, as also predicted by the receptor-ligand interactions, we do not observe major differences in Axin2 or CyclinD1 (new Supplementary Fig. 5B). A minor increase in the canonical Wnt targets Lef1 and Tcf4 expression were detected in CerS4^{epi-/-} outer bulge stem cells (new Supplementary Fig. 5C). Thus, together with the new functional data in the organoids that show rescue of the stem cell differentiation phenotype with non-canonical Wnt inhibition (new Fig. 5C-E), this indicates that the strongest defect is in non-canonical Wnt rather than canonical Wnt. We propose that this increase in non-canonical Wnt/calcium signaling primes CerS4^{epi-/-} stem cells in the organoid cultures to be susceptible to canonical Wnt signaling, as shown by the addition of Chir99021, inducing canonical Wnt driven fate transitions in CerS4^{epi-/-} organoids (Supplementary Fig. 6A, B).

8. *The increased BMP signaling and decreased Nfatc1 expression are seemingly contradictory.*

We apologize for the lack of clarity here. Receptor ligand analyses from single cell sequencing predicted increased BMP-signaling from outer bulge towards the inner bulge cell compartment (Supplementary Fig. 5A). No alteration in BMP signaling was predicted within the outer bulge stem cell compartment (Fig. 5A), where decreased NFATc1 expression was observed. We have removed the data on NFATc1 as we have now focus on non-canonical Wnt signaling.

9. *Paragraph starting with "It is interesting to note that ceramide availability was shown to regulate Wnt signaling in Drosophila through strong effects on recycling endocytosis of the receptors (Pepperl et al., 2013)." Is redundant in the manuscript.*

Revision Plan

We apologize for the accidental duplication of this paragraph and thank the reviewer for noticing this mistake. We have thoroughly edited the discussion to accommodate the new finding of the revised manuscript.

Reviewer #3 (Evidence, reproducibility and clarity (Required)):

The authors created CerS4 mutant mice to test the role of sphingolipids in hair follicle stem cells (HFSCs) and the hair cycle. This work extends previous studies that show that loss of this enzyme leads to defects in the hair cycle and eventually hair loss. In this study the authors look early on in the course of the deletion in an attempt to understand why loss of this enzyme leads to the phenotype described previously. They use single cell profiling, proteomics, and in situ imaging to pinpoint issues in the stem cell niche that drive phenotypes and propose novel interactions between sphingolipid metabolism, Wnt signaling, and inflammation in regulation of HFSC homeostasis. The data are nicely presented, and the text is well written. The conclusions are clearly defined.

We thank the reviewer for the positive assessment of our work and finding it well written and presented. We further appreciate the constructive comments that have further help us to improve the manuscript.

While the data are clearly presented, there are numerous issues that are confusing to this reviewer. In addition, some of the phenotypes described are subtle, and thus do not make a convincing case.

1, In figure 1C, the cell proportion analysis suggests there are no OBII or SG in WT. I am not sure how this could be possible. In addition, there appears to be almost no sebaceous cells in either, but the mutant supposedly has much larger sebaceous glands (in Fig 2). In Fig S11, there is no change in bulge cells? In Fig 1B, there is less HFSCs in the mutant than in the WT, but in 1C, there is more OBI in the mutant. The results in Fig 1B and C are confusing. Also, the schematic in Fig 1 is hard to read, the authors should color code the text with the image.

It is important to emphasize that the single cell RNA sequencing was carried out at P19 when the bulge stem cell compartment only starts to be established. This explains why only few bulge stem cells are detected at this point. Nevertheless, the OBII and SG clusters are visible also in the control (Fig. 1E in the revised manuscript). We have now included additional subclustering of the relevant subpopulations to make these populations more clearly visible also in the control (new Figure 1D). We have also edited the labels for clarity.

Mature sebocytes are very large cells, and inherent to the single cell sequencing workflows these large cells are excluded from the sequencing libraries. Importantly, we do not detect a change in bulge stem cells in a mouse line in which CerS4 was specifically deleted only in sebocytes (Supplementary Fig. 2E-I), to exclude that hair follicle phenotypes are secondary to sebaceous gland defects. Thus, this data

Revision Plan

does not contradict Fig. 1, as data in Fig. 1 was generated using a different mouse line in which CerS4 is deleted in the entire epidermal stem cell population using Keratin14-Cre. We have edited the manuscript to make this more clear.

Finally, data presented in Fig 1B and E focus on two different aspects. Data in Fig. 1B, C show the inefficient establishment and maintenance of CD34⁺/integrin- α 6⁺ bulge hair follicle stem cells. The quantification is based on FACS analyses of cells expressing these cell surface molecules/stem cell markers. Fig. 1E (Fig. 1 C in the previous version) shows the quantification of the various cell states based on single cell RNA expression and subsequent clustering of the control and CerS4^{epi-/-} epidermal cells together. The “outer bulge” cluster was annotated based on these cells expressing hair follicle stem cell markers. While the CerS4^{epi-/-} epidermis shows increased number of cells in this cluster, the expression of all key stem cell genes (CD34, Sox9, Krt15, Lhx2) is reduced in CerS4^{epi-/-} mice in this outer bulge 1 compartment compared to control. Thus, while this “outer bulge” population is expanded in the KO, the stem cell properties of this population are clearly attenuated, as defined by decreased expression of key stem cell transcription factors and increased expression of differentiation genes (see also new Supplementary Fig. 2K).

2, In Figure 5, the signaling chart shows a strong upregulation of non-canonical Wnt signaling in the mutant bulge. Canonical Wnt signaling appears to be unchanged between wt and ko. Thus, it is not clear why the authors came to the conclusion that Wnt signaling is induced in the mutant. They further show expression of Lef1 and Nfatc1, but these are not typical markers used to denote canonical wnt activation, as implied. In fact, the data in Fig S4B suggest the induction of Lef1 and Tcf4 is actually very subtle. Instead, the authors should use nuclear b-catenin or transcriptional targets such as Axin or CyclinD. The authors should in fact explore the observation of Wnt5, as that appears to be the most dramatic change. In addition, the authors should use an ontological analysis with the single cell data from the tissue in the same manner that they did for organoids to take another look at molecular consequences of loss of CerS4.

We greatly appreciate this criticism and agree that non-canonical Wnt shows stronger activation in the outer bulge communication (Fig. 5A), and non-canonical Wnt was also predicted to be enriched in communication between outer and inner bulge (Supplementary Fig. 5A). Thus, as suggested, we have now further analyzed non-canonical Wnt signaling downstream of Wnt5a.

To this end we have carried out differential gene expression analyses of the outer bulge compartment, and GO term analyses indeed implicate target genes of the non-canonical Wnt pathway are upregulated in CerS4-deficient mice (new Fig. 5B).

To test if an increase in non-canonical Wnt/calcium signaling or in the Wnt/JNK pathway would drive the impaired stem cell state establishment we inhibited these pathways via small molecule inhibitors KN95 (CamKII inhibitor), and SP600125 (JNK inhibitor) to the organoids. Inhibiting CamKII signaling restored HFSC numbers in CerS4^{epi-/-} organoids. However, inhibiting the non-canonical Wnt/PCP pathway was unable to rescue the stem cell reduction (new Fig. 5C-E). The data indicate that

Revision Plan

CerS4 is essential to adjust levels of non-canonical Wnt/calcium signaling to ensure hair follicle stem cell state identity. We propose that this increase in non-canonical Wnt/calcium signaling primes CerS4^{epi/-} stem cells in the organoid cultures to be susceptible to canonical Wnt signaling, as shown by the addition of Chir99021, inducing canonical Wnt driven fate transitions in CerS4^{epi/-} organoids (Supplementary Fig. 6).

3, The authors suggest that much of the phenotype is due to inflammation. In Fig 6A, they showed analysis of CD45 cells in the skin. However, the only change was a very subtle change in Th2 cells, while no other CD45+ cells were altered.

We agree with the reviewer and acknowledge that this experiment was unfortunately underpowered. We have repeated it with a larger cohort and have included CD4⁺, TH17⁺ and ILC3 cells. The subset of CD25⁺ ILC2 cells now needed to be excluded as the population was too small for further analysis. Note the dominance of a Th2 and ILC2 immune signature, a hallmark of an AD-like eczema in CerS4^{epi/-} skin, where no general inflammation or a shift to a Th17 response, characteristic for a psoriatic-immune response was detected (new Fig. 6A).

4, The authors showed upregulation of Immune response in Fig 6C, but then in Fig S2, the genes downregulated are also related to immune response...how do the authors reconcile this?

We apologize for this confusion. Importantly, keratinocyte-intrinsic downregulation of homeostatic immune modulating activity is a key driver of inflammatory skin barrier disease, like atopic dermatitis. This barrier intrinsic immune modulation is distinct from immune cell-mediated inflammation. There is a strong overlap of genes constituting the term “Inflammatory abnormality of the skin” (Human phenotype ontology terms) in Fig 6C and “Immune system process” (GOBP terms) (Supplementary Fig. 3F; Fig S2F in the previous version). To name some, i.e. Adam-, ALOX-, ASXL- family members are annotated by both terms. Mutations in these genes are known to cause skin diseases associated with immune dysregulation but are likewise known to regulate immune responses.

Data in Figure 6C shows enrichment of this term from both up and downregulated proteins in the CerS4^{epi/-} condition compared to control, indicating that proteins involved in “Inflammatory abnormality of the skin” are dysregulated in CerS4^{epi/-} organoids. Data in Supplementary Fig. 3F (Figure S2F in the previous version) shows the downregulation of these proteins in CerS4^{epi/-} organoids compared to control. We have clarified this in the text and figure legends.

5, The author propose that the phenotype in CerS4 null mice is due to disruption of the stem cell Niche. However, the authors have not shown evidence for such an effect through any in situ analysis. The single cell approaches are valuable, but in that case the niche is dissociated. The organoid work is also nice, but not exactly a stem cell niche either. The authors should instead test their hypothesis through an in situ analysis.

Revision Plan

We have used the term niche to describe the bulge compartment of the hair follicle, including the cellular interactions between stem cells and the other niche resident cells such as the Krt6⁺ inner bulge cells that have been analyzed here. This is consistent with the terminology widely used in the field (see for example references Blanpain & Fuchs, 2014; Chacon-Martinez *et al*, 2017; Hsu *et al*, 2011).

Importantly, we have now included additional *in situ* analyses to strengthen the conclusions on stem cell niche alterations:

1. We have performed *in vivo* lineage tracing analyses using Lgr5-Cre to show altered lineage progression of hair follicle stem cells into the upper hair follicle (new Fig. 2F)
2. We have analyzed BrdU incorporation *in vivo* to show gradual loss of quiescence of the hair follicle stem cells (new Supplementary Fig. 1E, F). Collectively we show that alterations in stem cell fate within the hair follicle bulge niche leads to gradual depletion of stem cells at this site and eventually severe alterations in the niche structure itself (new Supplementary Fig. 1F).

Description of the revisions that have already been incorporated in the transferred manuscript

We have incorporated revisions.

Description of analyses that authors prefer not to carry out

We have carried out all suggested analyses.

December 23, 2024

RE: JCB Manuscript #202403083R

Sara Wickström
Max Planck Society

Dear Prof. Wickström:

Thank you for submitting your revised manuscript entitled "Sphingolipid metabolism orchestrates establishment of the hair follicle stem cell niche". Thank you for your patience while we considered a decision, which was delayed due to editor availability. We would be happy to publish your paper in JCB pending the resolution of remaining reviewer concerns, as well as final revisions necessary to meet our formatting guidelines (see details below).

As you will see, reviewers are overall satisfied with the substantial improvements included in this revised manuscript. Reviewer 1 observed that HFSC numbers appeared inconsistent, which may agree with conclusions on altered lineage identity and differentiation but should be directly addressed in the text. In addition, Reviewers 1 and 3 expressed concern over the broader conclusions regarding the hair follicle niche itself vs the reported changes specific to HFSCs. Both comments suggest that terminology should be carefully deployed to describe and interpret only the observations made concerning stem cell lineage identity and differentiation. In revising terminology, the text should draw as clear a model as possible, in response to this final point by Reviewer 1. Resolving this issue may require changes to the title and abstract in addition to the main text. Finally please carefully consider the remaining points made by reviewers as you revise this work.

A. MANUSCRIPT ORGANIZATION AND FORMATTING:

Full guidelines are available on our Instructions for Authors page, <http://jcb.rupress.org/submission-guidelines#revised>. Submission of a paper that does not conform to JCB guidelines will delay the acceptance of your manuscript.

1) Text limits: Character count for Articles is < 40,000, not including spaces. Count includes abstract, introduction, results, discussion, and acknowledgments. Count does not include title page, figure legends, materials and methods, references, tables, or supplemental legends.

2) Figures limits: Articles may have up to 10 main figures and 5 supplemental figures/tables.

3) Figure formatting: Scale bars must be present on all microscopy images, including inset magnifications. Molecular weight or nucleic acid size markers must be included on all gel electrophoresis. Please avoid pairing red and green for images and graphs to ensure legibility for color-blind readers. If red and green are paired for images, please ensure that the particular red and green hues used in micrographs are distinctive with any of the colorblind types. If not, please modify colors accordingly or provide separate images of the individual channels.

4) Statistical analysis: Error bars on graphic representations of numerical data must be clearly described in the figure legend. The number of independent data points (n) represented in a graph must be indicated in the legend. Statistical methods should be explained in full in the materials and methods. For figures presenting pooled data the statistical measure should be defined in the figure legends. Please also be sure to indicate the statistical tests used in each of your experiments (either in the figure legend itself or in a separate methods section) as well as the parameters of the test (for example, if you ran a t-test, please indicate if it was one- or two-sided, etc.). Also, if you used parametric tests, please indicate if the data distribution was tested for normality (and if so, how). If not, you must state something to the effect that "Data distribution was assumed to be normal but this was not formally tested."

5) Abstract and title: The abstract should be no longer than 160 words and should communicate the significance of the paper for a general audience. The title should be less than 100 characters including spaces. Make the title concise but accessible to a general readership.

** Per Reviewer 3, please ensure the title uses terminology that most closely aligns with the data shown, for instance: "Sphingolipid metabolism orchestrates establishment of the hair follicle stem cell compartment"

6) Materials and methods: Should be comprehensive and not simply reference a previous publication for details on how an experiment was performed. Please provide full descriptions in the text for readers who may not have access to referenced manuscripts. We also provide a report from SciScore and an associate score, which we encourage you to use as a means of evaluating and improving the methods section.

7) Please be sure to provide the sequences for all of your primers/oligos, plasmids, and RNAi constructs in the materials and methods. You must also indicate in the methods the source, species, and catalog numbers (where appropriate) for all of your antibodies. Please also indicate the acquisition and quantification methods for immunoblotting/western blots.

8) Microscope image acquisition: The following information must be provided about the acquisition and processing of images:

- a. Make and model of microscope
- b. Type, magnification, and numerical aperture of the objective lenses
- c. Temperature
- d. Imaging medium
- e. Fluorochromes
- f. Camera make and model
- g. Acquisition software
- h. Any software used for image processing subsequent to data acquisition. Please include details and types of operations involved (e.g., type of deconvolution, 3D reconstitutions, surface or volume rendering, gamma adjustments, etc.).

10) Supplemental materials: There are strict limits on the allowable amount of supplemental data. Articles may have up to 5 supplemental figures. Please also note that tables, like figures, should be provided as individual, editable files. A summary of all supplemental material should appear at the end of the Materials and methods section.

13) ORCID IDs: ORCID IDs are unique identifiers allowing researchers to create a record of their various scholarly contributions in a single place. At resubmission of your final files, please provide an ORCID ID for all authors.

15) A data availability statement is required for all research article submissions. The statement should address all data underlying the research presented in the manuscript. Please visit the JCB instructions for authors for guidelines and examples of statements at (<https://rupress.org/jcb/pages/editorial-policies#data-availability-statement>).

Please note that JCB requires authors to submit Source Data used to generate figures containing gels and Western blots with all revised manuscripts. This Source Data consists of fully uncropped and unprocessed images for each gel/blot displayed in the main and supplemental figures. Since your paper includes cropped gel and/or blot images, please be sure to provide one Source Data file for each figure that contains gels and/or blots along with your revised manuscript files. File names for Source Data figures should be alphanumeric without any spaces or special characters (i.e., SourceDataF#, where F# refers to the associated main figure number or SourceDataFS# for those associated with Supplementary figures). The lanes of the gels/blots should be labeled as they are in the associated figure, the place where cropping was applied should be marked (with a box), and molecular weight/size standards should be labeled wherever possible. Source Data files will be directly linked to specific figures in the published article.

WHEN APPROPRIATE: The source code for all custom computational methods published in JCB must be made freely available as supplemental material hosted at www.jcb.org. Please contact the JCB Editorial Office to find out how to submit your custom macros, code for custom algorithms, etc. Generally, these are provided as raw code in a .txt file or as other file types in a .zip file. Please also include a one-sentence summary of each file in the Online Supplemental Material paragraph of your manuscript.

Journal of Cell Biology now requires a data availability statement for all research article submissions. These statements will be published in the article directly above the Acknowledgments. The statement should address all data underlying the research presented in the manuscript. Please visit the JCB instructions for authors for guidelines and examples of statements at (<https://rupress.org/jcb/pages/editorial-policies#data-availability-statement>).

B. FINAL FILES:

Thank you for your attention to these final processing requirements. Please revise and format the manuscript and upload materials within 7 days. If you need an extension for whatever reason, please let us know and we can work with you to determine a suitable revision period.

Thank you for this interesting contribution, we look forward to publishing your paper in Journal of Cell Biology.

Sincerely,

Elaine Fuchs, Ph.D.
Editor
The Journal of Cell Biology

Tim Fessenden
Scientific Editor
Journal of Cell Biology

Reviewer #1 (Comments to the Authors (Required)):

The authors have made substantial improvements to the manuscript in response to reviewer concerns. However I still have questions about the interpretation of the data as now presented.

1. In figure 1 C, there is fewer HFSCs in the KO. In 1D there are more HFSCs in the outer bulge I population. In 1E again the UMAP clearly shows more in the KO. Then in 2D, there is fewer stem cells in the KO, and 2F the images shows more HFSCs/germ in the KO, but the same number of cells in the KO according to quantification of the "lower HF". In S1B, there are fewer HFSCs (LHX2) in the KO.

2. The authors attempted to clarify the issues with canonical vs non-canonical wnt. It is worth noting that BMP looks quite induced in KO, and this pathway has a well-established role in inhibiting HFSC activation. This could have been a simpler explanation as the role for non-canonical WNT was not clarified here in vivo.

3. The authors talk about the structural integrity of the niche, but there is not much data to describe this phenotype or explanation for why the hair is lost. Is the ECM around the follicle affected? Is the dermal sheath affected? Is vasculature or lymphatics affected? The physical shape of the follicle and niche do not seem to be altered. I don't see evidence for a disrupted niche, nor a strong explanation for why the hair is lost. Are there fewer follicles? Or fewer shafts? A disrupted hair cycle? In all images at every age the hair cycle seems synchronized in wt and KO.

Reviewer #2 (Comments to the Authors (Required)):

The authors of this revised manuscript titled "Sphingolipid metabolism orchestrates the establishment of the adult hair follicle stem cell niche to control skin homeostasis" have done a good job to address previous review concerns (including mine). The lineage tracing and bulge cell quantification/proliferation data are powerful additions to support a role of CerS4 in HFSC establishment and maintenance. New data on non-canonical Wnt/Ca²⁺ signaling also provide mechanistic insights and lay a strong foundation for future investigation along this direction.

A few very minor issues were noted:

1. Line 60: "While the basal epidermal stem cells drive self-renewal of the interfollicular epidermis (IFE), the hair follicle and sebaceous glands are maintained by their own specific stem cells, the hair follicle stem cells (HFSCs) residing within the bulge stem cell niche" - this statement does not take into consideration of the multiple possible stem cell origins for sebaceous glands (see PMID: 33599012).
2. K15 staining in Supplementary Fig 1 does not appear specific (even dermal cells show scattered positivity), though this has no impact on the main conclusions of the paper.
3. Figure 5G - is it possible to perform statistical analysis of this data, rather than pooling all cells from 3 independent experiments?
4. Line 514: Lgr5eGFP-CreERT2 and Lgr5Cre seem to be used interchangeably, which is somewhat confusing. Details of the lineage tracing experiments should be added to Methods (e.g., tamoxifen treatment).

Reviewer #3 (Comments to the Authors (Required)):

The authors addressed the majority of my concerns and I believe this paper is very close to ready for publications. I really liked the going back and forth between the ex vivo and the in vivo situation that nicely complemented each other. However, I do still have an important editorial criticism. As also pointed out by another reviewer, referring to the hair follicle stem cells in the bulge as the "niche" is incorrect and is both confusing and misleading. While it is true that the bulge is the residence of the HFSC, and it has been referred to as a "niche" in some older studies, we also now know that the niche of the HFSCs is composed of multiple compartments surrounding the bulge stem cells. The niche is now known to be a complex microenvironment that surrounds the bulge. Calling the bulge stem cells themselves a "niche" is especially confusing in the context of this paper, as the phenotypes described here are stem cell autonomous, or cell-intrinsic, as rightly interpreted and demonstrated by the authors. Therefore the authors should simply refer to the bulge stem cell population as the bulge HFSC "compartment" and not as the stem cell "niche". The latter term should be reserved for the many cells and structures surrounding the stem cells that signal to the bulge stem cells. Those do not seem to be affected here or there is no data analyzing that aspect. This correction should include the abstract and the title and all throughout the paper and the discussion.

Point-by-point response to reviewers

Reviewer #1

The authors have made substantial improvements to the manuscript in response to reviewer concerns. However I still have questions about the interpretation of the data as now presented.

We thank the reviewer for this positive assessment of our revised manuscript and the additional comments that we have now addressed.

1. In figure 1 C, there is fewer HFSCs in the KO. In 1D there are more HFSCs in the outer bulge I population. In 1E again the UMAP clearly shows more in the KO. Then in 2D, there is fewer stem cells in the KO, and 2F the images shows more HFSCs/germ in the KO, but the same number of cells in the KO according to quantification of the "lower HF". In S1B, there are fewer HFSCs (LHX2) in the KO.

We thank the reviewer for the careful examination of the data. It is important to emphasize that the outer bulge population in Fig. 1C and E is a population of cells defined by transcriptional similarity. While this population is indeed increased in the KO, these cells show reduced transcriptional features of key stemcell identity genes compared to WT (Fig. 1F), fully consistent with the conclusion of altered lineage progression, resulting in decreased numbers and functionality of bona fide stem cells as defined by FACS analyses of surface protein markers, immunofluorescence analyses of increased expression of non-lineage proteins and functional analyses in the organoids. We have edited the manuscript for clarity.

2. The authors attempted to clarify the issues with canonical vs non-canonical wnt. It is worth noting that BMP looks quite induced in KO, and this pathway has a well-established role in inhibiting HFSC activation. This could have been a simpler explanation as the role for non-canonical WNT was not clarified here in vivo.

While we agree that some components of the BMP pathway are upregulated, it did not appear as a strong candidate in the KEGG pathway analyses of upregulated signaling in the outer bulge compartment in the KO (Fig. 5B). Also the changes appear less substantial than for Wnt 5A (Fig. 5A). This is why we chose to focus on the Wnt pathway, which in the functional analyses appear as a major culprit for the phenotype, but agree that the current experiments do not fully exclude some role for BMP signaling.

3. The authors talk about the structural integrity of the niche, but there is not much data to describe this phenotype or explanation for why the hair is lost. Is the ECM around the follicle affect? Is the dermal sheath affected? Is vasculature or lymphatics affected? The physical shape of the follicle and niche do not seem to be altered. I don't see evidence for a disrupted niche, nor a strong explanation for why the hair is lost. Are there fewer follicles? Or fewer shafts? A disrupted hair cycle? In all images at every age the hair cycle seems synchronized in wt and KO.

We acknowledge, also reflecting on the comment from Reviewer 3, that the use of the term “niche” generated confusion and is not appropriate in the context of our findings. We have replaced the term “niche” with “compartment” throughout the manuscript to underline that we are examining cell autonomous effects in this study.

Reviewer #2 (Comments to the Authors (Required)):

The authors of this revised manuscript titled "Sphingolipid metabolism orchestrates the establishment of the adult hair follicle stem cell niche to control skin homeostasis" have done a good job to address previous review concerns (including mine). The lineage tracing and bulge cell quantification/proliferation data are powerful additions to support a role of CerS4 in HFSC establishment and maintenance. New data on non-canonical Wnt/Ca²⁺ signaling also provide mechanistic insights and lay a strong foundation for future investigation along this direction.

We thank the reviewer for this positive assessment of our revised manuscript.

A few very minor issues were noted:

1. Line 60: "While the basal epidermal stem cells drive self-renewal of the interfollicular epidermis (IFE), the hair follicle and sebaceous glands are maintained by their own specific stem cells, the hair follicle stem cells (HFSCs) residing within the bulge stem cell niche" - this statement does not take into consideration of the multiple possible stem cell origins for sebaceous glands (see PMID: 33599012).

We have now modified this statement to: "While the basal epidermal stem cells drive self-renewal of the interfollicular epidermis (IFE), the hair follicle and sebaceous glands are maintained by distinct stem cells, the hair follicle stem cells (HFSCs) residing within the bulge stem cell niche." and included the suggested reference.

2. K15 staining in Supplementary Fig 1 does not appear specific (even dermal cells show scattered positivity), though this has no impact on the main conclusions of the paper.

We agree that this staining is not optimal and have removed this counterstain from the panel.

3. Figure 5G - is it possible to perform statistical analysis of this data, rather than pooling all cells from 3 independent experiments?

We have replaced the analyses with means from the 3 independent experiments and performed statistical analyses.

4. Line 514: Lgr5eGFP-CreERT2 and Lgr5Cre seem to be used interchangeably, which is somewhat confusing. Details of the lineage tracing experiments should be added to Methods (e.g., tamoxifen treatment).

We now use Lgr5eGFP-CreERT2 throughout with the exception of image labeling for brevity. We apologize for the inadvertent omission of lineage tracing methodology which we have now added.

Reviewer #3 (Comments to the Authors (Required)):

The authors addressed the majority of my concerns and I believe this paper is very close to ready for publications. I really liked the going back and forth between the ex vivo and the in vivo situation that nicely complemented each other.

We thank the reviewer for this positive assessment of our revised manuscript.

However, I do still have an important editorial criticism. As also pointed out by another reviewer, referring to the hair follicle stem cells in the bulge as the "niche" is incorrect and is both confusing and misleading. While it is true that the bulge is the residence of the HFSC,

and it has been referred to as a "niche" in some older studies, we also now know that the niche of the HFSCs is composed of multiple compartments surrounding the bulge stem cells. The niche is now known to be a complex microenvironment that surrounds the bulge. Calling the bulge stem cells themselves a "niche" is especially confusing in the context of this paper, as the phenotypes described here are stem cell autonomous, or cell-intrinsic, as rightly interpreted and demonstrated by the authors. Therefore the authors should simply refer to the bulge stem cell population as the bulge HFSC "compartment" and not as the stem cell "niche". The latter term should be reserved for the many cells and structures surrounding the stem cells that signal to the bulge stem cells. Those do not seem to be affected here or there is no data analyzing that aspect. This correction should include the abstract and the title and all throughout the paper and the discussion.

We appreciate this criticism and agree. We have followed the suggestion and have replaced “niche” with “compartment”.